# The seminal fluid protein SFP-1 regulates mated hermaphrodite aging and fat metabolism in *C. elegans*

Mingqing Chen [ID] & Jianke Gong [ID] [✉]

## Abstract

**Across the evolutionary spectrum, sexual interactions can significantly influence the physiology and somatic aging in various species. In Caenorhabditis, male pheromones, sperm, and seminal fluid shorten the lifespan of hermaphrodites through different mechanisms. However, the specific male seminal fluid proteins responsible for this effect remain unidentified. Here, we find that several of the previously observed physiological changes in mated hermaphrodites require a newly-identified seminal fluid protein. SFP-1 is packaged into secretory vesicles via the phospholipid scramblases ANOH-1 and ANOH-2 in male seminal vesicles, and after mating taken up by intestinal cells via endocytosis. Within these intestinal cells, the NTF2-like domain of SFP-1 interacts with and activates the transcription factor SKN-1, which induces post-mating somatic fat depletion and lifespan reduction. Together, these results define how a male seminal-fluid protein can trigger mating-induced physiological changes in sexual interactions.**

**Keywords** Seminal Fluid Protein; Longevity; Mating; Lipid Metabolism
**Subject Categories** Signal Transduction; Urogenital System

## Introduction

Sexual interactions are common in many species and can significantly influence the development, metabolism, and aging of both sexes (Gems and Riddle, 1996; Gendron et al, 2014; Hsin and Kenyon, 1999; Palopoli et al, 2015; Wigby and Chapman, 2005). In *Drosophila*, seminal fluid products from males regulate the post-mated aging in females (Chapman et al, 1995). Similarly, in mice, vasectomized males cause the co-house female to gain more body weight and shorten the lifespan, independent of fertilization (Garratt et al, 2020). In *Caenorhabditis elegans*, males impact the hermaphrodites using male sperm and seminal fluid during mating as well as male pheromones and compounds released by males (Ludewig et al, 2019; Maures et al, 2014; Shi and Murphy, 2014). These male components trigger significant alterations in gene expression, metabolic processes, and other physiological changes in hermaphrodites (Booth et al, 2022; Shi and Murphy, 2014). Male

sperm causes the hermaphrodites shrinking and lifespan decreases through DAF-9/DAF-12 dependent pathway (Shi and Murphy, 2014). Moreover, many metabolic genes, including the acyl-CoA binding protein ACBP-3, are upregulated in mated hermaphrodites via sperm (Booth et al, 2022). Seminal fluid components transferred into hermaphrodites can disrupt the nuclear localization of the transcription factor FOXO/DAF-16, a classical lifespan regulation pathway reinforced by INS-7 feedback (Shi and Murphy, 2014). Additionally, these components can influence the expression of neuron-enriched genes, such as *delm-2*, which regulate the metabolism of intestinal fat (Booth et al, 2022). However, the specific components in male seminal fluid affecting hermaphroditic metabolism and lifespan regulation after mating remain mostly unknown.

Seminal fluid contains a wide variety of components, primarily produced by accessory glands, including sugars, peptides, inorganic salts, small organic molecules, and proteins. Particularly, proteins secreted by the seminal vesicle, vas deferens, or other accessory glands play critical roles in regulating sperm motility, capacitation, and immune modulation (Schjenken and Robertson, 2020; Sirot et al, 2014; South and Lewis, 2011). In *Drosophila*, drosomycin, an antifungal protein derived from the ejaculatory ducts, is transferred to females with seminal fluid to induce immunostimulatory properties (Chapman, 2001). Additionally, sex peptide proteins, act as the factors to remote female reproductive behavior and change female post-mating energy balance to enhance egg production (Chapman et al, 2003). The above two examples illustrate the effects on females. The other one of the important functions of these proteins is to play a role in the maturation and activation of sperm. Heparin-binding proteins, which stimulate sperm capacitation in the female reproductive tract, are secreted into seminal fluid by male accessory glands in bulls and rats (Nass et al, 1990). In addition, bovine seminal fluid also contains a variety of bovine seminal plasma proteins modulating spermatozoa maturation through their calmodulin-binding activity (Manjunath and Sairam, 1987). Similarly, in *Caenorhabditis elegans*, seminal fluid proteins exhibit conserved functions. For example, TRY-5, a seminal fluid protease, acts as an extracellular activator to regulate sperm activation (Smith and Stanfield, 2011). Besides modulating female physiological processes and male sperm maturation, other functions of the proteins in seminal fluid have not been studied well.

Here, we reported the identification of a male-derived protein, SFP-1 (F56D2.8), which was expressed in the seminal vesicle,

---

Key Laboratory of Molecular Biophysics of MOE, and College of Life Science and Technology, Huazhong University of Science and Technology, 430074 Wuhan, Hubei, China.
[✉]E-mail: jiankeg@hust.edu.cn

secreted into seminal fluid, and transferred to hermaphrodites during mating. Loss of SFP-1 in males suppressed the mating-induced shortened lifespan in mated hermaphrodites. Furthermore, before seminal fluid ejaculation, SFP-1 is packaged into the secretory vesicles, a process that requires the mammalian phospholipid scramblase TMEM16F homolog ANOH-1 and ANOH-2 in the male seminal vesicles. SFP-1 is taken up by the intestinal cells via endocytosis after mating. In unmated hermaphrodites, ectopic expression of SFP-1 in intestinal cells leads to shrinking, lifespan decrease and fat depletion. Within the intestine, the NTF2-like domain of SFP-1 is crucial for its function. SFP-1 activates the transcription factor SKN-1 to induce post-mating somatic fat loss and shorten the lifespan. In summary, our findings suggest that SFP-1 is a seminal fluid protein that targets the intestinal cells to play a vital role in regulating post-mating longevity and lipid metabolism.

## Results

### SFP-1 is a seminal fluid protein and is upregulated through sexual interactions to regulate lifespan in hermaphrodites

Genome-wide transcriptional studies of mated and unmated males revealed that genes encoding the secreted proteins F56D2.8, K12H6.5, C16C8.10, and F40G9.15, which are expressed in the seminal vesicle, are upregulated after mating (Ebbing et al, 2018; Shi et al, 2017) (Fig. 1A). These proteins, approximately 15 kDa in size, exhibit simple three-dimensional structures. In contrast, *grd-4*, which encodes a hedgehog-like domain protein produced in the male tail, is significantly down-regulated post-mating (Aspock et al, 1999). It is still unknown why these genes express differently after mating. They may contribute to both male and hermaphrodite reproduction and post-mating physiological regulation. Therefore, the protein of unknown function in male seminal fluid has the potential to be an interactive factor for regulating the metabolism and death of hermaphrodites after mating.

Among these regulated genes, F56D2.8, a male-specific protein exclusively expressed in the seminal vesicle, was identified through transcriptomic analysis of cryo-sectioned adult males (Ebbing et al, 2018). Therefore, we named the gene that encodes this secreted protein *seminalfluid protein 1*, *sfp-1*. SFP-1 is a 133-amino acid protein containing an N-terminal signal peptide and a nuclear transport factor 2 (NTF2)-like domain, which plays an important role in the trafficking of macromolecules, ions, and small molecules between the cytoplasm and nucleus (Kennedy et al, 2001; Vognsen et al, 2013) (Fig. 1B). Structural predictions indicate that SFP-1 consists of four beta-sheets, three alpha helices, and five loops (Fig. 1C), suggesting that SFP-1 has the potential to act as an assistant for protein trafficking between the nucleus and cytoplasm.

Even though transcriptome-wide gene expression analysis has identified *sfp-1* as a male-specific gene with expression restricted to the seminal vesicle (Ebbing et al, 2018), we sought to determine its expression and localization patterns. We generated a *sfp-1::yfp* translational reporter to assess the localization and function of SFP-1. In males carrying the *sfp-1::yfp* reporter, the SFP-1::YFP fusion protein was predominantly expressed in the somatic gonad, specifically within the seminal vesicle (Fig. 1D). The male gonad

of *C.elegans* is a long tube. It primarily includes the seminal vesicle, which acts as a storage organ, a subset of somatic gonadal cells surrounding the spermatids. Additionally, the vas deferens serves as a channel for the transport of sperm, while a valve mechanism regulates the movement of sperm between the seminal vesicle and the vas deferens (Lints and Hall, 2009). To confirm the specific expression of SFP-1 in the seminal vesicle, we used MitoTracker (a fluorescent dye) to label spermatids in *sfp-1::yfp* males. We observed that SFP-1::YFP fluorescence surrounded spermatids (Fig. EV1A), confirming its localization to the seminal vesicle. To further determine SFP-1 as a seminal fluid protein, we mated *sfp-1::yfp* males, which contained fluorescently labeled sperm, with hermaphrodites and detected the translocation of SFP-1::YFP into the uterus of mated hermaphrodites (Fig. EV1B). No fluorescence signal was detected in the adult hermaphrodite germline or sperm (Fig. EV1C). Additionally, using the *Psfp-1::SFP-1::YFP* reporter, we noticed that in the seminal vesicle, the mated males showed a higher expression than the unmated males (Fig. 1E,F). To determine endogenous SFP-1 protein levels and localization, we used CRISPR-Cas9 mediated genome editing to fuse mNeonGreen (mNG) to the C-terminus of the SFP-1 protein, encoded by the *sfp-1* gene. Consistent with the *sfp-1::yfp* observations, bright fluorescent signals were detected in the seminal vesicles of *sfp-1::mNG* males, with increased protein accumulation in mated males (Fig. 1G–I). These findings reveal that SFP-1, a seminal fluid protein, is upregulated during mating, suggesting its potential functional role in mediating post-mating physiological effects.

To investigate the functional role of SFP-1 in mated hermaphrodites, we generated a loss-of-function allele of the *sfp-1* gene using CRISPR-Cas9 technology (Fig. 1B). We performed mating assays by mating the wild-type hermaphrodites with either wild-type males or *sfp-1* knockout males. Lifespan analysis revealed that hermaphrodites mated with *sfp-1* mutant males exhibited a significantly prolonged lifespan compared to those mated with wild-type males (Fig. 1J). Conversely, hermaphrodites mated with males overexpressing SFP-1 (*sfp-1::yfp*) showed a significant acceleration in lifespan shortening (Fig. EV1D). Additionally, the brood size of hermaphrodites mated with *sfp-1* mutants was decreased compared to those mated with wild-type males (Fig. EV1E). Notably, post-mating somatic shrinkage, a sperm-triggered phenotype, was still observed in hermaphrodites mated with *sfp*-1 mutants (Fig. EV1F). Moreover, the *sfp-1* mutant male-conditioned plates were still effective in initiating the shortened lifespan in cultured hermaphrodites (Fig. EV1G). These results imply that SFP-1, as a seminal fluid protein, modulates post-mating physiological changes in hermaphrodites without affecting sperm-mediated shrinking or male pheromone production (Maures et al, 2014). Thus, these results demonstrate that SFP-1 is a seminal fluid protein upregulated during sexual interactions, which regulates hermaphrodite longevity through a seminal fluid-dependent mechanism.

### SFP-1 spreads into the intestinal cells from the uterus in mated hermaphrodites via endocytosis

After mating with males, seminal fluid containing SFP-1 is transferred into hermaphrodites, where it disperses within the uterus and may subsequently be transported to other tissues. To test this hypothesis, we used the *glo-4* mutant hermaphrodites,

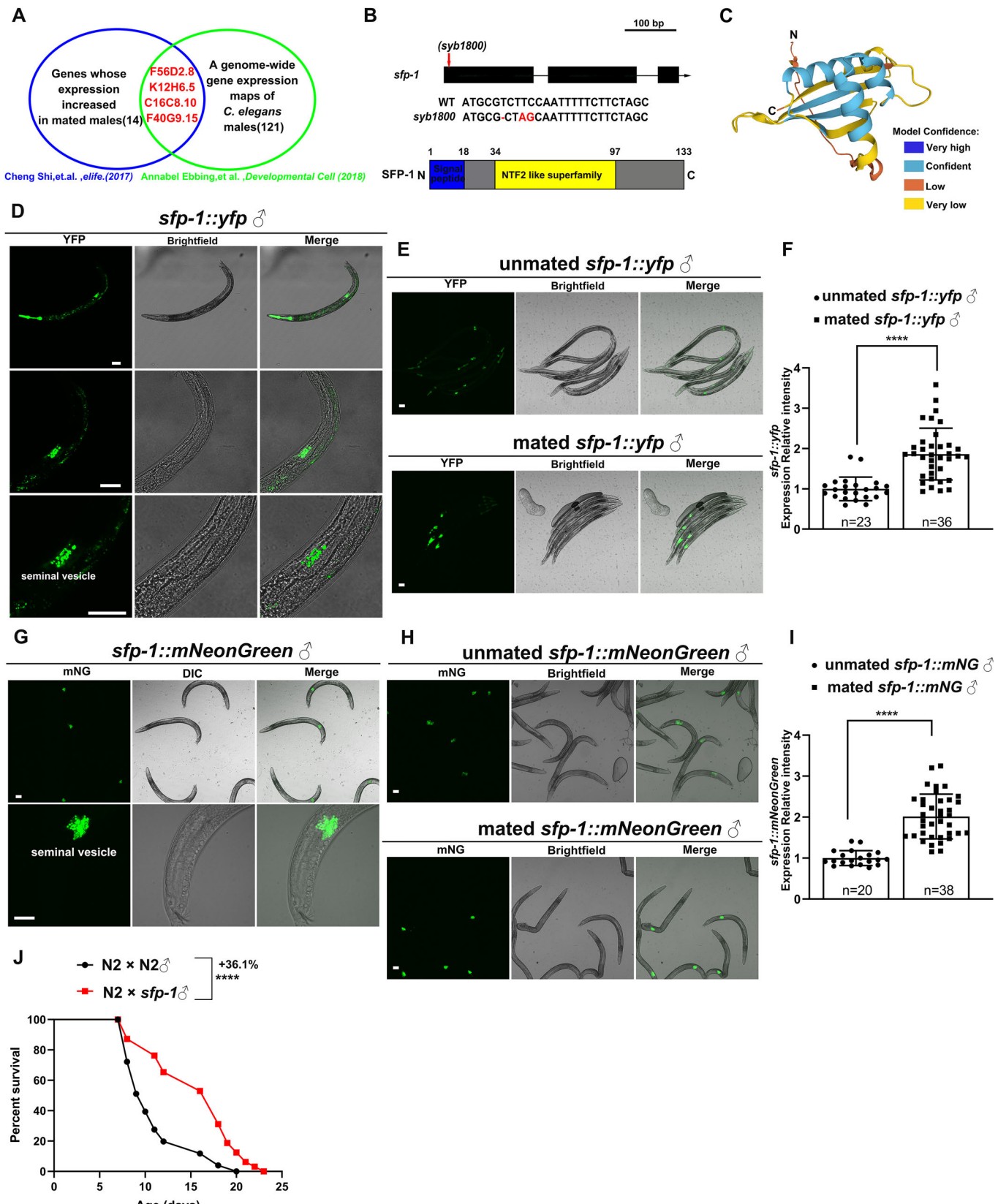

◄ **Figure 1.   The secreted protein SFP-1 is involved in male mating and regulates longevity of mated hermaphrodites.**

(A) Genes that are upregulated in mated males and shared in male-specific genes. (B) Schematic of the genomic location of *sfp-1* mutations. Domain architecture of SFP-1 and fragments, including signal peptide and NTF2L. (C) Predicted structure of SFP-1 generated using the AlphaFold2-Multimer model. (D) Expression of *sfp-1::yfp* in the somatic region of the male reproductive tract, specifically in the seminal vesicle. Due to SFP-1 being exclusively expressed in males, a pharyngeal fluorescence marker was included during microinjection, resulting in *sfp-1::yfp* animals exhibiting pharyngeal fluorescence. Intestinal autofluorescence was also observed. (Scale bars: 50 μm, 20 μm, 10 μm). (E, F) Increased expression of *sfp-1::yfp* in mated males compared to unmated males. Synchronized males were divided into two experimental groups: For unmated controls, males were maintained in groups of 10 worms per 6 cm NGM plate to prevent mating. For mated males, each 3.5 cm NGM plate contained one day 2 hermaphrodite and three day 1 males for 24-h mating. Unmated males were imaged on day 2 of adulthood, while mated males were allowed to mate on day 1 and imaged on day 2. Both groups were imaged under identical exposure conditions. Representative images are shown in (E), and quantification of *sfp-1::yfp* expression is presented in (F). $n \geq 23$ worms. ****$P = 1.3 \times 10^{-7}$ (two-tailed unpaired *t* test). Data are presented as mean ± SD. (Scale bars: 50 μm). (G) Expression of *sfp-1::mNeonGreen* in the somatic region of the male reproductive tract, specifically in the seminal vesicle. (Scale bars: 50 μm, 10 μm). (H, I) Increased expression of *sfp-1::mNeonGreen* in mated males compared to unmated males. Synchronized males were divided into two experimental groups: for unmated controls, males were maintained in groups of 10 worms per 6 cm NGM plate to prevent mating, unmated males were imaged on day 3 of adulthood, while mated males were allowed to mate on day 1 and imaged on day 3. Representative images are shown in (H), and quantification of *sfp-1::mNeonGreen* expression is presented in (I). $n \geq 20$ worms. ****$P = 6.652 \times 10^{-11}$ (two-tailed unpaired *t* test). Data are presented as mean ± SD. (Scale bars: 50 μm). (J) Lifespan of mated N2 worms. N2 × N2 ♂ : 11.01 ± 0.66 days, $n = 30$ worms; N2 × *sfp-1*♂ : 14.98 ± 0.72 days, $n = 30$ worms, ****$P = 4.77 \times 10^{-4}$. Kaplan–Meier analysis with log-rank (Mantel–Cox) method was performed to compare the lifespans between different groups in this study. See Dataset EV1 for all lifespan data summaries.

which lack functional gut granules due to a mutation in a guanine nucleotide exchange factor required for the biogenesis of lysosome-related organelles. (Hermann et al, 2005). This strain allows for better detection of weak fluorescence signals in the intestinal cells compared to the wild-type animals (Zhang et al, 2010). We placed *sfp-1::mNG* males with *glo-4* hermaphrodites, fluorescence imaging revealed that SFP-1 was transferred to hermaphrodites during mating, and then weak fluorescence signals were detected in the intestine cells 0.5 h after mating (Fig. 2A). In contrast, no fluorescence signal was detected in the intestine of *glo-4* mutant hermaphrodites mated with the wild-type males (Fig. 2A). We imaged multiple mated hermaphrodites and observed similar fluorescence signals in the intestinal cells (Fig. EV2A). Similar intestinal fluorescence signals were observed in hermaphrodites mated with *sfp-1::yfp* males (Fig. EV2B). To further confirm the location of SFP-1 in the intestines of mated hermaphrodites, we adopted the split-GFP system which relies on the reconstitution of GFP fluorescence when two non-fluorescent fragments, GFP1-10 and GFP11, are brought into proximity (Hefel and Smolikove, 2019). We inserted a short 16-aa GFP11 fragment in the intestine of *glo-4* mutant hermaphrodites and the complementary GFP1-10 fragment in SFP-1 of males. After mating, fluorescence signals were detected in the intestinal cells of mated hermaphrodites, while no signal was observed in control animals (Fig. 2B). Similar results were observed in multiple mated hermaphrodites (Fig. EV2B), confirming that SFP-1 spreads into intestinal cells after mating.

However, how SFP-1 transfers to intestinal cells remained unclear. In eukaryotic cells, transmembrane transport of macro-molecules and particulate matter, including proteins, polynucleotides, and polysaccharides, is mediated by endocytosis and exocytosis (Agarraberes and Dice, 2001). We hypothesized that SFP-1 transfers to intestinal cells via endocytosis. To test this hypothesis, we treated mated hermaphrodites with Bafilomycin A1 (BafA1), an inhibitor of endocytosis that blocks acidification-dependent vesicle turnover, thereby impairing endosome maturation and disrupting different aspects of endocytosis (Hurtado-Lorenzo et al, 2006; McEwan et al, 2012). BafA1-treated mated hermaphrodites exhibited a significantly longer lifespan compared to the control mated animals (Fig. 2C). Moreover, intestinal-specific RNAi knockdown of key endocytosis genes, such as *cav-1*, and *rab-10* (Chen et al, 2006; Sato et al, 2009; Sato et al, 2006) were

sufficient to protect against mating-induced death (Fig. 2D,E). According to these findings, SFP-1 in the seminal fluid is ejaculated into the uterus and subsequently spreads to intestinal cells via endocytosis in mated hermaphrodites.

## Transportation of SFP-1 requires phospholipid scramblase ANOH-1/ANOH-2 in males

The seminal vesicle (SV) consists of an inner tube composed of approximately 20 identifiable secretory cells, surrounded by a thin layer of cytoplasmic processes derived from three larger cells. Shortly following their differentiation, the smaller cells display a granular appearance when observed using differential interference contrast (DIC) optics, and small blebs are visible on their luminal surfaces (Kimble and Hirsh, 1979). Similar "secretory globules" have been observed in the *Ptry-5::TRY-5::GFP* males (Smith and Stanfield, 2011). In *sfp-1::yfp* males, many large globular structures are generally visible by DIC microscopy (Fig. 3A). Previous studies have shown that the major sperm protein MSP-1 and the seminal fluid protein ENPP-1 are transported via extracellular vesicles (EVs), and SFP-1 has been identified as a potential EV cargo (Kosinski et al, 2005; Nikonorova et al, 2022), suggesting that SFP-1 is transported within bilayer vesicles from the male seminal vesicle. The lipid bilayer formed into vesicles, which contain proteins, RNAs, lipids, and metabolites, mediating the intercellular and inter-tissue transport of biological macromolecules even organelles in most cell types (Melentijevic et al, 2017; Turek et al, 2021). Lipid asymmetry between the inner and outer leaflets of the plasma membrane induces curvature to drive lipid bilayer vesicles release (Wehman et al, 2011; Zwaal et al, 2004). Many enzymes are vital for maintaining bilayer asymmetry. Phospholipid scramblases ANOH-1/ANOH-2 translocate phospholipids from the inner to the outer leaflet and more importantly, trigger the phosphatidylserine-facilitated membrane fusion (Clupper et al, 2022; Furuta et al, 2021; Suzuki et al, 2010; Zhang et al, 2020).

In phospholipid scramblase mutants, we observed a significant reduction in secretory vesicles containing SFP-1 (Fig. 3A), consistent with potential defects in membrane fusion. Quantification of secretory vesicles revealed that both *anoh-1* and *anoh-2* single mutants showed fewer secretory vesicles compared to wild-type males (Fig. 3B). The diffuse distribution pattern of SFP-1 in

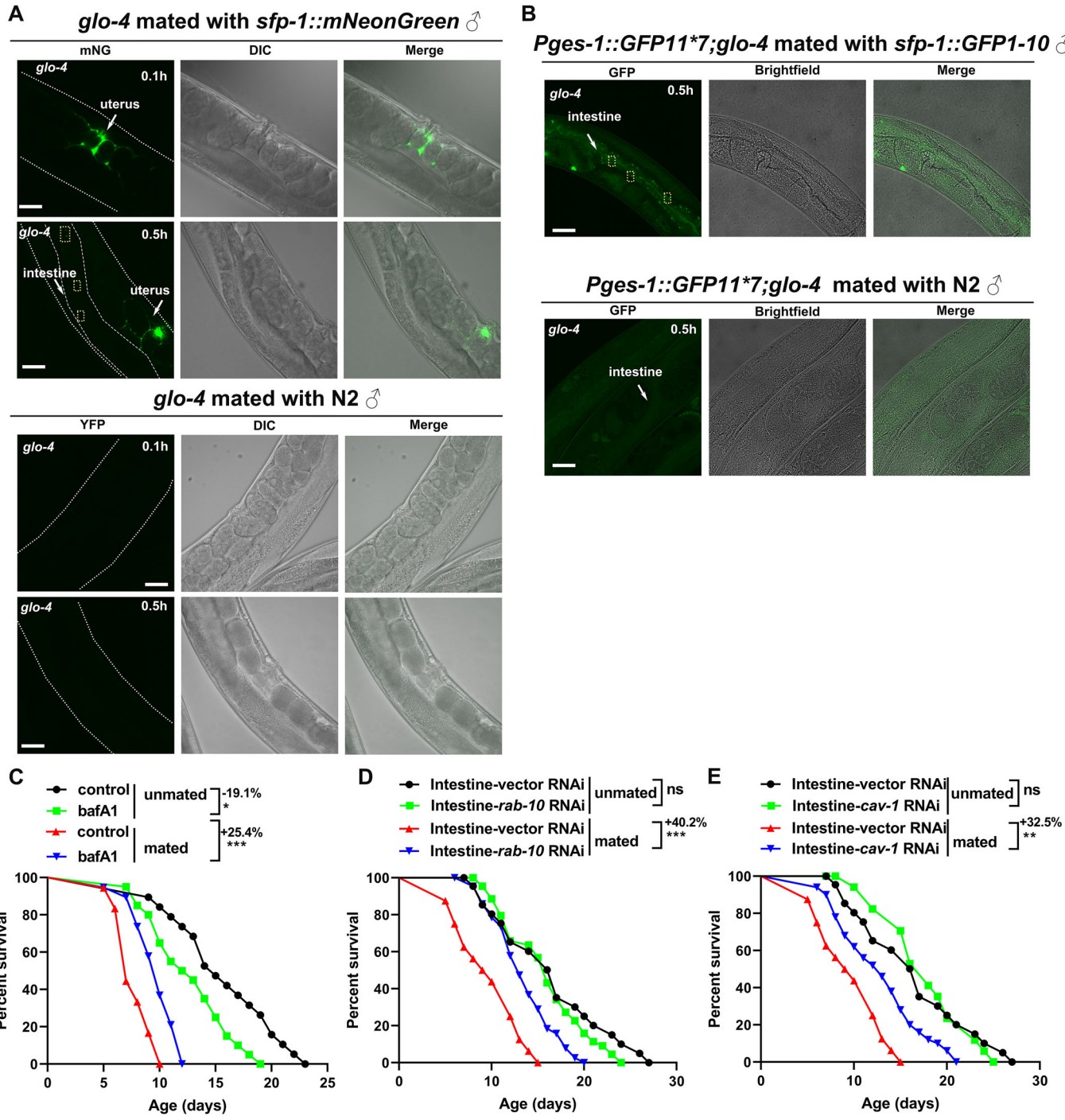

anoh-1; anoh-2 double mutants (Fig. 3B) showed correlation with both reduced vesicle counts and partial suppression of mating-induced lifespan shortening (Fig. 3C). These findings suggest a model where ANOH-1/2-mediated membrane remodeling facilitates SFP-1 packaging, and that disruption of this process correlates with modified post-mating responses. However, further studies are needed to establish whether this represents a direct packaging mechanism or reflects broader roles in reproductive function.

## Ectopic expression of SFP-1 in the intestinal cells imitates the post-mating phenotypes

Mating is a complex process, we demonstrated that seminal fluid protein SFP-1 is transferred to hermaphrodites and regulates longevity after mating. However, we were unable to exclude the influence of other factors during mating. Notably, we have observed the translocation of SFP-1 into intestinal cells, indicating that SFP-1 may have a functional role in the intestine after mating. To test

**Figure 2.   SFP-1 spreads from the uterus into the intestinal cells in mated hermaphrodites via endocytosis.**

(A) Male-to-hermaphrodite transfer experiment where *sfp-1::mNeonGreen* males were mated with low autofluorescence *glo-4* hermaphrodites. Imaging of the low autofluorescence *glo-4* hermaphrodites after copulation revealed the presence of male derived *sfp-1::mNeonGreen* in the uterus, and half an hour after mating, lots of weak fluorescence appeared in the intestinal cell. No fluorescence signal was observed in the intestine cells of *glo-4* mutant when mated with N2 males. The yellow rectangle delineates the fluorescent signal observed in the intestine. (Scale bars: 10 μm). (B) Male-to-hermaphrodite transfer experiment where *sfp-1::GFP1-10* males were mated with *Pges-1::GFP11\*7;glo-4* hermaphrodites. Half an hour after mating, lots of bright fluorescence appeared in the intestinal cell. No fluorescence signal was observed in the intestine cells of mated *Pges-1::GFP11\*7;glo-4* hermaphrodites when mated with N2 males. Autofluorescence of some sperm was observed. The yellow rectangle delineates the fluorescent signal observed in the intestine. (Scale bars: 10 μm). (C) Lifespan of mated and unmated N2 worms treated with the drug. N2 treated with DMSO: 15.63 ± 1.01 days, $n = 19$ worms; N2 treated with bafA1: 12.65 ± 0.79 days, $n = 20$ worms; N2 × N2 ♂ treated with DMSO: 7.72 ± 0.35 days, $n = 18$ worms; N2 × N2 ♂ treated with bafA1: 9.68 ± 0.44 days, $n = 19$ worms, *$P = 0.014$, ***$P = 0.0007$, comparisons were made using the Log-rank (Mantel–Cox) test. (D) Lifespan of mated and unmated intestine-specific *rab-10* RNAi. VP303-vector RNAi-unmated: 16.44 ± 1.29 days, $n = 20$ worms; VP303-*rab-10* RNAi-unmated: 15.84 ± 0.65 days, $n = 44$ worms; VP303-vector RNAi-mated: 9.63 ± 0.85 days, $n = 16$ worms; VP303-*rab-10* RNAi-mated: 13.49 ± 0.51 days, $n = 39$ worms, ***$P = 0.0002$, comparisons were made using the Log-rank (Mantel–Cox) test. (E) Lifespan of mated and unmated intestine-specific *cav-1* RNAi. VP303-vector RNAi-unmated: 16.44 ± 1.29 days, $n = 20$ worms; VP303-*cav-1* RNAi-unmated: 17.76 ± 1.06days, $n = 17$ worms; VP303-vector RNAi-mated: 9.63 ± 0.85 days, $n = 16$ worms; VP303-*cav-1* RNAi-mated: 12.76 ± 0.62 days, $n = 50$ worms, **$P = 0.003$, comparisons were made using the Log-rank (Mantel–Cox) test.

this hypothesis and specifically focus on the effects of SFP-1, we ectopically expressed full-length SFP-1 protein in four major tissues, including the intestine, germline, muscle, and neuron, utilizing their specific promoters. Only intestinal-specific expression of SFP-1 resulted in significant lifespan reduction (Fig. 4A). Furthermore, we observed that intestinal overexpression of SFP-1 in unmated hermaphrodites led to increased brood size (Fig. EV1E). Meanwhile, SFP-1 induced the shrinking phenotype through the intestinal cell in unmated hermaphrodites (Fig. EV1F), indicating that SFP-1 may activate the germline in the gut signal pathway to induce shrinking.

To determine whether the overexpression of SFP-1 in the intestine requires an intact germline to mediate lifespan reduction in self-fertilized hermaphrodites, we specifically overexpressed SFP-1 in the intestinal cells of the germline-deficient *glp-1(e2141)* mutants (Arantes-Oliveira et al, 2002). We found that the expression of SFP-1 significantly shortened the lifespan of these long-lived mutants (Fig. EV3A), indicating that intestinal overexpression of SFP-1 does not require an intact germline to mediate the reduction in lifespan of self-fertilized hermaphrodites.

To determine whether the signal peptide of the SFP-1 mediates protein secretion during ectopic expression, we generated *Pmyo-3::sfp-1::yfp* and *Pges-1::sfp-1::yfp* transgenic worms. In *Pmyo-3::sfp-1::yfp* animals, we observed bright fluorescent signals in coelomocytes, consistent with previous studies showing GFP secretion into coelomocytes when attached to a signal sequence and expressed in body wall muscles (Fares and Greenwald, 2001; Treusch et al, 2004). Importantly, ectopic expression of full-length SFP-1::YFP in either intestine or muscle did not result in detectable secretion to non-target tissues beyond the expected coelomocyte uptake (Fig. EV3B). We also ectopically expressed other seminal fluid proteins, such as K12H6.5 and F40G9.15 in the intestinal cells, however, we did not observe any shortened lifespan phenotype (Fig. EV3C). Among the four proteins screened, only SFP-1 contains an NTF2-like domain. We hypothesize that the negative effects observed in animals ectopically expressing SFP-1 in the intestine may be due to the presence of the NTF2-like domain. Together, our data show that intestinal SFP-1 ectopically expressed in intestinal cells is sufficient to induce key post-mating phenotypes observed in mated animals, including lifespan reduction, increased brood size, and shrinking.

## The NTF2-like domain is required for SFP-1 nuclear localization which causes post-mating like phenotypes

According to the blast analysis, nearly half of the full-length amino acid sequence of SFP-1 forms a nuclear transport factor 2-like domain (NTF2L), classifying SFP-1 within the NTF2-like superfamily. The NTF2-like superfamily consists of a wide variety of proteins present in bacteria, archaea, and eukaryotes (Aibara et al, 2015; Niikura et al, 2024). The NTF2-like domain is evolutionarily conserved and serves as a mediator for enzyme catalysis and protein-protein interactions within the nucleus (Kennedy et al, 2001). To identify the functional domain of SFP-1, we examined the lifespan of animals overexpressing three different SFP-1 fragments respectively and found that the absence of NTF2L and C-terminal fragment (98-132 aa) eliminated the function of SFP-1 to mediate shortened lifespan in ectopically expressed worms (Fig. 4B,C). Besides, the absence of NTF2L in SFP-1 failed to induce fat loss and increased brood size, which were observed in animals that overexpressed full-length SFP-1 (Fig. EV3D–F). Furthermore, in worms carrying the *Pges-1::sfp-1::yfp*, the full-length SFP-1 was localized around the nucleus of gut cells, where the NTF2L family protein presented exerts its function, in contrast, the NTF2L truncation of SFP-1 remains in the cytoplasm (Fig. 4D,E). Therefore, our results suggest that the NTF2-like domain is essential for nuclear translocation and leads to the post-mating like phenotypes.

## SFP-1 is associated with the translocation of the FOXO transcription factor homolog DAF-16 after mating

Previous studies have established that seminal fluid triggers DAF-16::GFP nuclear export, thereby suppressing longevity-promoting genes in mated *daf-2* hermaphrodites (Shi and Murphy, 2014). To investigate whether SFP-1 mediates this process, we established an experimental system using *daf-2* RNAi-treated hermaphrodites where DAF-16::GFP shows constitutive nuclear localization. This sensitized background allowed precise detection of mating-induced DAF-16 translocation. In contrast to wild-type males, which caused significant cytoplasmic localization of DAF-16, hermaphrodites mated with *sfp-1* mutant males remained predominantly nuclear DAF-16::GFP fluorescence (Fig. 5B), demonstrating that SFP-1 is required for mating-induced DAF-16 nuclear export.

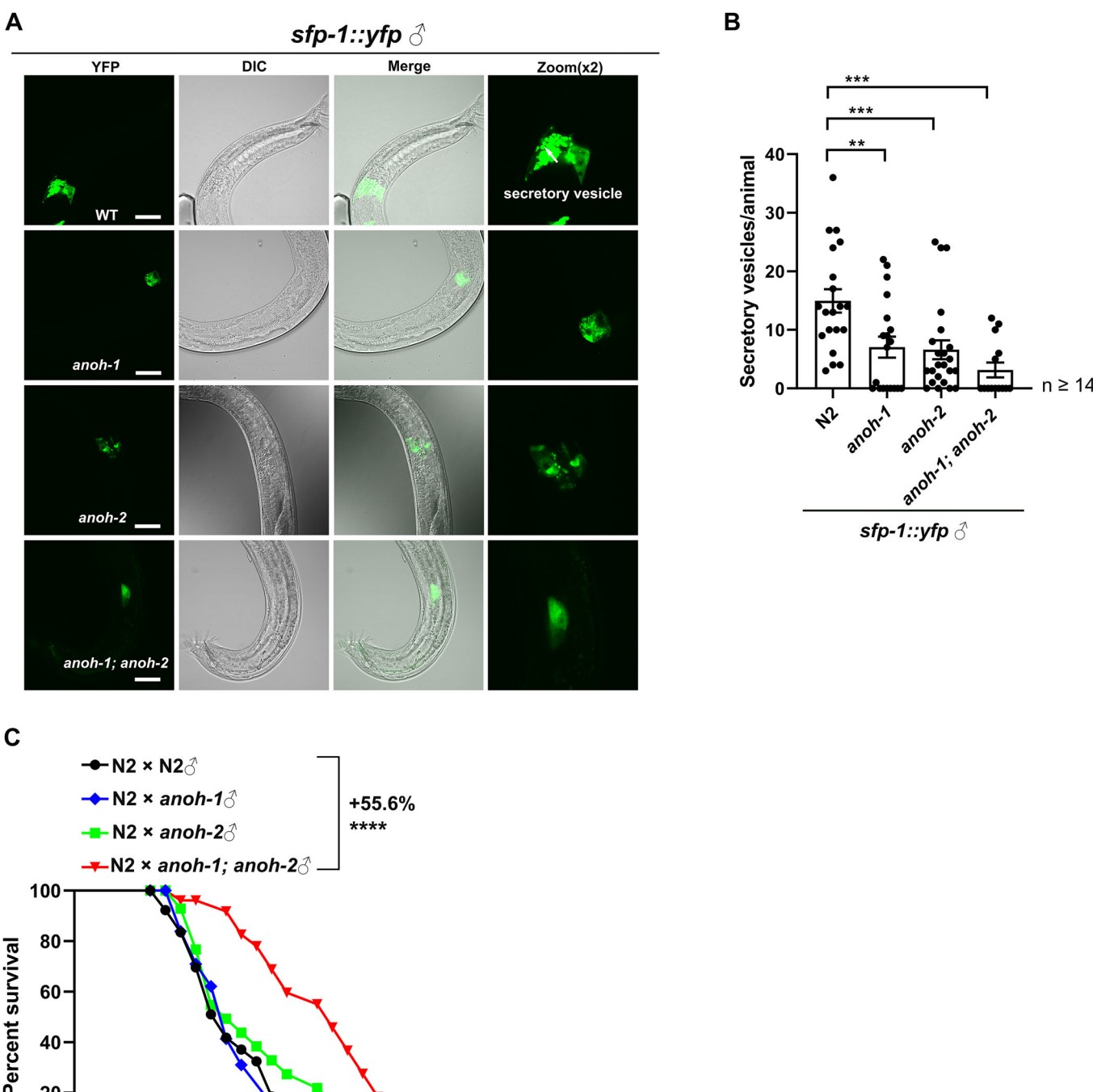

**Figure 3.  ANOH-1/ANOH-2 phospholipid scramblase is required for SFP-1 transportation in male *C. elegans*.**

(A) Representative images of SFP-1::YFP localization in secretory vesicles in wild-type (WT), *anoh-1*, *anoh-2*, and *anoh-1; anoh-2* double mutant males. (Scale bars: 10 μm). (B) Quantification of secretory vesicle genesis levels in males. Data are presented as mean ± SEM. Sample sizes: WT, *n* = 20; *anoh-1*, *n* = 19; *anoh-2*, *n* = 23; *anoh-1; anoh-2*, *n* = 14. For *anoh-1, anoh-1; anoh-2*, statistical significance was determined by one-way ANOVA followed by Bonferroni's multiple comparisons test, **$P$ = 0.0060, ***$P$ = 0.0001. For *anoh-2*, statistical significance was determined by Mann–Whitney test, ***$P$ = 0.0003. (C) Lifespan of mated N2 worms. N2 × N2 ♂: 10.45 ± 0.64 days, *n* = 22 worms; N2 × *anoh-1* ♂: 10.52 ± 0.56 days, *n* = 23 worms; N2 × *anoh-2* ♂: 11.93 ± 0.93 days, n = 20 worms; N2 × *anoh-1; anoh-2* ♂: 16.26 ± 0.91 days, *n* = 22 worms, ****$P$ = 2.28 × 10^{-6}, comparisons were made using the Log-rank (Mantel–Cox) test (Compared to hermaphrodite mated with N2 male).

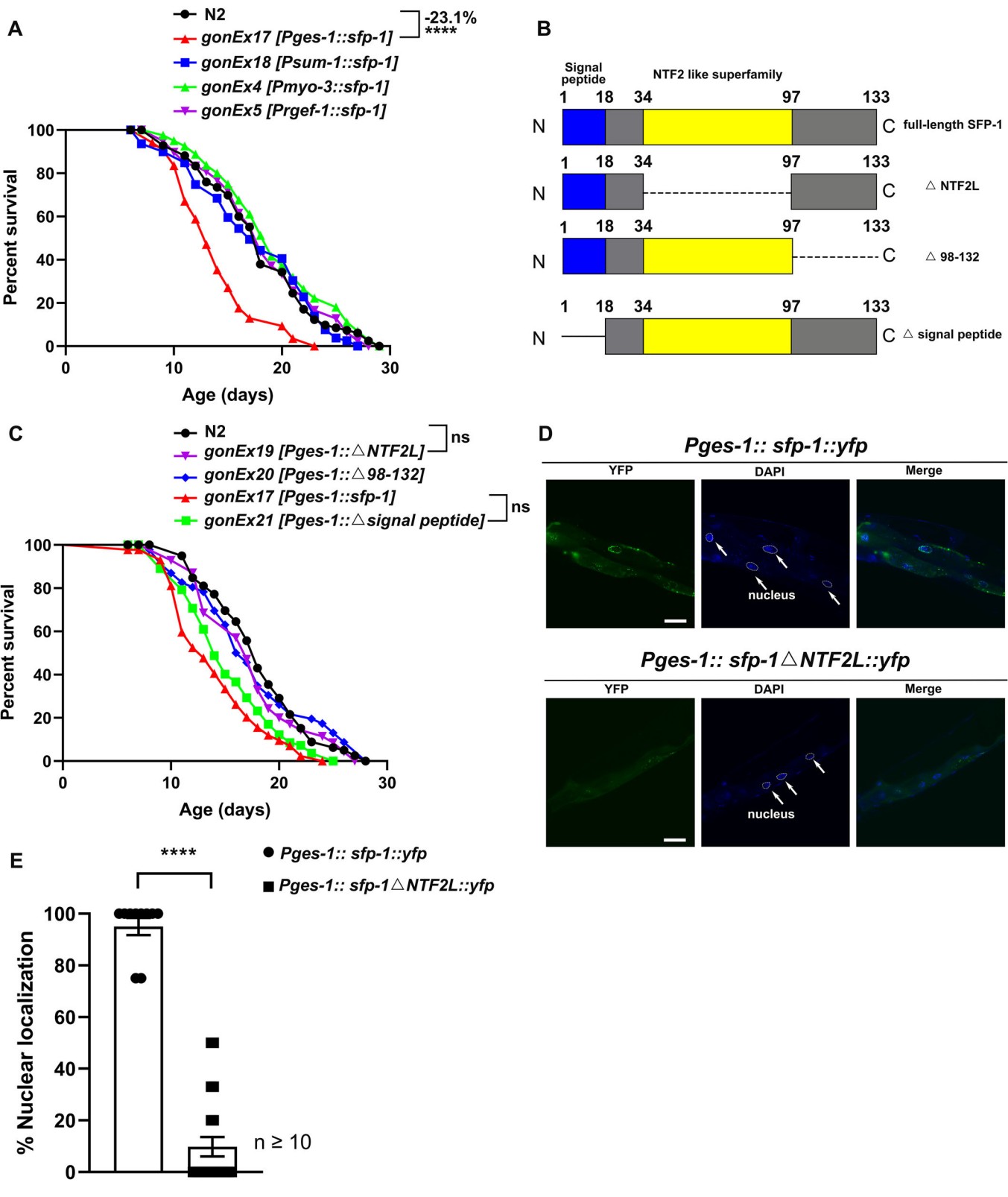

**Figure 4. Only intestinal overexpression of SFP-1 shortens lifespan in the transgene line.**

(A) Lifespan survival curves of WT animals and transgenic strains expressing SFP-1 in specific tissues under the *Pges-1* (intestine), *Pmyo-3* (muscle), or *Prgef-1* (pan-neuronal) promoters. For germline-specific expression, we employed the engineered *smu-1* system (Aljohani et al, 2020), which utilizes optimized germline promoters (*Psmu-2*, *Ppie-1*, or *Pmex-5*) coupled with stabilizing 3'-UTRs (*smu-2/tbb-2*) to ensure persistent transgene expression. Ectopic expression of SFP-1 in intestinal cells significantly shortened lifespan, while expression in other tissues did not affect longevity. N2: 17.90 ± 0.57 days, n = 82 worms; *gonEx17 [Pges-1::sfp-1]*: 13.77 ± 0.41 days, n = 85 worms; *gonEx18 [Psmu-1::sfp-1]*: 17.44 ± 0.61 days, n = 80 worms; *gonEx4 [Pmyo-3::sfp-1]*: 19.13 ± 0.60 days, n = 75 worms; *gonEx5 [Prgef-1::sfp-1]*: 18.17 ± 0.59 days, n = 78 worms. ****P = 4.95 × 10⁻⁹, comparisons were made using the Log-rank (Mantel–Cox) test. (B) Schematic representation of the domain architecture of SFP-1, highlighting the signal peptide and NTF2-like (NTF2L) domain. Δ, deletion. (C) Lifespan survival curves of WT animals and transgenic strains expressing truncated variants of SFP-1. Deletion of the signal peptide did not rescue the shortened lifespan phenotype, while deletion of the NTF2L domain abolished the lifespan reduction. N2: 18.04 ± 0.49 days, n = 79 worms; *gonEx17 [Pges-1::sfp-1]*: 13.97 ± 0.45 days, n = 84 worms; *gonEx19 [Pges-1::ΔNTF2L]*: 17.30 ± 0.55 days, n = 70 worms; *gonEx20 [Pges-1::Δ98-132]*: 17.48 ± 0.84 days, n = 46 worms; *gonEx21 [Pges-1::Δsignal peptide]*: 15.20 ± 0.47 days, n = 82 worms. Comparisons were made using the Log-rank (Mantel–Cox) test. (D, E) Representative images of nuclear localization in day 1 *Pges-1::sfp-1::yfp* and *Pges-1::sfp-1ΔNTF2L::yfp* animals. (Scale bars: 10 μm). (E) Quantification of the nuclear localization of intestinal cells per individual. *Pges-1::sfp-1::yfp*: n = 10, *Pges-1::sfp-1ΔNTF2L::yfp* n = 21. Statistical significance was determined by Mann–Whitney test. ****P = 2 × 10⁻⁸. Data are presented as mean ± SEM.

Further lifespan analyses uncovered additional regulatory complexity. When compared to unmated *daf-16* mutants, hermaphrodites mated with *sfp-1* mutant males still exhibited lifespan shortening, indicating that sperm from *sfp-1* mutants retains the partial capacity to trigger post-mating lifespan reduction (Fig. 5A). However, this reduction was significantly attenuated relative to mate with wild-type males, demonstrating that SFP-1 deficiency provides partial protection against mating-induced lifespan shortening in *daf-16* mutants (Fig. 5A). These findings suggest that SFP-1 mediates post-mating longevity through multiple pathways, including but not limited to the DAF-16-dependent transcriptional pathway. Specifically, these results indicated that SFP-1 facilitates DAF-16 nuclear export, thereby contributing partially to mating-induced lifespan reduction, while additional mechanisms such as lipid depletion may also play a role in mating-induced death.

## SFP-1 activates the Nrf2 transcription factor homolog SKN-1 after mating

These results revealed the relationship between SFP-1 and DAF-16/FOXO pathway, suggesting that SFP-1 may interact with multiple transcription factors to mediate post-mating phenotypes. To identify the transcription factors to regulate longevity in intestinal overexpressed SFP-1 animals, we performed RNAi screening for well-known transcription factors in aging, including *daf-16*, *hsf-1*, *pha-4*, *skn-1* (Kenyon, 2010) and *daf-12*, a known regulator of lifespan and energy metabolism (Wang et al, 2015). Additionally, we examined the transcription factors PQM-1 and CEH-60, which have been implicated in regulating the longevity of mated hermaphrodite (Booth et al, 2022). We hypothesized that the knockdown of transcription factors mediating SFP-1's effects would significantly attenuate or fully reverse the lifespan reduction caused by intestinal SFP-1 overexpression (Fig. EV4A–D). The lifespan analysis indicated that SKN-1 is important in regulating the lifespan of animals expressing SFP-1 in intestinal cells ectopically. Supporting this observation, we found that *skn-1* loss-of-function mutant hermaphrodites mated with wild-type males showed significantly shortened lifespan, while mating with *sfp-1* mutant males did not further reduce lifespan (Fig. 5C). These results demonstrate that SKN-1 is essential for mediating the SFP-1-dependent lifespan reduction during mating. Consistent with this finding, intestinal overexpression of SFP-1 did not further shorten the lifespan under the *skn-1* mutant, even the *skn-1* mutant had a

shorter lifespan (Fig. 5E), suggesting that SKN-1 acts downstream of SFP-1 in regulating mating-induced lifespan modulation. These findings reveal a complex regulatory relationship between SFP-1 and SKN-1. While intestinal overexpression of SFP-1 failed to further reduce lifespan in *skn-1* mutants, the unexpected lifespan extension observed when SFP-1 was overexpressed in *skn-1* mutants suggests the existence of compensatory regulatory mechanisms.

To further clarify the mechanism of SKN-1 regulation, we examined SKN-1 nuclear localization using a well-characterized strain expressing GFP-tagged SKN-1 protein (SKN-1B/C::GFP) (An and Blackwell, 2003). We observed robust nuclear accumulation of SKN-1::GFP in intestinal cells of SFP-1 overexpressing animals (Fig. 5D). Consistent with this finding, we further evaluated SKN-1 activity by measuring the fluorescence of animals expressing GFP downstream of the *gst-4* promoter (*Pgst-4::GFP*), a well-established reporter of SKN-1 (Link and Johnson, 2002), demonstrated significantly enhanced SKN-1 activity in intestinal SFP-1 overexpressing animals compared to wild-type animals (Fig. 5F).

Given that the MAPK signaling pathway regulates SKN-1 activation (Blackwell et al, 2015), we investigated the role of MAPK components in this process. Knockdown of *sek-1* and *pmk-1*, conserved MAPK pathway components, extended the lifespan of intestinal SFP-1 overexpressing animals, which was similar to the results observed in *skn-1* mutants (Inoue et al, 2005) (Fig. EV4E,F), suggesting that SFP-1 activates SKN-1 through MAPK signaling.

To investigate SKN-1's involvement in SFP-1-mediated regulation of DAF-16, we first analyzed DAF-16 nuclear export dynamics. Our results showed that *skn-1*(RNAi) effectively blocked the post-mating cytoplasmic translocation of DAF-16::GFP (Fig. 5G). To further assess the function of this regulation, we measured the expression of the canonical DAF-16 reporter *Psod-3::gfp* (Libina et al, 2003). *skn-1*(RNAi) abolished the enhanced *Psod-3::gfp* expression induced by mating with *sfp-1* males (Fig. 5H). These findings demonstrate that SKN-1 is required for both the nuclear localization and transcriptional suppression of DAF-16 in response to SFP-1.

Together, these findings establish a molecular pathway in which SFP-1 activates SKN-1, likely through MAPK signaling, to regulate post-mating lifespan reduction and modulate DAF-16 localization and activity, thereby positioning SKN-1 as a key downstream effector of SFP-1 in coordinating post-mating responses.

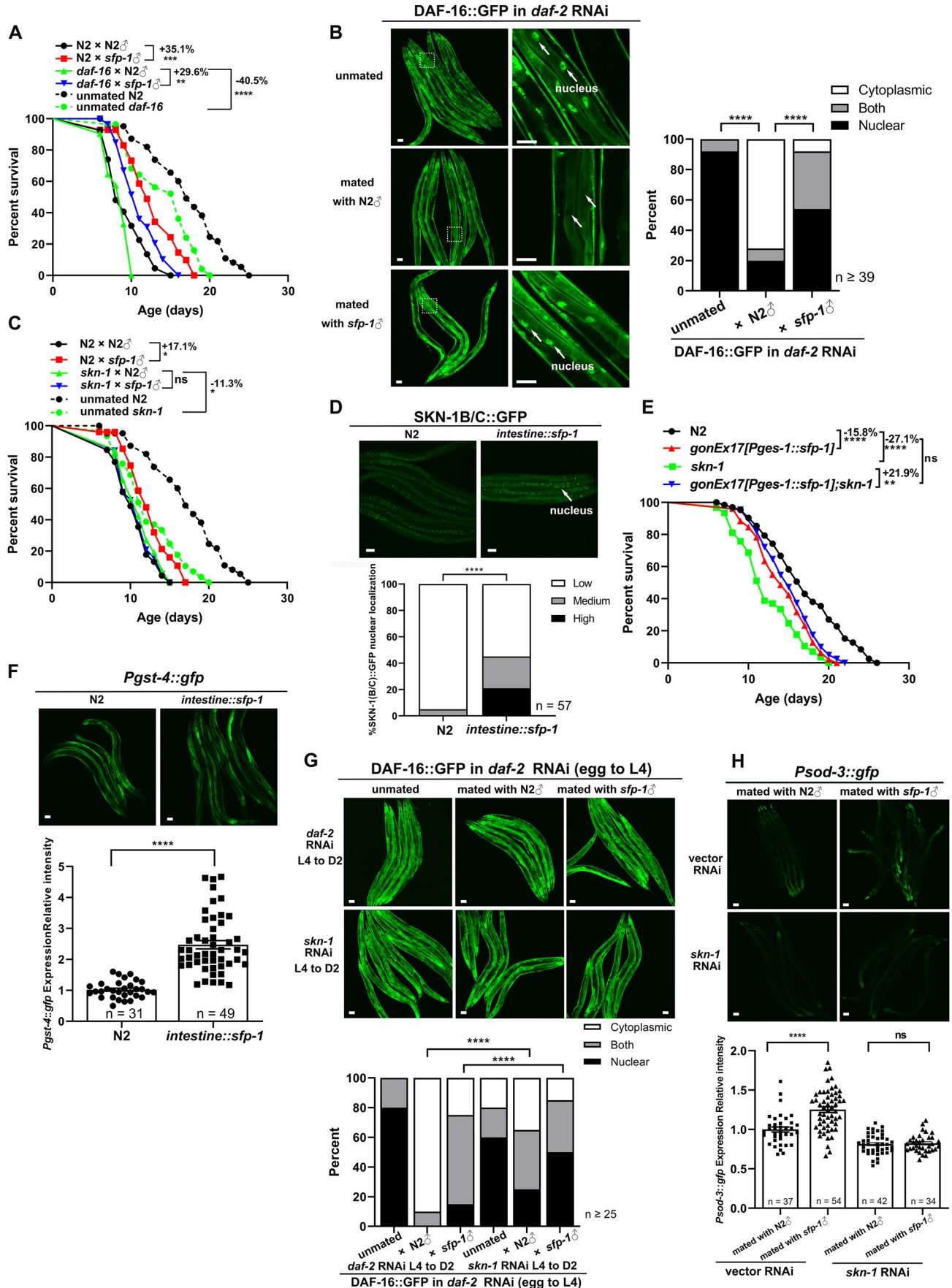

**Figure 5.  SFP-1 activates the Nrf2 transcription factor homolog SKN-1 after mating.**

(A) Lifespan survival curves of WT animals and *daf-16* mutant animals mated with WT males or *sfp-1* males. N2 × N2: 9.33 ± 0.49 days ($n = 25$); N2 × *sfp-1*: 12.60 ± 0.72 days ($n = 21$); *daf-16* × N2: 8.45 ± 0.36 days ($n = 12$); *daf-16* × *sfp-1*: 11.08 ± 0.55 days ($n = 21$). ***$P = 0.0004$ (N2 × N2 vs. N2 × *sfp-1*), **$P = 0.0019$ (*daf-16* × N2 vs. *daf-16* × *sfp-1*). Unmated N2: 17.18 ± 0.77 days ($n = 37$); unmated *daf-16*: 14.31 ± 0.76 days ($n = 25$), ****$P = 1.11 \times 10^{-6}$ (unmated *daf-16* vs. *daf-16* × N2). Comparisons were made using the Log-rank (Mantel–Cox) test. (B) Localization of DAF-16::GFP in adult day 5 animals cultured on *daf-2* RNAi bacteria. Representative images (×10 magnification) and quantitation of DAF-16::GFP localization are shown. Each worm was categorized based on DAF-16::GFP localization, and significance was determined by chi-square test: ****$P < 1 \times 10^{-15}$. (Scale bars: 100 μm, 10 μm). (C) Lifespan survival curves of WT animals and *skn-1* mutant animals mated with WT males or *sfp-1* males. N2 × N2: 10.39 ± 0.48 days ($n = 24$); N2 × *sfp-1*: 12.35 ± 0.63 days ($n = 19$); *skn-1* × N2: 11.00 ± 0.45 days ($n = 26$); *skn-1* × *sfp-1*: 10.74 ± 0.51 days ($n = 19$), *$P = 0.016$ (N2 × N2 vs. N2 × *sfp-1*). Unmated N2: 17.18 ± 0.77 days ($n = 37$); unmated *skn-1*: 12.40 ± 0.50 days ($n = 57$), *$P = 0.019$ (unmated *skn-1* vs. *skn-1* × N2). Significance was determined by the Log-rank (Mantel–Cox) test. (D) Effect of intestine-specific expression of *sfp-1* on nuclear localization of SKN-1::GFP. Representative images of day 1 *ldIs7 [skn-1B/C::GFP + pRF4(rol-6(su1006))]* in wild-type and ectopically SFP-1-expressing worms are shown. White arrows indicate nuclear SKN-1::GFP localization. Bottom panel: Quantification of nematodes with "low," "medium," or "high" nuclear SKN-1::GFP accumulation. Data are from 57 animals and significance was determined by chi-square test: ****$P = 2.63 \times 10^{-10}$. Scale bars: 50 μm. (E) The short-lived lifespan phenotype of *intestine::sfp-1* transgenic worms can be suppressed by *skn-1* mutation. N2: 17.01 ± 0.45 days, $n = 119$ worms; *gonEx17 [Pges-1::sfp-1]*: 14.33 ± 0.50 days, $n = 48$ worms; *skn-1 (zu135)* : 12.40 ± 0.50 days, $n = 57$ worms; *gonEx17 [Pges-1::sfp-1];skn-1 (zu135)* : 15.11 ± 0.52 days, $n = 41$ worms. ****$P = 3.55 \times 10^{-5}$, ****$P = 8.14 \times 10^{-11}$, **$P = 0.0019$ (from left to right). Significance was determined by the Log-rank (Mantel–Cox) test. (F) Fluorescent images of SKN-1 reporter animals in wild-type and SFP-1 expressed ectopically worms. Quantification of SKN-1 activation (SFP-1 expressed ectopically worms normalized to mean of wild-type), data shown are representative of $N = 3$ biological replicates with $n > 30$ animals per condition for each replicate. Statistical significance was determined by Mann–Whitney test. ****$P < 1 \times 10^{-15}$. Error bars: SEM. (Scale bars: 100 μm). (G) SKN-1 mediates SFP-1-dependent regulation of DAF-16 nuclear export after mating. (Top) Experimental timeline: DAF-16::GFP worms were cultured on *daf-2*(RNAi) to establish nuclear localization, then divided at L4 into *daf-2*(RNAi) or *skn-1*(RNAi) groups. Both were mated with N2 or *sfp-1* males on day 3 and imaged on day 4. (Bottom) Representative confocal images (left to right): Unmated controls; mating with N2 males; mating with *sfp-l* males. Scale bars: 100 μm. Top row: *daf-2*(RNAi), bottom row: *skn-1*(RNAi). (Right) Quantification of nuclear-to-cytoplasmic (N/C) ratios ($n \geq 25$ worms). ****$P = 3 \times 10^{-15}$, ****$P = 8.63 \times 10^{-7}$ (from left to right; chi-square test). *skn-1*(RNAi) blocks mating-induced export. (H) SKN-1 is required for SFP-1-mediated suppression of DAF-16 transcriptional activity after mating. *Psod-3*::GFP reporter expression in unmated worms vs. worms mated with N2 or *sfp-1* males. Scale bars: 100 μm. Quantification shows increased *Psod-3*::GFP after *sfp-l* mating (****$P = 2.55 \times 10^{-8}$ vs. mated with N2), suppressed by *skn-1* RNAi (ns $P > 0.05$ vs. mated with N2). Data: mean ± SEM ($n \geq 34$ worms). $P$ values were calculated using one-way ANOVA followed by Bonferroni's multiple comparisons test.

## SFP-1 regulates post-mating lipid metabolism and triggers Asdf through SKN-1

The transcription factor SKN-1 is critical in regulating lipid metabolism, thereby modulating health and lifespan (Blackwell et al, 2015). Hermaphrodites exhibit significant fat loss after mating, a process previously shown to be mediated by seminal fluid (Shi and Murphy, 2014). We found that SFP-1 enhanced SKN-1 activity, suggesting that SFP-1 might be the key seminal fluid component responsible for mating-induced fat loss. To investigate this hypothesis, we performed Oil Red O staining and triacylglycerol (TAG) analysis on hermaphrodites mated with either wild-type or *sfp-1* mutant males. Hermaphrodites mated with *sfp-1* mutants showed reduced fat loss compared to those mated with wild-type males, while hermaphrodites mated with *sfp-1::yfp* males, which overexpresses SFP-1 in the seminal vesicle, resulted in accelerated fat loss (Figs. 6A,B and EV4G). In addition, intestinal overexpression of SFP-1 in unmated animals was sufficient to induce fat loss (Figs. 6C,D and EV4G), indicating that SFP-1 may play an important role in lipid metabolism.

To further confirm the role of SKN-1 in mating-induced fat loss, we examined fat storage in *skn-1* mutant hermaphrodites mated with either wild-type or *sfp-1* mutant males. Indeed, the *skn-1* mutants effectively prevented fat loss no matter whether they were mated with wild-type or *sfp-1* males (Fig. 6E,F). Furthermore, intestinal overexpression of SFP-1 in the *skn-1* mutant background led to significantly greater fat accumulation compared to SFP-1 overexpression in N2 animals, resembling the phenotype observed in *skn-1* mutants (Fig. 6G,H). These findings indicated that SFP-1 plays a critical role in mediating fat loss, which is dependent on SKN-1 activity.

Surprisingly, when zooming in to observe the staining images, we found that fat was depleted severely in the somatic cells compared to the germline cells in the intestinal SFP-1

overexpressing animals in the wild-type background, while this was not observed in *skn-1* mutant background, which is known as age-dependent somatic depletion of fat (Asdf) (Lynn et al, 2015; Nhan et al, 2019) (Fig. 6I,J). This phenotype is triggered by SKN-1 overactivation or age-related lipid metabolism indeed worms were in the adult day 4 stage not aging yet. Given that Asdf is mediated by vitellogenin family proteins (Lynn et al, 2015), we sought to determine whether SFP-1-induced Asdf is dependent on vitellogenin function. To address this, we specifically overexpressed SFP-1 in the intestinal cells of *vit-1* and *vit-2* mutants. We found that the Asdf phenotype observed in the intestinal SFP-1 overexpressing animals was suppressed in both *vit-1* and *vit-2* mutant backgrounds (Fig. EV4H–K). To further confirm the role of vitellogenin in the shortened lifespan of intestinal SFP-1-overexpressing animals, we used RNAi to knock down *vit* genes and performed lifespan experiments. These results revealed that knockdown of the *vit* gene in the intestinal SFP-1 overexpressing animals did not affect lifespan (Fig. EV4J–O). These results demonstrate that SFP-1 activates SKN-1 to regulate post-mating lipid metabolism in intestinal cells, with vitellogenin proteins serving as essential downstream effectors of lipid distribution.

## SFP-1 leads to PUFA depletion in the intestinal cells

Consistent with the fat loss observed in mated hermaphrodites, ectopic expression of SFP-1 in intestinal cells of unmated hermaphrodites also exhibited lipid depletion. To investigate whether SFP-1 influences fatty acid composition, we performed lipidomic profiling of free fatty acids (FFAs). The analysis revealed that ectopic expression of SFP-1 specifically reduced polyunsaturated fatty acids (PUFAs), while levels of saturated fatty acids (SFAs) and monounsaturated fatty acids (MUFAs) remained unchanged (Fig. 7A). To further explore the role of SFP-1 in post-mating lipid metabolism, we compared fatty acid content in

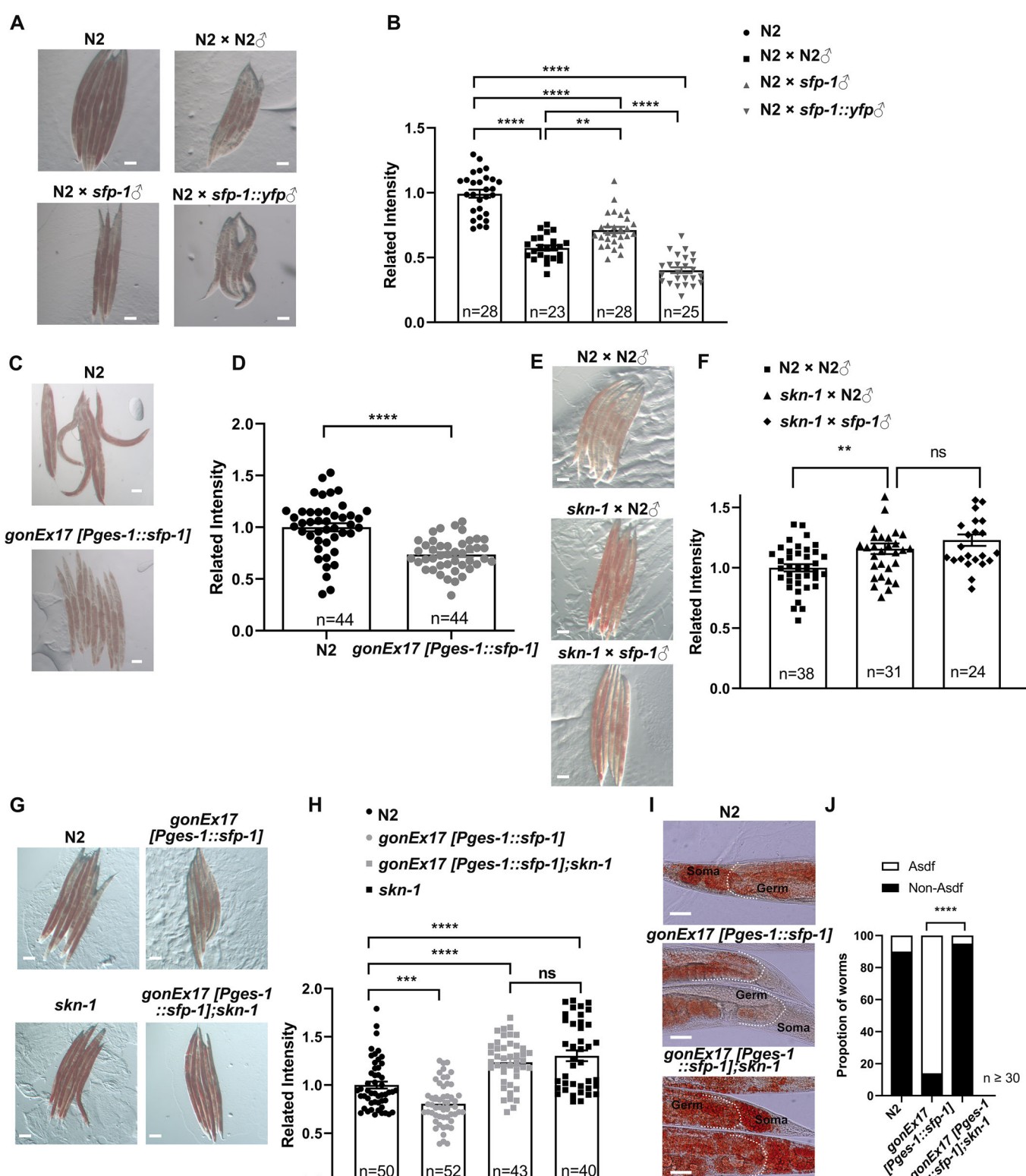

unmated hermaphrodites and those mated with different males (N2, *sfp-1* mutants). We found that hermaphrodites mated with N2 males significantly reduced levels of both monounsaturated and polyunsaturated fatty acids compared to unmated controls. In contrast, hermaphrodites mated with *sfp-1* mutant males maintained fatty acid profiles similar to unmated animals (Fig. 7B). These findings demonstrate that PUFA metabolism may play a more direct role in SFP-1-mediated effects.

In *C. elegans*, seven fatty acid desaturase (FAT) genes are involved in fatty acid metabolism. Among these, *fat-1* and *fat-2* that

**Figure 6.   SFP-1 regulates post-mating lipid metabolism and triggers Asdf through SKN-1.**

(A, B) Representative pictures of Oil Red O staining day 5 hermaphrodites in the presence of males starting at adult day 1. Quantification of Oil Red O staining revealed that neutral lipid levels in N2 hermaphrodites were significantly elevated following mating with *sfp-1* males compared to mating with N2 males. In contrast, neutral lipid levels in N2 hermaphrodites were significantly reduced after mating with *sfp-1::yfp* males. *P* values were calculated using one-way ANOVA followed by Bonferroni's multiple comparisons test. N2 × N2 ♂ vs. N2 unmated: ****$P < 1 \times 10^{-15}$; day 4, N2 × *sfp-1* ♂ vs. N2 unmated: ****$P = 9.81 \times 10^{-12}$; N2 × *sfp-1::yfp* ♂ vs. N2 unmated: ****$P < 1 \times 10^{-15}$ ; N2 × *sfp-1::yfp* ♂ vs. N2 × N2 ♂: ****$P = 7.48 \times 10^{-5}$. N2 × *sfp-1* ♂ vs. N2 × N2 ♂: **$P = 0.0017$. Error bars: SEM. (Scale bars: 100 μm). (C, D) Representative pictures of Oil Red O staining in 5 days adult control and intestinal overexpressing SFP-1 worms. Quantification of Oil Red O staining demonstrated a significant reduction in neutral lipid levels in animals overexpressing SFP-1 in the intestine compared to controls. ****$P = 8.70 \times 10^{-8}$ (two-tailed unpaired *t* test). Error bars: SEM. (Scale bars: 100 μm). (E, F) Representative micrographs of Oil Red O staining in day 5 adult *skn-1* mutants co-cultured with males from adult day 1. Quantification of Oil Red O staining revealed no significant differences in neutral lipid levels between *skn-1* hermaphrodites mated with *sfp-1* males and those mated with N2 males. *P* values were calculated using one-way ANOVA followed by Bonferroni's multiple comparisons test. **$P = 0.0071$. Error bars: SEM. (Scale bars: 100 μm). (G, H) Representative pictures of Oil Red O staining in 5 days old adult control and intestinal overexpressing SFP-1 in *skn-1* mutant nematodes. Quantification of Oil Red O fat staining indicated that intestinal overexpressing SFP-1 animals had significantly increased neutral lipid levels in the absence of *skn-1* compared to the WT background. Statistical significance was determined by Mann–Whitney test. *gonEx17* vs. N2 unmated: ***$P = 0.0001$; *gonEx17; skn-1* vs. N2 unmated: ****$P = 7.23 \times 10^{-6}$; *skn-1* vs. N2 unmated: ****$P = 1.12 \times 10^{-5}$. Error bars: SEM. (Scale bars: 100 μm). (I, J) In the absence of *skn-1*, intestinal overexpressing SFP-1 animals can partially suppress somatic lipid depletion (Asdf). ****$P < 1 \times 10^{-15}$; and *P* values were obtained Chi-square. (Scale bars: 10 μm).

encode ω-3 fatty acid desaturases and Δ12-desaturase, respectively required for PUFA biosynthesis (Watts and Browse, 2002). FAT-6 and FAT-7 are required to synthesize oleic acid and downstream PUFAs (Brock et al, 2006) (Fig. EV5A). Previous studies have reported significant changes in the expression of fatty acid biosynthesis-related genes in mated hermaphrodites (Choi et al, 2021). To investigate their roles in SFP-1 regulating longevity and fat levels, we employed RNAi to knock down these genes in worms with ectopic SFP-1 expression. We found that the knockdown of *fat-1* and *fat-2* significantly prevented the depletion of lipids induced by overexpression SFP in intestinal cells (Fig. 7C,D). Furthermore, knocking down *fat-1, fat-2* also restored the lifespan of the worms with ectopic SFP-1 expression to normal levels (Fig. 7E,F). Additionally, *fat-1* and *fat-2* RNAi were both resistant to mating-induced lifespan reduction, even when hermaphrodites were mated with *sfp::yfp* males (Fig. 7G,H). Similarly, RNAi knockdown of other fatty acid desaturases, including *fat-3, fat-4*, and *fat-5*, extended the lifespan of SFP-1 overexpressing worms compared to empty vector controls (Fig. EV5B–D). In contrast, *fat-6* and fat-7 were not required for lifespan regulation in worms with ectopic SFP-1 expression (Fig. EV5E,F). These findings support the hypothesis that SFP-1 triggers PUFA depletion and the knockdown of the *fat* genes could protect hermaphrodites from death by preventing PUFA depletion and fat loss. Collectively, these results highlight a critical role for PUFAs in mediating the physiological effects of SFP-1 on lipid metabolism and longevity.

## Discussion

Our study demonstrates that SFP-1, a male seminal fluid protein, functions as a key regulator of post-mating physiological changes in *C. elegans* hermaphrodites. SFP-1 is expressed in the seminal vesicle and is identified as a component of the secretory vesicles produced by the seminal vesicle. In mated hermaphrodites, SFP-1 could be transferred from the hermaphrodite uterus to intestinal cells via endocytosis, ultimately causing the hermaphrodites' post-mating physiological changes (Fig. EV5G). This function exhibits evolutionary conservation with the sex peptide (SP) in *Drosophila*, which similarly regulates post-mating responses and longevity modulation in female flies (Chen et al, 1988). Like SP, which is a small protein with 36 amino acids and transferred to females during copulation

(Chen et al, 1988; Liu and Kubli, 2003; Peng et al, 2005). SFP-1 is also a small protein containing signal peptide sequences and is predominantly expressed in the male reproductive tract. However, SP has well-characterized functional domains, the tryptophan-rich N-terminus, the hydroxyproline-rich mid-segment, and the disulfide-bond-containing C-terminus (Domanitskaya et al, 2007; Peng et al, 2005; Rezaval et al, 2012), the functional significance of the NTF2-like domain in SFP-1 remains to be fully studied. While the NTF2-like domain in *Drosophila* NFX1 protein affects the formation of the meiotic spindle in females (Golubkova et al, 2009), its role in SFP-1 function needs further investigation. Notably, we observed that while the signal peptide is essential for SFP-1 secretion in males, it appears dispensable for protein function during ectopic expression in hermaphrodites, suggesting distinct mechanisms of cellular uptake and localization depending on the physiological environment. Future studies should investigate whether the signal peptide is required for the endogenous function of SFP-1 in males.

Our findings establish SFP-1 as a novel, seminal fluid-mediated mechanism that modulates post-mating longevity in hermaphrodites, distinct from male sperm-mediated and pheromones-dependent pathways (Ludewig et al, 2019; Maures et al, 2014; Shi and Murphy, 2014; Shi et al, 2017). Overexpression of SFP-1 in intestinal cells significantly shortened lifespan, while mating with *sfp-1* mutant males partially attenuated the mating-induced lifespan shortening in hermaphrodites, suggesting an important role for SFP-1 in post-mating longevity. SFP-1 mediates these effects through the conserved transcription factors DAF-16 and SKN-1, which regulate longevity, stress responses, and lipid metabolism. (An and Blackwell, 2003; Blackwell et al, 2015; Lehrbach and Ruvkun, 2019; Paek et al, 2012; Pang et al, 2014; Steinbaugh et al, 2015). Moreover, SFP-1 appears to regulate the nucleocytoplasmic shuttling of DAF-16 and SKN-1, specifically, SFP-1 promotes DAF-16 nuclear export, maintaining its cytoplasmic localization and thereby shortening lifespan in mated hermaphrodites. Conversely, SFP-1 overexpression in intestinal cells enhanced SKN-1 activity and promoted its nuclear translocation. The lifespan-shortening effect of SFP-1 appears to be mediated through SKN-1 activation, as *skn-1* knockdown eliminated the extended post-mating lifespan observed in wild-type hermaphrodites mated with *sfp-1* males. This observation is consistent with previous reports demonstrating that SKN-1 activation can shorten lifespan (Lynn et al, 2015; Nhan et al, 2019; Ramos and Curran, 2023). Our findings suggested

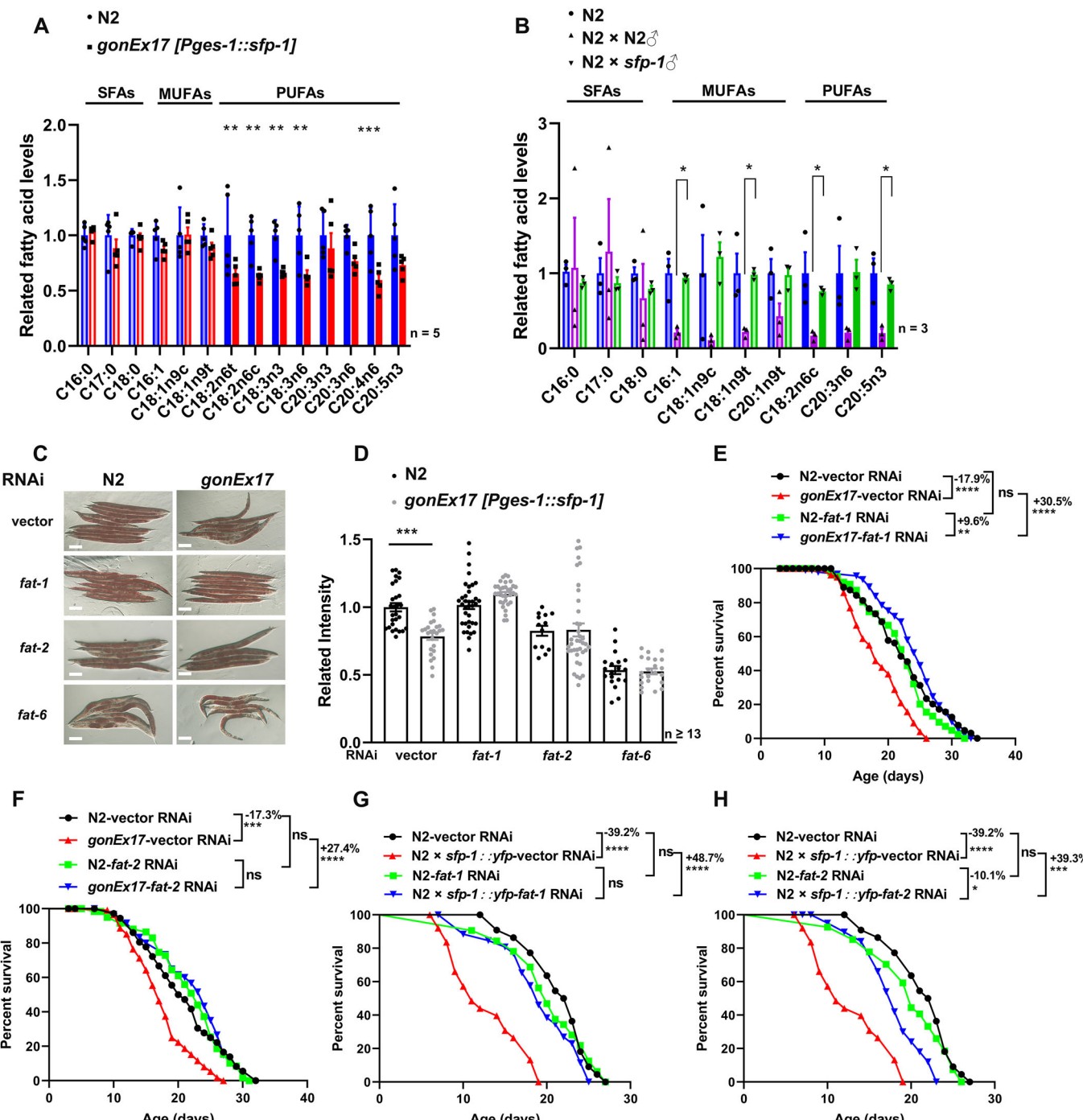

that SFP-1 modulates both DAF-16 localization and SKN-1 activity to regulate longevity. We proposed that the balance between these transcription factors may be maintained by upstream components of the Insulin/IGF-1 signaling (IIS) pathway, such as DAF-18 (Park et al, 2021). This observation raises the possibility that additional IIS components may mediate SFP-1-induced regulation of post-mating lifespan.

Our results suggested that the reduction in fatty acids observed in mated hermaphrodites is likely mediated by the overactivation of SKN-1, triggered by the seminal fluid protein SFP-1. Additionally,

Asdf characterization—the typical lipid metabolism aberrant phenotype of the *skn-1gf* mutation—can be observed in worms overexpressing SFP-1. Mated hermaphrodites also exhibit fat loss, and their significant lipid consumption is suppressed when mating with *sfp-1* mutant males. This suppression may explain the reduced brood size in hermaphrodites mated with *sfp-1* mutant males compared to those mated with wild-type males, less energy and fewer nutrients are being transported from somatic cells to the germline. Therefore, SFP-1 enhances SKN-1 activity to regulate post-mating lipid metabolism.

◄ **Figure 7. SFP-1 triggers PUFAs depletion in intestinal cells.**

(A) Fatty acid profile by GC-MS following intestinal overexpressing SFP-1. Fatty acid levels were normalized to the control condition. Data are the mean ± SEM of five independent experiments. Significant $P$ values are shown. $P$ values: two-way ANOVA with Bonferroni's multiple comparison test. $**P = 0.0048$, $**P = 0.0022$, $**P = 0.0068$, $**P = 0.0033$, $***P = 0.0006$ (from left to right). (B) Fatty acid profile by GC-MS following N2 worms in different conditions. Fatty acid levels were normalized to the control condition. Data are the mean ± SEM of three independent experiments. Significant $P$ values are shown. $P$ values: two-way ANOVA with Bonferroni's multiple comparison test. $*P = 0.0106$, $*P = 0.0186$, $*P = 0.0282$, $*P = 0.0435$ (from left to right). (C, D) Representative images of Oil Red O staining in day 5 adult nematodes (C) and (D) quantification of lipid levels in control and *intestine::sfp-1* transgenic worms under EV (empty vector) and *fat* gene RNAi conditions. *Intestine::sfp-1* transgenic worms exhibited reduced fat storage when grown on EV, a phenotype that was abolished under *fat* gene RNAi. Error bars: SEM; $n \geq 13$; $***P = 0.0004$; one-way ANOVA followed by Bonferroni's multiple comparisons test. (Scale bars: 100 μm). (E) Lifespan survival curves of WT animals and intestinal SFP-1 overexpressing (*intestine::sfp-1,gonEx17*) animals treated with EV or *fat* RNAi. The short-lived lifespan phenotype *of intestine::sfp-1* transgenic worms can be suppressed by *fat-1* gene knockdown. N2-vector RNAi: 22.41 ± 0.75 days ($n = 64$); N2-*fat-1* RNAi: 22.11 ± 0.81 days ($n = 85$); *gonEx17*-vector RNAi: 18.39 ± 0.49 days ($n = 77$); *gonEx17*-*fat-1* RNAi: 24.00 ± 0.54 days ($n = 93$). $****P = 1.13 \times 10^{-6}$, $**P = 0.005$, $****P = 8.64 \times 10^{-15}$ (from left to right). Significance was determined by the Log-rank (Mantel–Cox) test. (F) Lifespan survival curves of WT animals and intestinal overexpressing SFP-1 animals treated with EV or *fat* RNAi. The short-lived lifespan phenotype *of intestine::sfp-1* transgenic worms can be suppressed by *fat-2* gene knockdown. N2-vector RNAi: 20.81 ± 1.01 days ($n = 36$); N2-*fat-2* RNAi: 21.56 ± 0.76 days ($n = 59$); *gonEx17*-vector RNAi: 17.29 ± 0.41 days ($n = 113$); *gonEx17*-*fat-2* RNAi: 22.03 ± 0.76 days ($n = 60$). $***P = 1.07 \times 10^{-4}$, $****P = 1.02 \times 10^{-9}$. Significance was determined by the Log-rank (Mantel–Cox) test. (G) Lifespan survival curves of unmated hermaphrodites and hermaphrodites mated with *sfp-1::yfp♂* when treated with EV or *fat* RNAi. The short-lived lifespan phenotype of mating with *sfp-1::yfp♂* can be suppressed by *fat-1* gene knockdown. N2-vector RNAi: 21.45 ± 0.78 days ($n = 22$); N2-*fat-1* RNAi: 21.50 ± 0.61 days ($n = 32$); N2 × *sfp-1::yfp♂*-vector RNAi: 12.72 ± 0.89 days ($n = 23$); N2 × *sfp-1::yfp♂*-*fat-1* RNAi: 18.97 ± 0.85 days ($n = 26$). $****P = 4.63 \times 10^{-9}$, $****P = 1.86 \times 10^{-7}$ (from left to right). Significance was determined by the Log-rank (Mantel–Cox) test. (H) Lifespan survival curves of unmated hermaphrodites and hermaphrodites mated with *sfp-1::yfp♂* when treated with EV or *fat* RNAi. The short-lived lifespan phenotype of mating with *sfp-1::yfp♂* can be suppressed by *fat-2* gene knockdown. N2-vector RNAi: 21.45 ± 0.78 days ($n = 22$); N2-*fat-2* RNAi: 19.78 ± 0.91 days ($n = 27$); N2 × *sfp-1::yfp♂*-vector RNAi: 12.72 ± 0.89 days ($n = 23$); N2 × *sfp-1::yfp♂*-*fat-2* RNAi: 17.84 ± 0.83 days ($n = 17$). $****P = 4.63 \times 10^{-9}$, $*P = 0.024$, $***P = 0.001$. Significance was determined by the Log-rank (Mantel–Cox) test.

Surprisingly, intestinal overexpression of SFP-1 in wild-type worms induces post-mating phenotypes, including sperm-dependent shrinking and increased brood size, despite the requirement of functional male sperm for these effects in mated hermaphrodites. We propose that overexpressed SFP-1 may utilize metabolites derived from intestinal cells to stimulate the germline that induces the shrinking phenotype. Besides, reproduction activity needs support from somatic cells such as the intestine (Kimble and Sharrock, 1983). However, limited research has examined the direct role of gut cells in reproductive processes. Our observations revealed a significant depletion of PUFAs in somatic tissues of SFP-1 overexpressing worms. Given the critical role of PUFAs in various physiological processes (Chamoli et al, 2020; Delmastro-Greenwood et al, 2014), we hypothesize that SFP-1 may utilize PUFAs to stimulate germline activity, potentially promoting gamete production. This model could explain the observation that germline-deficient mutants still exhibit fat loss after mating (Shi and Murphy, 2014). Together, these findings suggest that SFP-1 acts as a key regulator of fat depletion to facilitate resource allocation after mating.

## Methods

### Reagents and tools table

| Reagent/resource | Reference or source | Identifier or catalog number |
| --- | --- | --- |
| **Experimental models** | | |
| *Caenorhabditis elegans strains* | This study | Dataset EV2 |
| **Recombinant DNA** | | |
| Plasmid: *PBS77::Psfp-1::sfp-1::yfp* | This study | |
| Plasmid: *PBS77::Pges-1::yfp* | This study | |

| Reagent/resource | Reference or source | Identifier or catalog number |
| --- | --- | --- |
| Plasmid: *PBS77::Pmyo-3::sfp-1::yfp* | This study | |
| Plasmid: *PBS77::Pmyo-3::sfp-1::SL2::yfp* | This study | |
| Plasmid: *PBS77::Prgef-1::sfp-1::SL2::yfp* | This study | |
| Plasmid: *PBS77::Pges-1::K12H6.5::SL2::yfp* | This study | |
| Plasmid: *PBS77::Pges-1:F40G9.15::SL2::yfp* | This study | |
| Plasmid: *PBS77::Pges-1(△NTF2L)::SL2::yfp* | This study | |
| Plasmid: *PBS77::Pges-1::sfp-1::SL2::mCherry2* | This study | |
| Plasmid: *PBS77::Pges-1::sfp-1::SL2::yfp* | This study | |
| Plasmid: *pCFJ150::Psmu-1::sfp-1* | This study | |
| Plasmid: *PBS77::Pges-1::sfp-1(△signal peptide)::SL2::yfp* | This study | |
| Plasmid: *PBS77::Pges-1::sfp-1(△98-132)::SL2::yfp* | This study | |
| Plasmid: *PBS77::Pges-1::GFP10*7* | This study | |
| Plasmid: *PBS77::Psfp-1::sfp-1::GFP1-10* | This study | |
| Plasmid: L4440 vector RNAi | This study | |
| Plasmid: L4440 *daf-2* RNAi | This study | |
| Plasmid: L4440 *pqm-1* RNAi | Ahringer RNAi library | |
| Plasmid: L4440 *ceh-60* RNAi | This paper | |
| Plasmid: L4440 *pha-4* RNAi | Ahringer RNAi library | |

| Reagent/resource | Reference or source | Identifier or catalog number |
|---|---|---|
| Plasmid: L4440 *hsf-1* RNAi | Ahringer RNAi library | |
| Plasmid: L4440 *daf-12* RNAi | Ahringer RNAi library | |
| Plasmid: L4440 *pmk-1* RNAi | Ahringer RNAi library | |
| Plasmid: L4440 *sek-1* RNAi | This paper | |
| Plasmid: L4440 *vit-1* RNAi | Ahringer RNAi library | |
| Plasmid: L4440 *vit-2* RNAi | This study | |
| Plasmid: L4440 *vit-3* RNAi | Ahringer RNAi library | |
| Plasmid: L4440 *vit-4* RNAi | Ahringer RNAi library | |
| Plasmid: L4440 *vit-5* RNAi | Ahringer RNAi library | |
| Plasmid: L4440 *vit-6* RNAi | This study | |
| Plasmid: L4440 *fat-1* RNAi | This study | |
| Plasmid: L4440 *fat-2* RNAi | Ahringer RNAi library | |
| Plasmid: L4440 *fat-3* RNAi | Ahringer RNAi library | |
| Plasmid: L4440 *fat-4* RNAi | Ahringer RNAi library | |
| Plasmid: L4440 *fat-5* RNAi | Ahringer RNAi library | |
| Plasmid: L4440 *fat-6* RNAi | Ahringer RNAi library | |
| Plasmid: L4440 *fat-7* RNAi | Ahringer RNAi library | |
| **Oligonucleotides and other sequence-based reagents** | | |
| Primers | This study | Dataset EV2 |
| Gene synthesis for truncation of *sfp-1* | This study | Dataset EV2 |
| **Chemicals, enzymes and other reagents** | | |
| DpnI | New Englands BioLabs | R0176S |
| Phanta Max Super-Fidelity DNA Polymerase | Vazyme | P505-d1 |
| Seamless Cloning Kit | Beyotime | D7010M |
| 2×Hieff PCR Master Mix | Yeasen | 10101 |
| DAPI solution | Solarbio | C0065 |
| MitoTracker Red CMXRos dye | Invitrogen | M7512 |
| isopropyl-b-D-thiogalactoside | Sigma | Cat# 367-93-1 |
| Carbenicillin | Sigma | Cat# 4800-94-6 |
| Oil Red O | Solarbio | O8010 |
| Tissue Triglyceride (TG) Content Assay Kit | Applygen | E1013 |
| Genomic DNA Extraction Kit | Tiangen | DP304 |

| Reagent/resource | Reference or source | Identifier or catalog number |
|---|---|---|
| **Software** | | |
| GraphPad Prism 8 | GraphPad | https://www.graphpad.com/scientific-software/prism/ |
| SPSS Statistics | IBM | Version 21.0.0.0 |

## *C. elegans* strains and maintenance

N2 Bristol was used as the wild-type strain. Worms were maintained at 20 °C on nematode growth medium (NGM) plates with OP50 *E. coli* bacteria as the seeding source, following standard protocol (Brenner, 1974). To induce the mutant phenotype, *glp-1(e2141)* was cultured at 25 °C. Transgenic lines were generated by microinjecting plasmid DNA directly into the gonads of young adult hermaphrodites. For the construction of plasmids, primers used are listed in Dataset EV2. To generate SFP-1 fusion-expressing nematodes, a 5 kb upstream region of the *sfp-1* gene was used as the promoter to drive the expression of SFP-1 fused with YFP at its C-terminus. The coding sequences of the *sfp-1* promoter and *sfp-1::YFP* were cloned into the vector PBS77. To generate tissue-specific SFP-1 fusion-expressing nematodes, the P*ges-1* (intestinal-specific) and P*myo-3* (muscle-specific) promoters were used to drive the expression of SFP-1 fused with YFP at the C-terminus were cloned into the vector PBS77. For tissue-specific expression of SFP-1, the coding sequence of *sfp-1* was cloned between the tissue-specific promoter and SL2::YFP. These constructs were microinjected (50 ng/µL) into the gonads of young adult hermaphrodites to generate nematodes carrying extrachromosomal transgene arrays. All stains and plasmids are listed in Dataset EV2.

## CRISPR–Cas9

*sfp-1(syb1800)*, *sfp-1::mNG (syb9898)* were generated by SunyBiotech upon our request. sgRNA was designed according to the required mutant genes, and cas9 sgRNA expression plasmids were constructed. The constructed plasmids were injected into the gonad using a microinjection. PCR and sequencing were used to determine if the gene was successfully recombined and if it had mutated.

## Lifespan assay

Lifespan assays were performed on 60 mm NGM plates at 20 °C. ~100 worms of each genotype were used for each assay. Worms were transferred to fresh NGM plates every other day, and survival rates were scored every 1–2 days. Worms were censored if they crawled off the plate, bagged, or exhibited protruding vulva. In all cases, the first day of adulthood was scored as day 1. Lifespan data was analyzed with GraphPad Prism 8 (GraphPad Software, Inc.) and IBM SPSS Statistics 21 (IBM, Inc.). Log-rank (Kaplan–Meier) was used to calculate *P* values. All lifespan experiments were performed in at least two independent replicates. Detailed lifespan data and statistical analyses are provided in Dataset EV1.

## RNAi treatment

Freshly streaked single colonies of HT115 bacteria containing either empty vector L4440 or RNAi plasmid were grown overnight (20–24 h) at 37 °C in LB medium supplemented with carbenicillin (100 μg/ml). The next day, bacteria were diluted into fresh LB and cultured for a few hours, and subsequently, IPTG (200 mM final concentration) was added in liquid culture when bacterial density reached $OD_{600} = 0.6$ to induce the production of dsRNA for 4 h at 37 °C. Two days before experiments, freshly grown RNAi bacteria were seeded on RNAi nematode growth media (NGM) plates supplemented with carbenicillin (25 μg/ml) and IPTG (1 mM). All RNAi feeding assays were started from eggs throughout lifespan at 20 °C. Tissue-specific gene knockdown using the strains targeting RNAi to the intestine (VP303), Synchronized eggs of VP303 were cultured on NGM plates seeded with *E. coli* HT115 RNAi clones at 20 °C until the progeny reached the L4 stage.

## Mating assay

3.5-cm NGM plate was seeded with 20 μl OP50. For hermaphrodites, one synchronized day 2 adult hermaphrodite and three young males (day 1- day 2) were transferred into each plate and allowed to mate for 24 h. On day 3, each hermaphrodite was numbered for the experiment and was transferred onto new NGM plates every day.

For unmated male controls, approximately 50 synchronized males were maintained in isolation on individual 6 cm NGM plates (10 males per plate), and these unmated males were imaged at either day 2 or day 3 of adulthood to ensure no mating occurred during the experiment. For mated male assays, one day 2 adult hermaphrodite and three day 1 adult males were placed together on a 3.5 cm NGM plate for 24 h, and after mating, the original plates were kept for an additional 3 days to assess the male/hermaphrodite ratio among progeny. Worms from plates that failed to produce cross-progeny, indicating unsuccessful mating, were excluded from further analysis.

## Brood assay

The nematodes were distributed onto NGM plates with a diameter of 30 mm and one nematode per plate. A minimum of ten plates were utilized for each brood assay experiment. Egg production and hatching rates were quantified daily, and the entire experiment was replicated three times. *P* values were calculated by one-way ANOVA, Bonferroni's multiple comparisons test.

## Body length measurement

The middle line of each worm was delineated using the segmented line tool and the total length was documented as the body length of the worm. *T* test was performed to compare the body size difference between groups of worms on the same day.

## Confocal microscopy

Live worms were mounted on 2% agarose pads with 20 mM levamisole and examined using an Olympus FV3000 scanning confocal microscope. For imaging of SFP-1 release during mating, three *glo-4* hermaphrodites were placed with ten males of the *sfp-1::mNG* strain in 3.5-cm NGM with 20ul OP50. For imaging of

SFP-1 located in the intestine after mating, three *Pges-1::GFP11*7;glo-4* hermaphrodites were placed with ten males of the *sfp-1::GFP1-10* strain in 3.5-cm NGM with 20 μl OP50. The mating process lasts for 30 min. The mated hermaphrodites were imaged 0.1 h and 0.5 h after mating. Imaging was performed using ×60 objective.

## Imaging analysis

To quantify the fluorescence intensity of *sfp-1::yfp* and *sfp-1::mNG*, 20-30 worms from each experimental group were anesthetized using 20 mM levamisole on 2% agarose pads. Images were acquired using an FV3000 confocal microscope (Olympus) with a ×10 objective. Fluorescence intensity analysis was performed using ImageJ software (NIH). For each worm, the seminal vesicle area was delineated by selecting a rectangular region of interest (ROI) in ImageJ. The mean YFP intensity within the ROI was measured for all males, ensuring consistent ROI dimensions across samples to maintain comparability. Statistical comparisons of YFP intensity between different groups were conducted using an unpaired two-tailed *t* test.

To quantify *gst-4::gfp* fluorescence intensity, at least 30 worms day 5 adult worms were anesthetized with 20 mM levamisole on 2% agarose pads. Images were acquired on FV3000 (Olympus) under ×10 objective and analyzed with ImageJ (NIH). Mann–Whitney test was performed to compare the GFP intensity of different groups of worms.

To quantify the fluorescence intensity of *sod-3::gfp*, at least 100 worms were cultured under different conditions until day 1 of adulthood, then mated with different male genotypes for 24 h. On D3, GFP fluorescence images were captured using an FV3000 confocal microscope (Olympus) with a ×10 objective. Fluorescence intensity analysis was performed using ImageJ software (NIH). For each worm, the intestinal region was delineated by selecting a rectangular region of interest (ROI) of consistent dimensions to ensure comparability across samples. The mean GFP intensity within each ROI was measured after subtracting background fluorescence. Statistical comparisons between different groups were performed using one-way ANOVA followed by Bonferroni's post-hoc test.

## Scoring secretory vesicles and fluorescence microscopy

The representative pictures presented in the manuscript were acquired with an inverted FV3000 confocal microscope with a 60× oil immersion objective. For each secretory vesicle scoring assay, the circular puncta at the location of the seminal vesicle in the pictures taken of the males was counted. *P* values were calculated by one-way ANOVA, Bonferroni's multiple comparisons test.

## Oil Red O staining and quantification

Mid-L4 stage animals were placed on OP50 NGM plates and raised to day 5 adult stage. Worms were washed off the plates using PBS containing 0.1% Triton X-100 and incubated in 60% isopropyl alcohol for 3 min with gentle rocking. After pelleting, the worms were stained overnight with Oil Red O (ORO) dissolved in distilled water. Following staining, the worms were washed three times with PBS containing 0.1% Triton X-100 to remove excess dye. Stained worms were imaged at 5× magnification using a ZEISS steREO

Discovery V8 color camera and Mshot MS23 software. For lipid content quantification, color images were background-subtracted and converted to grayscale using ImageJ. Signal intensities were measured and compared across worms grown under different experimental conditions. A minimum of 20 worms were analyzed for each experiment. Statistical analysis was performed using an unpaired $t$ test to compare fat staining between two groups. For comparisons involving more than two groups, $P$ values were calculated using one-way ANOVA followed by Bonferroni's multiple comparisons test. Non-parametric tests as the data are non-normally distributed.

## DAPI staining and counting of intestinal nucleus

Worms were stained according to the Bio-protocol (https://bio-protocol.org/exchange/protocoldetail?id=77&type=1). At least ten worms per day 1 adult worms were anesthetized with 20 mM levamisole on 2% agarose pads. Images were taken with an FV3000. Images were acquired on FV3000 (Olympus) under ×60 objective. Intestinal nuclei were identified by DAPI staining, and nuclear localization of the fluorescent signal was quantified by counting the number of intestinal nuclei surrounded by YFP within the merged images. Statistical analysis was performed using the Mann–Whitney test to compare the percentage of nuclear localization between groups.

## MitoTracker staining

MitoTracker Red CMXRos (Invitrogen) dye, which selectively stains mitochondria, was used for labeling *sfp-1::yfp* male sperm, MitoTracker does not affect sperm function or motility. 100–150 males were transferred to a 1.5 ml tube containing 10 µM MitoTracker in 300 µl M9 buffer. Males were incubated in the dark for 4 h, transferred to an NGM plate mated with *glo-4*.

## SKN-1 nuclear localization assay

SKN-1 nuclear localization was analyzed using the transgenic LG326 nematode that expresses a fusion protein of SKN-1 tagged with a GFP reporter. Fluorescent images were captured by using FV3000 (Olympus) at ×20 magnification. For quantification, the LG326 nematodes were categorized into three groups (i.e., "low," "medium," and "high") depending on the levels of SKN-1::GFP nuclear accumulation in the intestinal nuclei. "Low" represents that SKN-1::GFP nuclear accumulation is barely detectable through the intestine, "medium" refers to the nematodes in which SKN-1::GFP nuclear accumulation is partially present in intestine, and "high" indicates that a strong SKN-1::GFP signal is observed in almost all intestinal nuclei (An and Blackwell, 2003). Day 1 adult worms ($n > 50$) per strain in each experiment were randomly selected to score sub-cellular localization of GFP-fused proteins, and the percentage of nematodes in each category was calculated. All experiments were repeated independently at least two times with similar results. Chi-square test was used to determine the significance.

## FOXO/DAF-16 localization assay

Hermaphrodites were synchronized using a timed egg lay on *daf-2* RNAi plates. Starting at adult day 1, synchronized hermaphrodites were selected to be either cultured only with other hermaphrodites

(around 35 per 6-cm plate) or with males (around 15 hermaphrodites and around 45 adult day 1 *sfp-1* males or N2 males). After 4 days (adult day 5), the worms were immediately anesthetized in M9 containing 20 mM levamisole, mounted on 2% agar pads and imaged. Fluorescent images were captured by using FV3000 (Olympus) with a ×10 objective. Individual worms scored as primarily cytoplasmic, primarily nuclear or nuclear and cytoplasmic DAF-16::GFP localization in the anterior intestinal cells. Chi-square test was used to determine the significance.

Due to the co-localization of the *DAF-16::GFP* transgene and the *skn-1* locus on the same chromosome, conventional genetic methods were unable to generate a *skn-1* mutant in the *DAF-16::GFP* background. To overcome this limitation, we utilized a sequential RNAi strategy. Specifically, *DAF-16::GFP* worms were initially cultured on *daf-2*(RNAi) plates to induce nuclear localization of DAF-16. Synchronized populations were then divided at the L4 stage. One group continued to be maintained on *daf-2*(RNAi), while the other was transferred to *skn-1*(RNAi) plates from L4 until day 2 of adulthood. Both groups were subsequently mated with either N2 or *sfp-1* males on day 3 and subjected to imaging on day 4.

## Asdf quantification

Worms stained with ORO were photographed using Olympus FV3000 and a ×20 objective. Worms were scored as previously described (Inoue et al, 2005). Fat levels of worms were placed into three categories: non-Asdf, intermediate, and Asdf. Non-Asdf worms display no loss of fat and are stained dark red throughout most of the body (somatic and germ cells). Intermediate worms display significant fat loss from the somatic tissues, with portions of the intestine being clear, but ORO-stained fat deposits are still visible (somatic < germ cells). Asdf worms had most, if not all, observable somatic fat deposits depleted (germ cells only) or significant fat loss from the somatic tissues with portions of the intestine being clear (somatic < germ) (Lynn et al, 2015). Chi-square test was used to determine the significance.

## Protein prediction

Alphafold2 was used to predict the structure of SFP-1, SFP-1 structure prediction file O01420. Model confidence around the caspase cleavage site is very high (pLDDT >90).

## TAG measurement

Day 5 adult hermaphrodites ($n = 100$) were collected, washed with M9, and homogenized in 150 µL lysis Buffer. TAG was measured by using a colorimetric assay kit (E1013-50, Applygen). TAG concentrations were normalized relative to protein concentration using the BCA protein assay. $P$ values were obtained by $T$ test.

## GC–MS analysis of fatty acid profiles

300 age-synchronized old adult (adult day 5) animals were collected in M9 buffer and washed three times to remove residual bacteria in the worm pellets. A small steel ball was added into the sample tube, then homogenating for 30 min. After that, the sample was mixed with 150 µL MeOH, 200 µL MTBE and 50 µL 36% phosphoric acid/water (precooled at −20 °C). The suspension was vortexed for

3 min under the condition of 2500 r/min and centrifuged at 12,000 r/min for 5 min at 4 °C. Take 200 μL of supernatant into a new centrifuge tube blow dry and add 300 μL methanol solution of 15% boron trifluoride, vortex for 3 min under the condition of 2500 r/min, kept in the oven at 60 °C for 30 min. Cool to room temperature, then accurately add 500 μL n-hexane and 200 μL saturated sodium chloride solution. After vortexing for 3 min and centrifugation at 4 °C and 12,000 r/min for 5 min, transfer 100 μL n-hexane layer solution for further GC-MS analysis. The fatty acid concentration of the interventions was normalized to the fatty acid concentration of the control. The final ratio is expressed as relative fatty acid levels in the graph. Each experiment was carried out at least three times independently. Relative fatty acid abundances were plotted using Prism 8 and statistically significant differences between samples were assessed using a two-way analysis of variance with Bonferroni's multiple comparison test. When the $P$ value was less than 0.05, there was a significant difference, which was recorded as *. $P < 0.01$ was marked as **, $P < 0.001$ was marked as ***, and $P < 0.0001$ was marked as ****.

## Data availability

The datasets generated and analyzed during the current study are available from the corresponding author upon reasonable request. The source data of this paper are collected in the following database record: biostudies: https://www.ebi.ac.uk/biostudies/bioimages/studies/S-BIAD1657.

The source data of this paper are collected in the following database record: biostudies:S-SCDT-10_1038-S44318-025-00610-1.

## Peer review information

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

## Acknowledgements

We thank Y Wang, W Yang, and L Zhu for technical assistance. We thank X Zhang for her excellent assistance in preparing the synopsis, model figure, and male gonad schematic. We thank Wei Zou, Lijun Kang from Zhejiang University, Xiajing Tong from Shanghai Science and Technology University, and Shawn Xu from University of Michigan for the assistance and valuable suggestions provided in this study. Some strains were obtained from the Caenorhabditis Genetics Center. This work was funded by the program for HUST Academic Frontier Youth Team (5001170068), the Outstanding Young Scientist Project of Hubei Province (2023AFA060), and the National Natural Science Foundation of China (32100604).

## Author contributions

**Mingqing Chen**: Resources; Data curation; Formal analysis; Validation; Visualization; Writing—review and editing. **Jianke Gong**: Conceptualization; Resources; Data curation; Formal analysis; Supervision; Funding acquisition; Validation; Investigation; Visualization; Methodology; Writing—original draft; Project administration; Writing—review and editing.

Source data underlying figure panels in this paper may have individual authorship assigned. Where available, figure panel/source data authorship is listed in the following database record: biostudies:S-SCDT-10_1038-S44318-025-00610-1.

## Disclosure and competing interests statement

The authors declare no competing interests.

# Expanded View Figures

**Figure EV1.   The secreted protein SFP-1 is involved in mating-induced phenotypes, while *sfp-1* mutant male sperm and pheromones still influence hermaphrodites.**  ▶

(A) Left panel: Diagram outlining the pathway of sperm maturation in males, beginning with mitosis and progressing through meiosis I to produce mature spermatids (pink) that are stored in the seminal vesicle (blue) prior to mating. Right panel: Fluorescence micrograph of male worms stained with MitoTracker Red (red) to visualize spermatids. SFP-1 (green) surrounds the labeled spermatids, confirming SFP-1 protein localization specifically within the seminal vesicle, as illustrated in the schematic. (Scale bars: 10 μm). (B) Representative fluorescence micrograph showing mating between an SFP-1::YFP-expressing male (green) and N2 hermaphrodite. The white arrow indicates the transfer of SFP-1::YFP protein (green) along with sperm (red) during copulation. The exclusive presence of sperm (red) in the uterus of N2 hermaphrodites mated with non-fluorescent control males. (Scale bars: 10 μm). (C) High-magnification images of the spermatheca region in hermaphrodites expressing *sfp-1::yfp*, confirming the male-specific expression of SFP-1. (Scale bars: 10 μm). (D) Lifespan of mated N2 worms. N2 × N2 ♂ : 10.22 ± 0.78 days, $n = 16$ worms; N2 × *sfp-1::yfp* ♂ : 8.22 ± 0.54 days, $n = 20$ worms; N2 × *sfp-1*♂ : 12.93 ± 0.93 days, $n = 22$ worms. $^*P = 0.025$, $^{**}P = 0.003$, indicate significance vs. the N2 × N2 ♂ group (Log-rank test). (E) Total brood size of N2 worms in different conditions. Fertility was assayed by measuring the total offspring production of individual hermaphrodites at 20 °C. $P$ values were calculated by one-way ANOVA with Bonferroni's multiple comparisons test. $^{**}P = 0.0039$, $^{***}P = 0.0002$, $^{****}P = 4.943 \times 10^{-11}$ for comparisons against the unmated N2 control; $^{****}P = 5.02 \times 10^{-7}$ for N2 × N2 ♂ vs. N2 × *sfp-1*♂. Data are presented as mean ± SEM, based on at least 15 worms for each condition. (F) Body length of N2 worms was measured under different mating and genetic conditions over seven days, with statistical analysis performed separately for each day using ordinary one-way ANOVA followed by Bonferroni's multiple comparisons test against the N2 unmated control group. On day 3, *gonEx17*: $^{***}P = 1.66 \times 10^{-4}$; N2 × N2 ♂: $^{****}P = 2.02 \times 10^{-6}$; N2 × *sfp-1*♂: $^{****}P = 8.35 \times 10^{-8}$. On day 4, *gonEx17*: $^*P = 0.014$; N2 × N2 ♂: $^{***}P = 4.11 \times 10^{-4}$; N2 × *sfp-1*♂: $^{****}P = 1.24 \times 10^{-6}$. On day 5, *gonEx17*: $^{****}P = 1.42 \times 10^{-7}$; N2 × N2 ♂: $^{****}P = 2.24 \times 10^{-7}$; N2 × *sfp-1*♂: $^{****}P = 1.44 \times 10^{-10}$. On day 6, *gonEx17*: $^{****}P = 2.06 \times 10^{-9}$; N2 × N2 ♂: $^{****}P = 3.87 \times 10^{-9}$; N2 × *sfp-1*♂: $^{****}P = 4.09 \times 10^{-12}$. On day 7, *gonEx17*: $^{****}P = 1.09 \times 10^{-12}$; N2 × N2 ♂: $^{****}P = 8.00 \times 10^{-15}$; N2 × *sfp-1*♂: $^{****}P = 2.50 \times 10^{-14}$. Data are presented as mean ± SEM, based on at least 15 worms for each condition. (G) Lifespan of N2 under different male conditions. N2 in no male condition: 16.96 ± 0.76 days, $n = 36$ worms; N2 in N2 male condition: 11.72 ± 0.28 days, $n = 97$ worms; N2 in *sfp-1* male condition: 10.89 ± 0.31 days, $n = 79$ worms, $P = 0.066$; $^{****}P = 9.52 \times 10^{-12}$, comparisons were made using the Log-rank (Mantel–Cox) test.

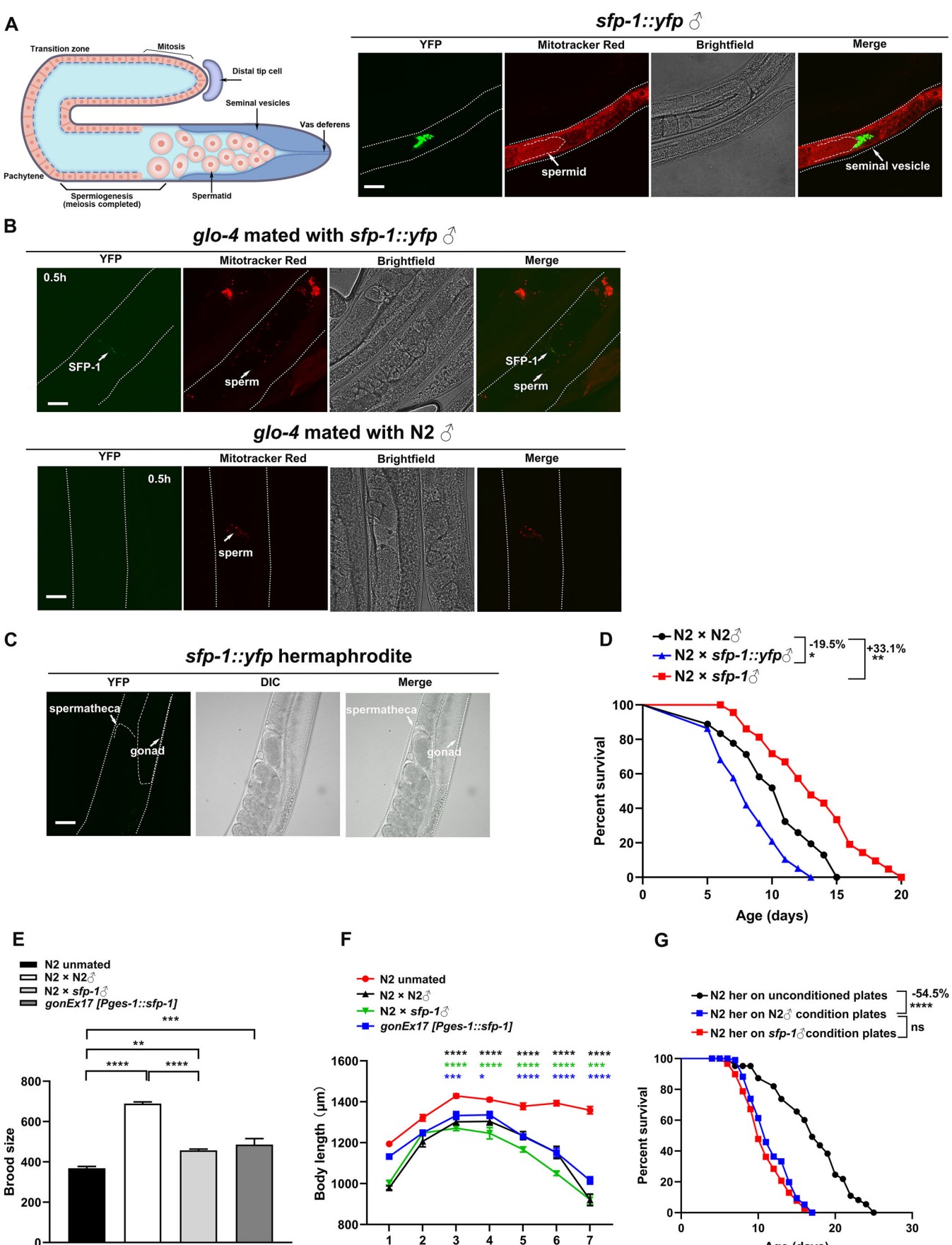

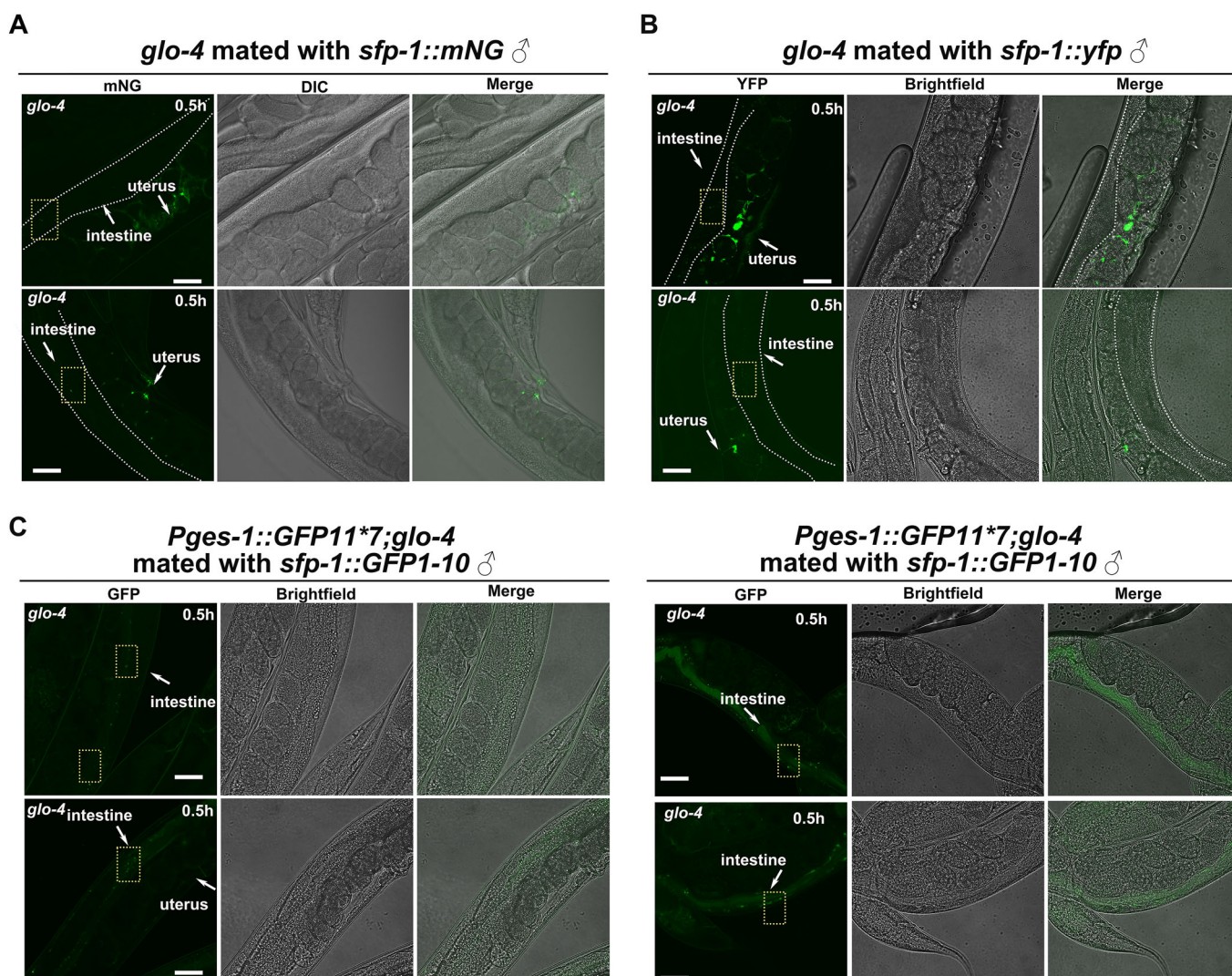

**Figure EV2. SFP-1 is transferred from the uterus into the intestinal cells in mated hermaphrodites.**

(A) Male-to-hermaphrodite transfer of SFP-1::YFP. *sfp-1::yfp* males were mated with *glo-4* hermaphrodites (low gut autofluorescence). Imaging of mated *glo-4* hermaphrodites revealed the presence of male-derived SFP-1::YFP in the uterus immediately after copulation. Within 30 min post-mating, weak fluorescence signals were detected in intestinal cells. The yellow rectangle highlights the fluorescent signal observed in the intestine. (Scale bars: 10 μm). (B) Male-to-hermaphrodite transfer of SFP-1::YFP. *sfp-1::yfp* males were mated with *glo-4* hermaphrodites (low gut autofluorescence). Imaging of mated *glo-4* hermaphrodites revealed the presence of male-derived SFP-1::YFP in the uterus immediately after copulation. Within 30 min post-mating, weak fluorescence signals were detected in intestinal cells. The yellow rectangle highlights the fluorescent signal observed in the intestine. (Scale bars: 10 μm). (C) Male-to-hermaphrodite transfer of SFP-1::GFP1-10. *sfp-1::GFP1-10* males were mated with *Pges-1::GFP11*7;glo-4* hermaphrodites. Half an hour after mating, bright fluorescence appeared in the intestine cell, confirming the transfer and endocytosis of SFP-1. Autofluorescence from male sperm was observed. The yellow rectangle highlights the fluorescent signal observed in the intestine. (Scale bars: 10 μm).

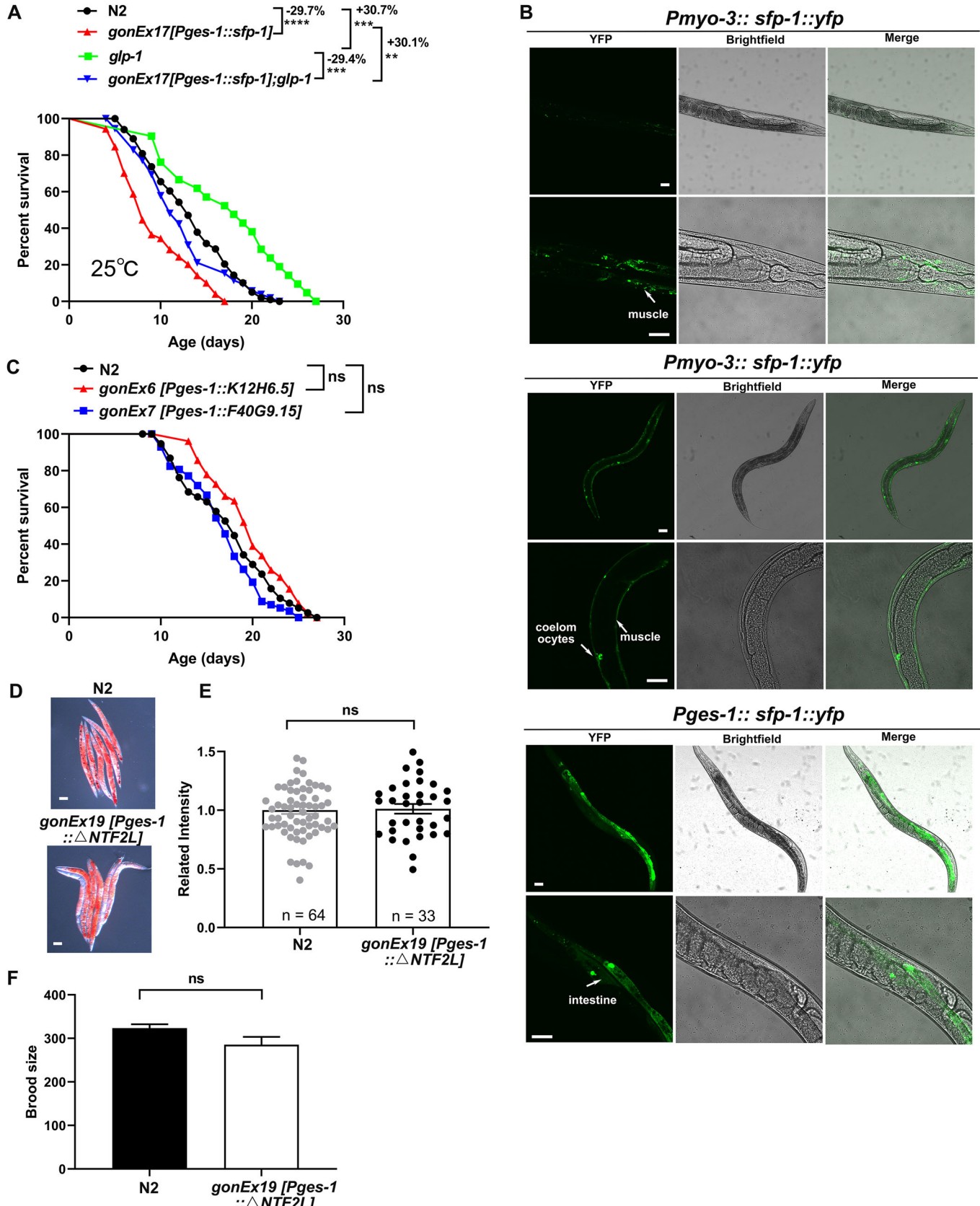

◄ **Figure EV3. The potential function of NTF2L in longevity, lipid metabolism, and reproduction.**

(A) Lifespan survival curves of *glp-1(e2141)* worms, *gonEx17* worms, wild-type N2 worms and *gonEx17;glp-1(e2141)* worms when grown and aged at 25 °C. N2: 13.17 ± 0.46 days, $n = 98$ worms; *gonEx17 [Pges-1::sfp-1]*: 9.25 ± 0.52, $n = 53$ worms; *glp-1(e2141)*: 17.33 ± 1.30, $n = 21$ worms; *gonEx17 [Pges-1::sfp-1];glp-1(e2141)*: 12.10 ± 0.63, $n = 52$ worms. Comparisons were made using the Log-rank (Mantel–Cox) test, $^{****}P = 5.44 \times 10^{-8}$, $^{***}P = 0.0003$; $^{***}P = 0.00015$; $^{**}P = 0.01$ (from left to right). (B) Expression patterns of *Pmyo-3::sfp-1::yfp* and *Pges-1::sfp-1::yfp*. In *Pmyo-3::sfp-1::yfp* animals, low-magnification images show whole worms, while high-magnification images highlight coelomocytes and muscles. Bright fluorescent signals were observed in coelomocytes. In *Pges-1::sfp-1::yfp* animals, low-magnification images show whole worms, and high-magnification images focus on the intestine. Scale bars: 50 μm (low magnification), 10 μm (high magnification). (C) Lifespan survival curves of WT animals and intestinal overexpressing K12H6.5 or F40G9.15 animals. The other seminal fluid proteins such as K12H6.5 and F40G9.15 in the intestine had almost no effect on longevity. N2: 17.40 ± 0.79 days, $n = 48$ worms; *gonEx6 [Pges-1::K12H6.5]*: 19.61 ± 0.46 days, $n = 77$ worms; *gonEx7 [Pges-1::F40G9.15]*: 16.74 ± 0.53 days, $n = 57$ worms, comparisons were made using the Log-rank (Mantel–Cox) test. (D, E) Representative pictures of Oil Red O staining in day 5 adult control and *gonEx19 [Pges-1::△NTF2]* worms. Quantification of Oil Red O fat staining. Compared to the control, the neutral lipid level difference was abolished in the absence of NTFL. Data are presented as mean ± SEM, based on at least 32 worms for each condition. *P* values were calculated by two-tailed unpaired *t* test. (F) Total brood size of N2 worms and *gonEx19 [Pges-1::△NTF2]* worms. In contrast to the fertility-promoting effects observed in mated N2 and *intestine::sfp-1* transgenic worms, the lack of NTF2L significantly inhibited fertility. Data are presented as mean ± SEM, based on at least 8 worms for each condition. *P* values were calculated by two-tailed unpaired *t* test.

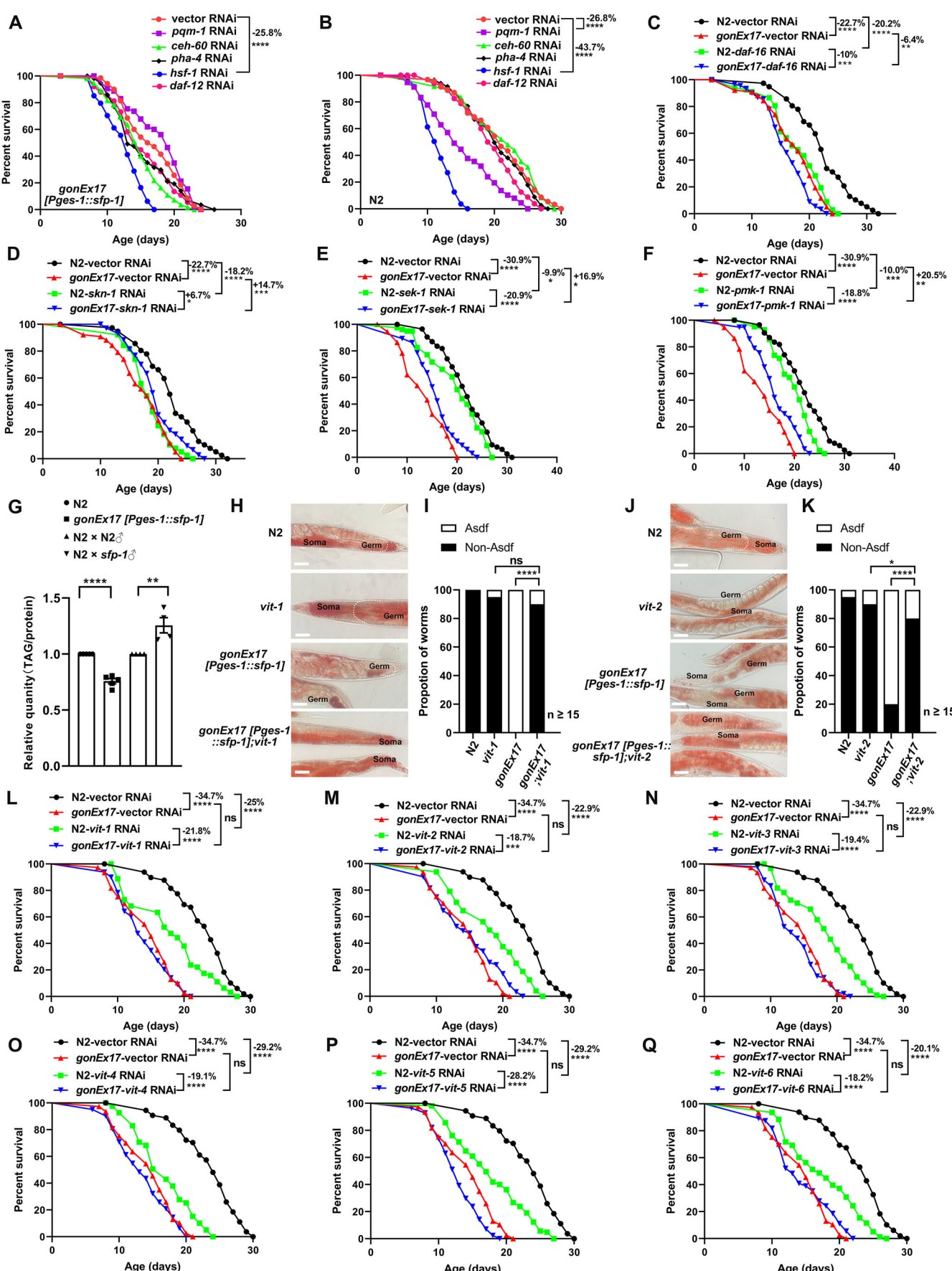

◀ **Figure EV4. Knockdown of the SKN-1 signal pathway major involving genes increased the lifespan of intestinal overexpressing SFP-1 transgenic line.**

(A) Lifespan survival curves of intestinal SFP-1-overexpressing animals treated with RNAi targeting classical longevity-associated transcription factors. None of the tested transcription factors extended the lifespan of intestinal SFP-1-overexpressing animals. *gonEx17-vector* RNAi: 16.76 ± 0.51 days ($n = 70$); *gonEx17-pqm-1* RNAi: 17.77 ± 0.80 days ($n = 52$); *gonEx17-ceh-60* RNAi: 14.58 ± 0.52 days ($n = 55$); *gonEx17-pha-4* RNAi: 15.33 ± 0.62 days ($n = 57$); *gonEx17-hsf-1* RNAi:12.43 ± 0.40 days ($n = 51$); *gonEx17-daf-12* RNAi: 15.26 ± 0.52 days ($n = 74$). $P = 2.19 \times 10^{-9}$. Significance was determined by the Log-rank (Mantel–Cox) test. (B) Lifespan survival curves of WT (N2) animals treated with classical transcription factor genes RNAi. N2-vector RNAi: 20.71 ± 0.55 days ($n = 92$); N2-*pqm-1* RNAi:15.16 ± 0.56 days ($n = 88$); N2-*ceh-60* RNAi: 21.19 ± 0.79 days ($n = 48$); N2-*pha-4* RNAi: 20.36 ± 0.53 days ($n = 80$); N2-*hsf-1* RNAi: 11.66 ± 0.31 days ($n = 56$); N2-*daf-12* RNAi: 19.44 ± 0.49 days ($n = 80$).

$P = 3.64 \times 10^{-11}$ (N2-vector RNAi vs. N2-*pqm-1* RNAi). $P = 6.20 \times 10^{-12}$ (N2-vector RNAi vs. N2-*hsf-1* RNAi). Significance was determined by the Log-rank (Mantel–Cox) test. (C) Knockdown of transcription factor gene *daf-16* by RNAi reduces the lifespan of SFP-1-overexpressing worms. N2-vector RNAi: 22.21 ± 0.55 days ($n = 77$); N2-*daf-16* RNAi:17.76 ± 0.58 days ($n = 67$); *gonEx17-vector* RNAi: 17.06 ± 0.53 days ($n = 76$); *gonEx17-daf-16* RNAi: 15.95 ± 0.40 days ($n = 90$). $P = 1.10 \times 10^{-10}$ (N2-vector RNAi vs. *gonEx17-vector* RNAi), $P = 1.58 \times 10^{-4}$ (N2-*daf-16* RNAi vs. *gonEx17-daf-16* RNAi), $P = 4.69 \times 10^{-8}$ (N2-*daf-16* RNAi vs. N2-vector RNAi), $P = 0.003$. Significance was determined by the Log-rank (Mantel–Cox) test. (D) Knockdown of transcription factor gene *skn-1* by RNAi extends the lifespan of SFP-1-overexpressing worms. N2-vector RNAi: 22.21 ± 0.55 days ($n = 77$); N2-*skn-1* RNAi:18.39 ± 0.37 days ($n = 77$); *gonEx17-vector* RNAi: 17.06 ± 0.53 days ($n = 76$); *gonEx17-skn-1* RNAi: 19.50 ± 0.40 days ($n = 104$). $P = 1.10 \times 10^{-10}$ (N2-vector RNAi vs. *gonEx17-vector* RNAi), $P = 1.80 \times 10^{-9}$ (N2-*skn-1* RNAi vs. N2-vector RNAi), $P = 0.001$, $P = 0.012$. Significance was determined by the Log-rank (Mantel–Cox) test. (E, F) Knockdown of SKN-1 upstream effector *sek-1* (E) and *pmk-1* (F) by RNAi extends the lifespan of SFP-1 expressed ectopically worms. N2-vector RNAi: 22.00 ± 0.43 days ($n = 116$); N2-*sek-1* RNAi:19.81 ± 0.64 days ($n = 71$); *gonEx17-vector* RNAi: 13.43 ± 0.71 days ($n = 37$); *gonEx17-sek-1* RNAi: 15.71 ± 0.48 days ($n = 66$); N2-*pmk-1* RNAi: 19.79 ± 0.58 days ($n = 42$); *gonEx17-pmk-1* RNAi: 16.18 ± 0.52 days ($n = 57$). For (E), $P = 4.97 \times 10^{-22}$ (N2-vector RNAi vs. *gonEx17-vector* RNAi), $P = 1.17 \times 10^{-8}$ (N2-*sek-1* RNAi vs. *gonEx17-sek-1* RNAi), $P = 0.021$, $P = 0.026$. For (F), $P = 4.97 \times 10^{-22}$ (N2-vector RNAi vs. *gonEx17-vector* RNAi), $P = 1.07 \times 10^{-5}$ (N2-*pmk-1* RNAi vs. *gonEx17-pmk-1* RNAi), $P = 0.001$, $P = 0.003$. Significance was determined by the Log-rank (Mantel–Cox) test. (G) Comparative analysis of triglyceride (TAG) levels in N2 worms under different mating conditions. Data are presented as mean ± SEM from ≥4 independent biological replicates. Statistical significance was determined by two-tailed unpaired Student's *t* test: $P = 5.51 \times 10^{-6}$ (N2 vs. gonEx17); $P = 0.0095$ (N2 × N2 vs. N2 × *sfp-1*). Error bars represent SEM. Note: Each mated condition was normalized to its respective control group (N2 or N2 × N2) due to the substantial post-mating TAG reduction. (H, I) The absence of *vit-1* suppressed somatic lipid depletion (Asdf). Lipid distribution was assessed in at least 15 animals. $P < 1 \times 10^{-15}$ and $P$ values were obtained Chi-square. (Scale bars: 10 μm). (J, K) The absence of *vit-2* suppressed somatic lipid depletion (Asdf). Lipid distribution was assessed in at least 15 animals. $P < 1 \times 10^{-15}$, $P = 0.047$, and $P$ values were obtained by Chi-square. (Scale bars: 10 μm). (L–Q) Lifespan survival curves of WT animals and intestinal overexpressing SFP-1 animals treated with EV or *vit* RNAi. *vit* RNAi had almost no effect on the lifespan of SFP-1 expressed ectopically worms. N2-vector RNAi: 22.79 ± 0.61 days ($n = 49$); *gonEx17-vector* RNAi: 15.33 ± 0.42 days ($n = 67$); N2-*vit-1* RNAi: 17.81 ± 0.71 days ($n = 63$); *gonEx17-vit-1* RNAi: 13.84 ± 0.42 days ($n = 78$); N2-*vit-2* RNAi: 18.15 ± 0.73days ($n = 48$); *gonEx17-vit-2* RNAi: 14.72 ± 0.62 days ($n = 59$); N2-*vit-3* RNAi: 18.06 ± 0.53 days ($n = 88$); *gonEx17-vit-3* RNAi: 13.89 ± 0.38 days ($n = 91$); N2-*vit-4* RNAi: 16.71 ± 0.66 days ($n = 40$); *gonEx17-vit-4* RNAi: 13.40 ± 0.45 days ($n = 62$); N2-*vit-5* RNAi: 17.70 ± 0.51 days ($n = 105$); *gonEx17-vit-5* RNAi: 12.73 ± 0.35 days ($n = 84$); N2-*vit-6* RNAi: 17.50 ± 0.59 days ($n = 78$); *gonEx17-vit-6* RNAi: 14.31 ± 0.59 days ($n = 54$). $P = 5.49 \times 10^{-19}$ (N2-vector RNAi vs. *gonEx17-vector* RNAi). For (L), $P = 3.50 \times 10^{-8}$ (N2-*vit-1* RNAi vs. *gonEx17-vit-1* RNAi), $P = 4.90 \times 10^{-5}$ (N2-*vit-1* RNAi vs. N2-vector RNAi). For (M), $P = 1.19 \times 10^{-4}$ (N2-*vit-2* RNAi vs. *gonEx17-vit-2* RNAi), $P = 2.89 \times 10^{-6}$ (N2-*vit-2* RNAi vs. N2-vector RNAi). For (N), $P = 2.53 \times 10^{-11}$ (N2-*vit-3* RNAi vs. *gonEx17-vit-3* RNAi), $P = 8.26 \times 10^{-8}$ (N2-*vit-3* RNAi vs. N2-vector RNAi). For (O), $P = 8.00 \times 10^{-5}$ (N2-*vit-4* RNAi vs. *gonEx17-vit-4* RNAi), $P = 3.13 \times 10^{-10}$ (N2-*vit-4* RNAi vs. N2-vector RNAi). For (P), $P = 1.95 \times 10^{-14}$ (N2-*vit-5* RNAi vs. *gonEx17-vit-5* RNAi), $P = 3.50 \times 10^{-7}$ (N2-*vit-5* RNAi vs. N2-vector RNAi). For (Q), $P = 2.30 \times 10^{-5}$ (N2-*vit-6* RNAi vs. *gonEx17-vit-6* RNAi), $P = 1.42 \times 10^{-7}$ (N2-*vit-6* RNAi vs. N2-vector RNAi). Significance was determined by the Log-rank (Mantel–Cox) test.

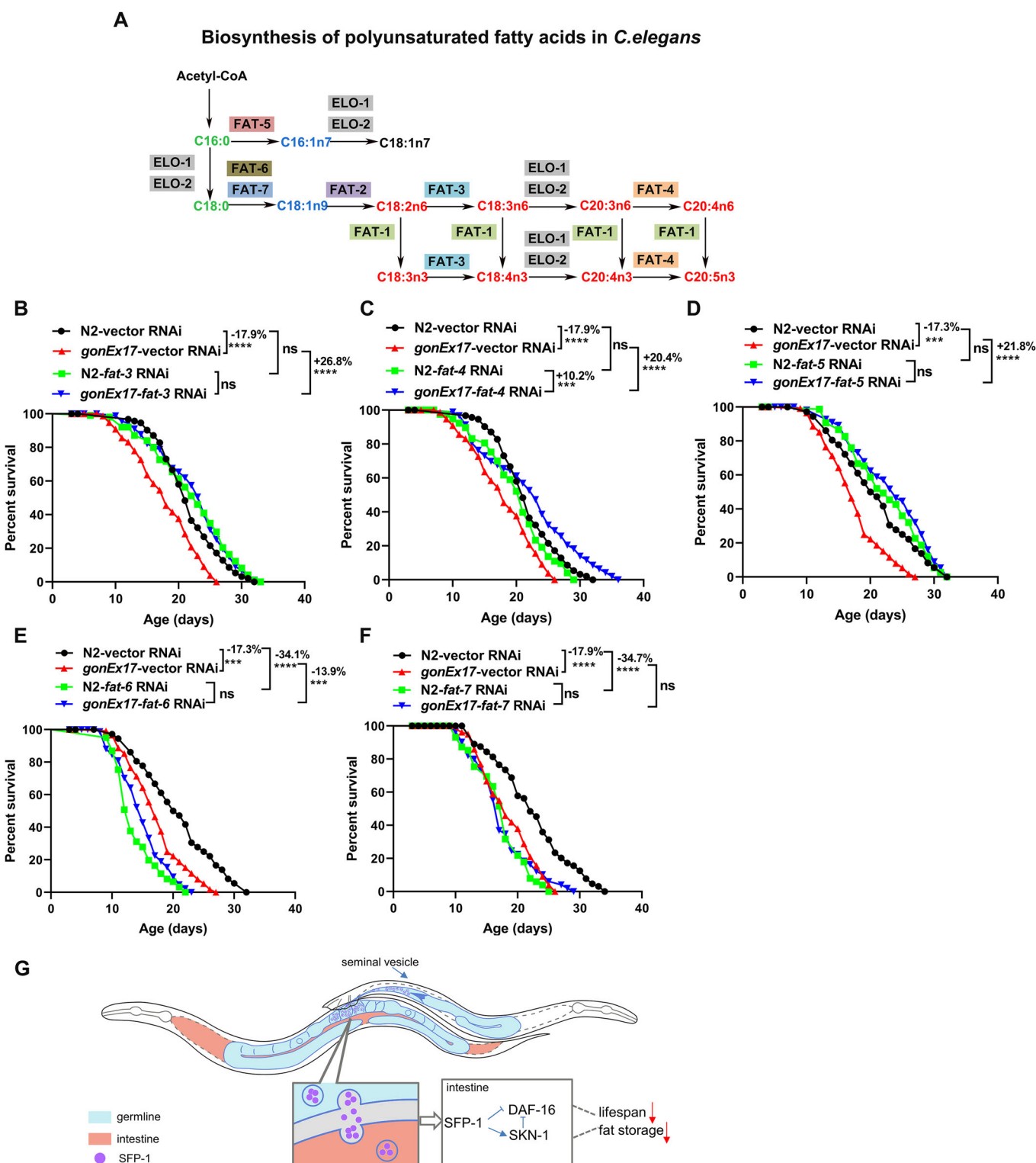

**A**

**Biosynthesis of polyunsaturated fatty acids in *C.elegans***

**G**

seminal vesicle

germline
intestine
SFP-1

intestine

SFP-1 — DAF-16 --- lifespan ↓
SKN-1 --- fat storage ↓

**Figure EV5. Fatty acid desaturases are involved in the lifespan regulation of intestinal overexpressing SFP-1 animals.**

(A) The pathway of PUFA synthesis in *Caenorhabditis elegans*. (B, C) Lifespan survival curves of WT animals and intestinal SFP-1 overexpressing (*intestine::sfp-1,gonEx17*) animals treated with EV or *fat* RNAi. Knockdown of *fat-3* (B) and *fat-4* (C) suppressed the short-lived phenotype of *intestine::sfp-1* transgenic worms. (B) N2-vector RNAi: 22.41 ± 0.75 days ($n = 64$); *gonEx17*-vector RNAi: 18.39 ± 0.49 days ($n = 77$); N2-*fat-3* RNAi: 23.96 ± 0.52 days ($n = 138$); *gonEx17-fat-3* RNAi: 23.32 ± 0.62 days ($n = 98$). $^{****}P = 1.13 \times 10^{-6}$, $^{****}P = 2.06 \times 10^{-11}$ (from left to right). (C) N2-vector RNAi: 22.41 ± 0.75 days ($n = 64$); *gonEx17*-vector RNAi: 18.39 ± 0.49 days ($n = 77$); N2-*fat-4* RNAi: 19.91 ± 0.60 days ($n = 74$); *gonEx17-fat-4* RNAi: 22.15 ± 0.75 days ($n = 93$). $^{****}P = 1.13 \times 10^{-6}$, $^{***}P = 0.0007$; $^{****}P = 5.17 \times 10^{-7}$ (from left to right). (D) Lifespan survival curves of WT animals and intestinal SFP-1 overexpressing (*intestine::sfp-1,gonEx17*) animals treated with EV or *fat-5* RNAi. Knockdown of *fat-5* suppressed the short-lived phenotype of *intestine::sfp-1* transgenic worms. N2-vector RNAi: 20.81 ± 1.01 days ($n = 36$); *gonEx17*-vector RNAi: 17.29 ± 0.41 days ($n = 113$); N2-*fat-5* RNAi: 21.31 ± 0.58 days ($n = 84$); *gonEx17-fat-5* RNAi: 21.06 ± 0.94 days ($n = 48$). $^{***}P = 1.07 \times 10^{-4}$, $^{****}P = 1.58 \times 10^{-6}$. (E, F) Lifespan survival curves of WT animals and *intestine::sfp-1* animals treated with EV or *fat* RNAi. *fat-6* (E) and *fat-7* (F) RNAi had almost no effect on the lifespan of worms with ectopic SFP-1 expression. (E) N2-vector RNAi: 20.81 ± 1.01 days ($n = 36$); N2-*fat-6* RNAi: 13.71 ± 0.43 days ($n = 61$); *gonEx17*-vector RNAi: 17.29 ± 0.41 days ($n = 113$); *gonEx17-fat-6* RNAi: 14.89 ± 0.42 days ($n = 84$). $^{***}P = 1.07 \times 10^{-4}$, $^{****}P = 4.04 \times 10^{-10}$, $^{***}P = 5.18 \times 10^{-5}$ (from left to right). (F) N2-vector RNAi: 22.41 ± 0.75 days ($n = 64$); N2-*fat-7* RNAi: 17.25 ± 0.56 days ($n = 51$); *gonEx17*-vector RNAi: 18.39 ± 0.49 days ($n = 77$); *gonEx17-fat-7* RNAi: 17.50 ± 0.66 days ($n = 49$). $^{****}P = 1.13 \times 10^{-6}$, $^{****}P = 2.45 \times 10^{-8}$ (from left to right). (G) A schematic model illustrating the pathway of SFP-1 transport and the regulation pathways in post-mating longevity and fat metabolism.

