## [Peer Review File · The EMBO Journal]

A seminal fluid protein SFP-1 regulates mated hermaphrodite aging and fat metabolism in *C. elegans*

Mingqing Chen and Jianke Gong

Corresponding author(s): Jianke Gong (jiankeg@hust.edu.cn)

Review Timeline:

Transfer from Review Commons:	5th Nov 24
Editorial Decision:	26th Nov 24
Revision Received:	19th Feb 25
Editorial Decision:	13th May 25
Revision Received:	21st Jun 25
Editorial Decision:	27th Aug 25
Revision Received:	8th Sep 25
Accepted:	29th Sep 25

Editor: Cornelius Schneider

Transaction Report: This manuscript was transferred to The EMBO JOURNAL following peer review at Review Commons.

Review #1

1. Evidence, reproducibility and clarity:

Evidence, reproducibility and clarity (Required)

Across the evolutionary spectrum, it has been observed that sexual interactions can significantly influence the physiology and somatic aging of individual animals. The underlying mechanisms of this phenomenon have only recently begun to be understood. In their manuscript titled "A male seminal fluid protein SFP-1 regulates hermaphrodite post-mating longevity and fat metabolism in *Caenorhabditis elegans*," Chen and Gong provide, for the first time, a comprehensive molecular understanding of how a male seminal fluid protein contributes to post-mating death in *C. elegans* hermaphrodites.

The authors identify SFP-1 as the key seminal fluid component by cleverly comparing two published genome-wide expression profiles of *C. elegans* males, significantly narrowing down the list of candidates. The role of SFP-1 in regulating post-mating lifespan in mated worms is convincingly verified through well-designed experiments using both a translational fusion reporter and a loss-of-function knockout strain. The authors then present a series of elegant experiments that comprehensively elucidate the functional mechanisms of SFP-1. On the male side, they show that SFP-1 is secreted into exophers as cargo from the male seminal vesicle, a process that requires the phospholipid scramblase ANOH-1 and ANOH-2. Once the seminal fluid is transferred into the hermaphrodites through mating, some exophers cross the gonad boundary and are taken up by the intestinal cells via endocytosis. The authors support their microscopic observations with convincing results from unmated hermaphrodites, where ectopic expression of SFP-1 in the intestine can largely mimic the physiological changes of mated hermaphrodites, such as post-mating shrinking and lifespan reduction. Further, the authors illustrate that the NTF2-like domain of SFP-1 is crucial for its function, and they show that SFP-1 interacts with the transcription factors SKN-1 and DAF-16, regulating lipid metabolism, particularly PUFA levels, in mated worms. Overall, this is a high-quality study that will be of considerable interest to researchers in this field. My detailed comments are listed below.

****Major:****

1. Lifespan comparisons

The manuscript includes a substantial number of lifespan assays, and the authors should be commended for their efforts. However, the data could be analyzed differently to provide more informative comparisons than those currently presented. For example, when

investigating whether a specific gene is involved in SFP-1-dependent post-mating lifespan regulation, the authors performed quite a few lifespan assays with the gonEx17 [Pges-1::sfp-1] strain. In the manuscript, the authors focus on comparing strains with SFP-1 overexpression, assuming that any factor increasing the lifespan of gonEx17 indicates its involvement in SFP-1-dependent lifespan regulation. However, I believe it would be equally, if not more, informative to assess whether a factor can suppress the lifespan reduction caused by SFP-1 overexpression.

For instance, in Figure 7F, in addition to the comparisons made by the authors-(1) N2 vector RNAi vs. N2 fat-2 RNAi and (2) gonEx17 vector RNAi vs. gonEx17 fat-2 RNAi-I suggest also comparing (3) N2 vector RNAi vs. gonEx17 vector RNAi and (4) N2 fat-2 RNAi vs. gonEx17 fat-2 RNAi. The comparison (3) serves as a positive control, where the lifespan of gonEx17 vector RNAi should be significantly shorter due to SFP-1 overexpression. (Of course, the gonEx17 strain should be backcrossed to the relevant strain to eliminate genetic background differences, which is standard procedure in the field, and I trust the authors have done so.) Comparing N2 fat-2 RNAi with gonEx17 fat-2 RNAi would clarify whether fat-2 is involved in SFP-1-dependent lifespan regulation. If fat-2 is required, there should be no significant difference in lifespans (as shown in Figure 7E). If fat-2 were not required, a similar lifespan reduction due to gonEx17, as seen in the control comparison (3), would be expected.

A different pattern emerges in Figure S4C-D, where these two ways of comparison yield opposite conclusions. The authors suggest that RNAi of sek-1 and pmk-1 increases the lifespan of SFP-1-overexpressing worms, implicating these genes in SFP-1-dependent lifespan regulation. However, when comparing the lifespans differently, gonEx17 still induces lifespan reduction in the presence of sek-1 or pmk-1 RNAi, though to a lesser extent than with vector RNAi. This result contrasts with the strong evidence presented for skn-1 (as shown in Figures 5C and 5E, where SFP-1-mediated lifespan regulation is entirely dependent on skn-1), suggesting that sek-1 and pmk-1 may play a much lesser role. The authors should conduct additional lifespan comparisons and revise their model accordingly. Below are the relevant figure panels: Figures 5E, S4C-D, S5 D-I, 7E-F, and S6A-E.

2. How ectopic expression of SFP-1 imitates the post-mating phenotypes

The authors demonstrate that ectopic expression of SFP-1 in the intestine of unmated hermaphrodites can lead to a decrease in body size over time, similar to that observed in mated hermaphrodites (Figure S1B). They suggest that this intestinal SFP-1 overexpression alters germline activity, as evidenced by the increased brood size (Figure S1A), which they

propose is responsible for the shrinking phenotype. However, in Figure 1A, the brood size of gonEx17 is still significantly lower than that of mated hermaphrodites. This discrepancy makes the current data less convincing. To better understand whether SFP-1 overexpression in the intestine requires an intact germline to mediate lifespan decrease in self-fertilized hermaphrodites, it would be helpful to assess these phenotypes in germlineless worms such as *glp-1(e2141)* with SFP-1 overexpression.

****Minor:****

1. The figure legends should include more detailed information about the results. Currently, some lifespan data are missing in the figure legends for the following figures: Figure 4A, 4C, S3A, 5E, S4A-D, S5D-I, 7E-F, S6A-E.
2. There are quite a few discrepancies between the supplementary lifespan results table and the figure panel labeling in the manuscript.
3. To broaden the manuscript's appeal, the authors should consider including a discussion comparing SFP-1 with the sex peptide proteins previously reported to regulate post-mating physiology and lifespan in *Drosophila*. For instance, do the fly sex peptide proteins also contain an NTF-2 domain?
4. The language should be refined to enhance the clarity and comprehensibility of the manuscript.

2. Significance:

Significance (Required)

Sexual interactions have been observed to significantly impact the physiology and somatic aging of individual animals across the evolutionary spectrum. The mechanisms underlying this phenomenon have only recently begun to be understood. In this manuscript, Chen and Gong offer the first comprehensive molecular insight into how a male seminal fluid protein contributes to post-mating death in *C. elegans* hermaphrodites. This high-quality study will be of considerable interest to researchers in the fields of *C. elegans* biology, sexual interactions, lipid metabolism, and aging. The manuscript's appeal could be further enhanced by including additional statistical comparisons and a more in-depth discussion on the evolutionary conservation of the seminal fluid protein identified by the authors.

3. How much time do you estimate the authors will need to complete the suggested revisions:

Estimated time to Complete Revisions (Required)

(Decision Recommendation)

Between 1 and 3 months

4. Review Commons values the work of reviewers and encourages them to get credit for their work. Select 'Yes' below to register your reviewing activity at Web of Science Reviewer Recognition Service (formerly Publons); note that the content of your review will not be visible on Web of Science.

Yes

Review #2

1. Evidence, reproducibility and clarity:

Evidence, reproducibility and clarity (Required)

****Summary:****

In this manuscript the authors present analysis of a seminal fluid protein, SFP-1, in *C. elegans*. The authors present relatively convincing data that this protein is expressed in males, transferred to hermaphrodites upon mating, and influences hermaphrodite longevity post-mating. Specifically, hermaphrodites mated to males lacking *sfp-1* live longer than hermaphrodites mated to wild-type males with SFP-1. The authors then seek to identify the tissue from which this protein exerts its effects and connect it to others pathways (FOXO/DAF-16, SKN-1, PUFA biogenesis). They do show that ectopic expression in the hermaphrodite gut has an effect, but evidence that this is physiological or that the protein makes it to the gut after mating is not substantiated by the data presented. The data suggesting the effect occurs through *skn-1* is interesting but could use more careful discussion, while the data seems to suggest that the effect is independent of *daf-16*, however it is not clear that this is the conclusion of the authors. This manuscript represents a large body of work and contains some very interesting results and observations. However, in its current state it is not yet ready for publication. It requires extensive editing for grammar and language for the sake of clarity and accuracy. Other concerns regarding images, controls, and interpretation of results also exist.

****Major comments:****

- Fig 1D and 1E need higher magnification images for the reader to accurately assess the data. While it may be an artifact of the pdf format provided, the resolution of all fluorescent images was also too low for accurate assessment.
- Fig 1D vs unmated in 1E, why does E show fluorescence in the pharynx but D doesn't? Aren't these images of the same genotype? What is the interpretation that the pharynx expression also appears brighter after mating in 1E?
- In Figure 2A the background for the N2 control is black while the background for the yfp image is green. This strongly indicates that the levels or image exposure are not comparable between the images and therefore does not allow one to make any conclusions about what may be autofluorescence and what may real signal in the yfp images. For this reason, the localization the authors claim to see in the intestine is not convincing and not supported by the data as shown.
- Aren't Fig 4D and S3B (lower) of the same genotype? But the "nuclear" localization that is supposedly in 4D isn't observable in S3B.
- Due to gut autofluorescence it's important to see a control image for 4D that shows worms that don't express *sfp-1::yfp* under the same imaging conditions (exposure, brightness levels, etc).
- Figure 5B at minimum needs arrows in the middle column indicating where the nucleus is. Imaging with a nuclear signal would be more convincing, but at least need to be able to point out where the nucleus is. Otherwise it's just an overall loss of signal and not a change in localization.
- Figure 5F again seems to be observing overall differences in protein levels but does not convincingly show nuclear localization.
- Figure 6A needs an unmated control.
- Figure 6I, the images are not clear enough to convincingly show germline vs soma (vs gut on top of germline based on the orientation of the worm on the slide). The conclusion about asdf is not supported based on these data alone.
- Data in final results section and Figure 7 does not make much sense and its physiological meaning is not clear as it's based entirely on over-expression in the intestine. If PUFAs are lowered in response to SFP-1, and SFP-1 expression reduces lifespan then why does further reducing PUFAs with the fat mutants extend lifespan?
- Supplemental Figure 1C needs a control of hermaphrodites on unconditioned plates. There is clearly variability from experiment to experiment in regards to longevity. This control is needed to show that the expected reduction on conditioned plates was indeed observed.

- Bafilomycin A1 (bafA1) does not inhibit endocytosis. Inhibiting the maturation of an endosome to lysosome is not the same as inhibiting endocytosis. Need to remove or reinterpret these results.
- Line 209-211 and Figure 3A, how were exophers determined? What evidence was used to show that any observed puncta are exophers and not intracellular vesicles? How were punta/vesicles defined for quantification? Also important to note that these "vesicular" structures are not observed when the protein is in the hermaphrodite reproductive tract after mating.
- anoh-1 and anoh-2 mutations were in males, results can not be used to support a model of what happens to the protein once it is transferred to the hermaphrodite.
- Line 230-231: This result is very confusing. Sperm are the limiting factor for hermaphrodite self-fertility. These data would suggest that SFP-1 in the hermaphrodite intestine led to increased spermatogenesis/more sperm being made by the hermaphrodite in the L4 stage? There are no previous reports of long-lived strains having increased self-fertility. Were the brood sizes for N2 unmated and gonEx17 unmated measured in the same assay/at the same time?
- Line 235-236 and Fig S3B: While possible, this result is not consistent with other reported data for other proteins. Did the authors look in the coelomocytes where it would be expected to be found if it was secreted from the muscle? Did they do a control with ss:GFP (GFP with a signal sequence) which is known to be secreted out of the muscle for comparison?
- Grammar and coherence of writing need to be addressed. Examples include, but are not limited to:
 - "sex interaction" - Do the authors mean interaction between the sexes?
 - "cross over the somatic gonad uterus"
 - "using the sfp-1 gene its own promoter and performed confocal imaging."

****Minor comments:****

- Worth noting that over-expression (all experiments with SFP-1::GFP) could impact whether the protein ends up in vesicles and the type of localization seen in both the male and hermaphrodite. While not necessary to repeat with an endogenously tagged line, this caveat to the results should be acknowledged.
- Introduction, Line 41: Either this is the wrong reference or "sex peptides" should be "seminal fluid proteins". "Sex peptide" is a specific SFP in Drosophila and reference #7 is not about sex peptide.
- For Image analysis of SFP-1::GFP, how was the region of the seminal vesicle determined/defined?

- Fig 6C it appears that the pattern of the staining changes but where there is staining the intensity doesn't appear to be different from N2 control.
- It would be helpful if the authors were consistent in the color choices for graphs and figures. For example, control (N2 x N2) is black in Fig 1G but red in Fig 3C.
- Line 55-56 in Introduction needs a citation.
- Line 100-102 states "SFP-1 is the first protein demonstrated to be a transferred component in the seminal fluid into *C. elegans*", but this has already been published for TRY-5.
- Line 141-142 and Fig 1E and 1F, it seems strange that if the protein is transferred from the male to the hermaphrodite during mating it is seen in higher levels in the male after mating. Do the authors have any possible explanation for this?
- Line 149-150 "hermaphrodite brood size was enhanced ... by the *sfp-1* mutant male". This is wrong. Brood size is NOT enhanced compared to WT mating group. Higher than unmated, but significantly lower than mated with WT males.
- Optional: Would like to see the transfer upon mating to N2 hermaphrodites. Don't expect to see a difference compared to the *glo-4* mutant, but signal seems bright enough right after mating to be able to observe without having to use a mutant hermaphrodite?
- Don't understand the purpose of S2B. Sperm are not known to leave the reproductive tract under wild type conditions. What were the authors trying to test here?
- Line 238-240: Was this expected? Is this because they don't function through the intestine or because they don't affect lifespan at all?
- Line 272-274, Predictions suggest this. The authors don't provide empirical evidence of this interaction. Need to adjust language.
- Longevity data with *daf-16* and *skn-1* are interesting, but could use more careful discussion of possible interpretations regarding independent pathways (*daf-16*) or downstream actors (*skn-1*). These are certainly challenging data to know how to interpret but clearer language regarding results and interpretation of those results is needed.

2. Significance:

Significance (Required)

This study is of a previously uncharacterized protein, SFP-1, that was identified as a likely component of male seminal fluid in *C. elegans* in previous work. The work here confirms its identity as a seminal fluid protein and links it to the reduced longevity seen in mated hermaphrodites. Of interest to *C. elegans* researchers and researchers interested in male fertility factors and molecular mechanisms of post-mating responses, especially as they relate to longevity. However, care needs to be taken in regard to controls, presentation of data, and interpretation of data before publication.

3. How much time do you estimate the authors will need to complete the suggested revisions:

Estimated time to Complete Revisions (Required)

(Decision Recommendation)

More than 6 months

No

Revision Plan

Manuscript number: RC-2024-02650

Corresponding author(s): Jianke Gong

1. General Statements [optional]

We would like to thank the referees for their precious time reviewing our manuscript and their valuable comments to improve this study. Our detailed responses to their concerns are listed below on a point-by-point basis.

2. Description of the planned revisions

This document reports the comments made by two Review Commons reviewers (in black font). In response to this feedback, we have performed additional experiments and revised our manuscript accordingly. Below, we address each comment, detailing the revisions we made (in blue font) as well as our action plan for each item (in bolded blue font).

Revision Plan

Reviewer #1 (Evidence, reproducibility and clarity (Required)):

Across the evolutionary spectrum, it has been observed that sexual interactions can significantly influence the physiology and somatic aging of individual animals. The underlying mechanisms of this phenomenon have only recently begun to be understood. In their manuscript titled "A male seminal fluid protein SFP-1 regulates hermaphrodite post-mating longevity and fat metabolism in *Caenorhabditis elegans*," Chen and Gong provide, for the first time, a comprehensive molecular understanding of how a male seminal fluid protein contributes to post-mating death in *C. elegans* hermaphrodites.

The authors identify SFP-1 as the key seminal fluid component by cleverly comparing two published genome-wide expression profiles of *C. elegans* males, significantly narrowing down the list of candidates. The role of SFP-1 in regulating post-mating lifespan in mated worms is convincingly verified through well-designed experiments using both a translational fusion reporter and a loss-of-function knockout strain. The authors then present a series of elegant experiments that comprehensively elucidate the functional mechanisms of SFP-1. On the male side, they show that SFP-1 is secreted into exophers as cargo from the male seminal vesicle, a process that requires the phospholipid scramblase ANOH-1 and ANOH-2. Once the seminal fluid is transferred into the hermaphrodites through mating, some exophers cross the gonad boundary and are taken up by the intestinal cells via endocytosis. The authors support their microscopic observations with convincing results from unmated hermaphrodites, where ectopic expression of SFP-1 in the intestine can largely mimic the physiological changes of mated hermaphrodites, such as post-mating shrinking and lifespan reduction. Further, the authors illustrate that the NTF2-like domain of SFP-1 is crucial for its function, and they show that SFP-1 interacts with the transcription factors SKN-1 and DAF-16, regulating lipid metabolism, particularly PUFA levels, in mated worms. Overall, this is a high-quality study that will be of considerable interest to researchers in this field. My detailed comments are listed below.

Major:

1. Lifespan comparisons

The manuscript includes a substantial number of lifespan assays, and the authors should be

Revision Plan

commended for their efforts. However, the data could be analyzed differently to provide more informative comparisons than those currently presented. For example, when investigating whether a specific gene is involved in SFP-1-dependent post-mating lifespan regulation, the authors performed quite a few lifespan assays with the gonEx17 [Pges-1::sfp-1] strain. In the manuscript, the authors focus on comparing strains with SFP-1 overexpression, assuming that any factor increasing the lifespan of gonEx17 indicates its involvement in SFP-1-dependent lifespan regulation. However, I believe it would be equally, if not more, informative to assess whether a factor can suppress the lifespan reduction caused by SFP-1 overexpression.

Reply: Thank you for your recognition and suggestion, which is very helpful.

For instance, in Figure 7F, in addition to the comparisons made by the authors-(1) N2 vector RNAi vs. N2 fat-2 RNAi and (2) gonEx17 vector RNAi vs. gonEx17 fat-2 RNAi-I suggest also comparing (3) N2 vector RNAi vs. gonEx17 vector RNAi and (4) N2 fat-2 RNAi vs. gonEx17 fat-2 RNAi. The comparison (3) serves as a positive control, where the lifespan of gonEx17 vector RNAi should be significantly shorter due to SFP-1 overexpression. (Of course, the gonEx17 strain should be backcrossed to the relevant strain to eliminate genetic background differences, which is standard procedure in the field, and I trust the authors have done so.) Comparing N2 fat-2 RNAi with gonEx17 fat-2 RNAi would clarify whether fat-2 is involved in SFP-1-dependent lifespan regulation. If fat-2 is required, there should be no significant difference in lifespans (as shown in Figure 7E). If fat-2 were not required, a similar lifespan reduction due to gonEx17, as seen in the control comparison (3), would be expected.

Reply: Thank you for your valuable advice. We have included a comparison between N2 in *fat-2* RNAi and gonEx17 in *fat-2* RNAi. Our analysis revealed no significant differences between these two groups. Similar findings were observed for *fat-1*, *fat-3*, *fat-4*, and *fat-5*. This suggests that the inhibition of a single PUFA synthase may extend lifespan, potentially due to redundancy in the function of the fat enzymes within the synthetic pathway.

A different pattern emerges in Figure S4C-D, where these two ways of comparison yield opposite conclusions. The authors suggest that RNAi of *sek-1* and *pmk-1* increases the lifespan of SFP-1-overexpressing worms, implicating these genes in SFP-1-dependent lifespan regulation. However, when comparing the lifespans differently, gonEx17 still induces lifespan reduction in the presence of *sek-1* or *pmk-1* RNAi, though to a lesser extent than with vector RNAi. This

Revision Plan

result contrasts with the strong evidence presented for *skn-1* (as shown in Figures 5C and 5E, where SFP-1-mediated lifespan regulation is entirely dependent on *skn-1*), suggesting that *sek-1* and *pmk-1* may play a much lesser role. The authors should conduct additional lifespan comparisons and revise their model accordingly. Below are the relevant figure panels: Figures 5E, S4C-D, S5 D-I, 7E-F, and S6A-E.

Reply: We apologize for not clearly explaining the interaction between *sek-1*, *pmk-1*, and *skn-1*. The SEK-1-PMK-1 pathway is not the only pathway upstream of SKN-1; therefore, RNAi targeting *sek-1* or *pmk-1* can only partially downregulate the expression of SKN-1 rather than completely abolishing the lifespan reduction induced by *gonEx17*. In contrast, SKN-1 expression can be further diminished by directly lacking *skn-1*, which extends lifespan.

2. How ectopic expression of SFP-1 imitates the post-mating phenotypes

The authors demonstrate that ectopic expression of SFP-1 in the intestine of unmated hermaphrodites can lead to a decrease in body size over time, similar to that observed in mated hermaphrodites (Figure S1B). They suggest that this intestinal SFP-1 overexpression alters germline activity, as evidenced by the increased brood size (Figure S1A), which they propose is responsible for the shrinking phenotype. However, in Figure 1A, the brood size of *gonEx17* is still significantly lower than that of mated hermaphrodites. This discrepancy makes the current data less convincing. To better understand whether SFP-1 overexpression in the intestine requires an intact germline to mediate lifespan decrease in self-fertilized hermaphrodites, it would be helpful to assess these phenotypes in germlineless worms such as *glp-1(e2141)* with SFP-1 overexpression.

Reply: We appreciated that you pointed out the importance of further investigation on the effect of intestinal SFP-1 overexpression on germline to induce lifespan shortening. Our observations indicate that the brood size of *gonEx17* remains significantly lower than that of mated hermaphrodites. This difference may come from the contribution of sperm from males, which enhances the brood size of mated hermaphrodites, whereas *gonEx17* relies only on its own sperm. We propose that the increase in brood size for *gonEx17* may be attributed to germline activity. Specifically, an increase in meiotic zones promotes a greater production of oocytes in post-mating hermaphrodites (PMID: 24356112). **We are currently in the process of crossing**

Revision Plan

gonEx17 with a *glp-1* background to replace the N2 background, and we will perform lifespan experiments to determine whether the overexpression of SFP-1 in the intestine requires an intact germline to mediate the lifespan shorten in hermaphrodites. Furthermore, previous study found that *glp-1* mutants still exhibit a shortened lifespan after mating with N2 males, they suggested that the shortened lifespan may not be directly linked to an active germline or increased brood size (PMID: 24356112). We will address this in our manuscript based on the results of the lifespan experiments. Thank you for your insightful suggestion.

Minor:

1. The figure legends should include more detailed information about the results. Currently, some lifespan data are missing in the figure legends for the following figures: Figure 4A, 4C, S3A, 5E, S4A-D, S5D-I, 7E-F, S6A-E.

Reply: We've added the figure legends in the text according to your advice in line 545-549, line 552-557, line 566-570, line 601-604, line 619-642, line 679-687, line 702-711, line 715-726.

2. There are quite a few discrepancies between the supplementary lifespan results table and the figure panel labeling in the manuscript.

Reply: We've screened the whole manuscript and corrected the non-corresponding labeling in the text. Thank you.

3. To broaden the manuscript's appeal, the authors should consider including a discussion comparing SFP-1 with the sex peptide proteins previously reported to regulate post-mating physiology and lifespan in *Drosophila*. For instance, do the fly sex peptide proteins also contain an NTF-2 domain?

Reply: We've added more discussion about comparative sex peptides with SFP-1. Related text on line 376-391.

4. The language should be refined to enhance the clarity and comprehensibility of the manuscript.

Reply: We've polished the manuscripts and improved the language. Again, we really appreciate your great advice on our study and hope it would meet your command.

Revision Plan

Reviewer #1 (Significance (Required)):

Sexual interactions have been observed to significantly impact the physiology and somatic aging of individual animals across the evolutionary spectrum. The mechanisms underlying this phenomenon have only recently begun to be understood. In this manuscript, Chen and Gong offer the first comprehensive molecular insight into how a male seminal fluid protein contributes to post-mating death in *C. elegans* hermaphrodites. This high-quality study will be of considerable interest to researchers in the fields of *C. elegans* biology, sexual interactions, lipid metabolism, and aging. The manuscript's appeal could be further enhanced by including additional statistical comparisons and a more in-depth discussion on the evolutionary conservation of the seminal fluid protein identified by the authors.

Reviewer #2 (Evidence, reproducibility and clarity (Required)):

Summary:

In this manuscript the authors present analysis of a seminal fluid protein, SFP-1, in *C. elegans*. The authors present relatively convincing data that this protein is expressed in males, transferred to hermaphrodites upon mating, and influences hermaphrodite longevity post-mating. Specifically, hermaphrodites mated to males lacking *sfp-1* live longer than hermaphrodites mated to wild-type males with SFP-1. The authors then seek to identify the tissue from which this protein exerts its effects and connect it to others pathways (FOXO/DAF-16, SKN-1, PUFA biogenesis). They do show that ectopic expression in the hermaphrodite gut has an effect, but evidence that this is physiological or that the protein makes it to the gut after mating is not substantiated by the data presented. The data suggesting the effect occurs through *skn-1* is interesting but could use more careful discussion, while the data seems to suggest that the effect is independent of *daf-16*, however it is not clear that this is the conclusion of the authors. This manuscript represents a large body of work and contains some very interesting results and observations. However, in its current state it is not yet ready for publication. It requires extensive editing for grammar and language for the sake of clarity and accuracy. Other concerns regarding images, controls, and interpretation of results also exist.

Revision Plan

Major comments:

- Fig 1D and 1E need higher magnification images for the reader to accurately assess the data. While it may be an artifact of the pdf format provided, the resolution of all fluorescent images was also too low for accurate assessment.

Reply: We sincerely apologize for the confusion. We have added higher magnification images (120X) in Figure 1D, while the images of mated males at 60X magnification will be displayed below. Additionally, we have re-uploaded all original TIFF versions of the figures, and we hope the resolution meets for the reader to accurately assess the data.

- Fig 1D vs unmated in 1E, why does E show fluorescence in the pharynx but D doesn't? Aren't these images of the same genotype? What is the interpretation that the pharynx expression also appears brighter after mating in 1E?

Reply: We apologize for the discrepancies observed in the fluorescence within the pharynx. Firstly, since SFP-1 is exclusively expressed in males, we added a pharynx fluorescence as a marker during microinjection. Our primary focus was on the fluorescence in the seminal vesicle, leading us to ignore the effects of pharynx fluorescence. Mating does not affect pharynx fluorescence. We think this variability is due to the microscope focus on the seminal vesicle

Revision Plan

when the image was captured; however, the markers expressed in the pharynx may be out of focus. We have standardized these figures to ensure that images have the same quality in both groups.

- In Figure 2A the background for the N2 control is black while the background for the yfp image is green. This strongly indicates that the levels or image exposure are not comparable between the images and therefore does not allow one to make any conclusions about what may be autofluorescence and what may real signal in the yfp images. For this reason, the localization the authors claim to see in the intestine is not convincing and not supported by the data as shown.

Reply: We're sorry for confusing you. We've reunified fluorescence images in Figure 2A in the same brightness and contrast. Our conclusions remained unchanged. We have placed *P_{sfp-1::sfp-1::GFP}* (1-10) male with *P_{ges-1::sfp-1::GFP}* (11x7) hermaphrodites. **We will perform observations to determine whether fluorescent signals can be detected in the intestines of mated hermaphrodites.**

- Aren't Fig 4D and S3B (lower) of the same genotype? But the "nuclear" localization that is supposedly in 4D isn't observable in S3B.

Reply: Thank you for your advice. We have observed that expressing SFP-1 in the intestine leads to aggregation around the nuclei of intestinal cells. In Figure S3B, we examine whether SFP-1 expression in the intestine leaks fluorescent signals to other tissues. In this experiment, we did not focus on nuclear localization; therefore, no staining of the nuclei was conducted. We will revise the language to enhance clarity and avoid confusion.

- Due to gut autofluorescence it's important to see a control image for 4D that shows worms that don't express *sfp-1::yfp* under the same imaging conditions (exposure, brightness levels, etc).

Reply: Thank you for your suggestion, **we will perform the experiment using the progeny in the *P_{ges-1::sfp-1::yfp}*** strain which have not fluoresce, which will help us avoid the effects of intestinal autofluorescence.

- Figure 5B at minimum needs arrows in the middle column indicating where the nucleus is.

Imaging with a nuclear signal would be more convincing, but at least need to be able to point out where the nucleus is. Otherwise it's just an overall loss of signal and not a change in localization.

Reply: Thank you for your suggestion. We have added arrows to Figure 5B to clearly indicate the location of the nucleus, and we apologize for any confusion this may have caused.

Revision Plan

- Figure 5F again seems to be observing overall differences in protein levels but does not convincingly show nuclear localization.

Reply: Thank you for your suggestion. **We will examine changes in nuclear localization in unmated *skn-1::gfp* and in hermaphrodites after mating with different males (N2, *sfp-1, sfp::yfp*, respectively) as well as *skn-1::gfp ; ges-1::sfp-1::mcherry2*.**

- Figure 6A needs an unmated control.

Reply: Thank you for your suggestion, we have added unmated as a control and performed the experiment, as shown in Figure 6A, our conclusions have not changed.

- Figure 6I, the images are not clear enough to convincingly show germline vs soma (vs gut on top of germline based on the orientation of the worm on the slide). The conclusion about asdf is not supported based on these data alone.

Reply: Thank you for your suggestion, we have selected more representative images, the *vit* family is responsible for transferring stored lipids from the intestine to the germline. Inhibition of the *vit* family appears to rescue the Asdf associated with intestinal overexpression of SFP-1, potentially by preventing fat loss in the intestine. Our experiments regarding the *vit* family may provide additional support for Asdf. Furthermore, we observed that Asdf is not present in the animals with SFP-1 overexpression when SKN-1 is absent.

- Data in final results section and Figure 7 does not make much sense and its physiological meaning is not clear as it's based entirely on over-expression in the intestine. If PUFAs are lowered in response to SFP-1, and SFP-1 expression reduces lifespan then why does further reducing PUFAs with the fat mutants extend lifespan?

Reply: Thank you for pointing out this problem. Our consideration regarding this part is based on findings that genes associated with fatty acid biosynthesis exhibit significant expression changes in mated hermaphrodites (PMID: 34179018). Additionally, *fat-2* mutants that contain excess endogenous oleic acid are protected from lifespan loss after mating. We hypothesize that *fat* genes are crucial in the regulation of longevity and lipid levels by SFP-1. It has been observed that polyunsaturated fatty acids (PUFAs) decrease in response to SFP-1, which shorten lifespan. This reduction in PUFAs allows *fat* mutants to extend their lifespan, as the prevention of PUFA synthesis facilitates the endogenous accumulation of oleic acid, thus contributing to lifespan extension. Our experimental results indicated that SFP-1 is unable to prolong lifespan in the absence of *fat-6* and *fat-7* due to a further deficiency of oleic acid. Conversely, *fat-1, fat-2, fat-3,*

Revision Plan

fat-4, and *fat-5* have all shown lifespan extension, likely due to their role in preventing the depletion of PUFAs. As you point out, our entire data is based on overexpressing lines. **We will examine changes in fatty acid content in unmated hermaphrodites and in hermaphrodites after mating with different males (N2, *sfp-1*, *sfp::yfp*, respectively) as well as lifespan experiments in the presence of inhibition of *fat-1*, *fat-2*.** We will also check our language to ensure that readers understand our rationale for inhibiting the fat family. We appreciate your suggestions, which are invaluable in enhancing the quality of our work.

- Supplemental Figure 1C needs a control of hermaphrodites on unconditioned plates. There is clearly variability from experiment to experiment in regards to longevity. This control is needed to show that the expected reduction on conditioned plates was indeed observed.

Reply: Thank you for your suggestion. We have revised the lifespan result figure, as illustrated in Fig. S1C. This revision includes the data for N2 without any treatment (with no males crawling through the plates) as a control. Our conclusions remain unchanged.

- Bafilomycin A1 (*bafA1*) does not inhibit endocytosis. Inhibiting the maturation of an endosome to lysosome is not the same as inhibiting endocytosis. Need to remove or reinterpret these results.

Reply: We thank the reviewer for raising this important point. Previous study has demonstrated that Bafilomycin-A1 (*bafA1*) can disrupt different aspects of endocytosis (PMID: 22902558). Specifically, *bafA1* prevents acidification-dependent vesicle turnover, which is required for endosome maturation (PMID: 16415858).

- Line 209-211 and Figure 3A, how were exophers determined? What evidence was used to show that any observed puncta are exophers and not intracellular vesicles? How were punta/vesicles defined for quantification? Also important to note that these "vesicular" structures are not observed when the protein is in the hermaphrodite reproductive tract after mating.

Reply: We appreciate the reviewer's insightful comments. Upon discovering that SFP-1 localized in the seminal vesicle of male worms, we found a relevant description in WormAtlas: "Shortly after their differentiation, the small cells exhibit a granular appearance when viewed with DIC optics, with small blebs evident on their luminal surfaces" (PMID: 478167). We examined *sfp-1::yfp* males with DIC optics and observed similar vesicles; however, we were unable to define these structures or find relevant literature to confirm it. Similar vesicles were also identified in

Revision Plan

TRY-5. Previous studies have identified a series of proteins that may serve as extracellular vesicle (EV) cargo (PMID: 35334227), including SFP-1. We attempted to demonstrate that the observed vesicles could be either EVs or exophers, although neither category has a definitive marker in worms. Based on the evidence that these structures can be secreted from the males, we believe the possibility of them being intracellular vesicles is unlikely. Our quantification relies on recognizable globular secretory vesicles. We will adjust the language in our manuscript to describe them as secretory vesicles and demonstrate experimentally in the future that they are EV or exopher. Additionally, we noted that these vesicles were not observed in mated hermaphrodites, which we speculate may result from diffusion after entering the uterus. We also explored existing literature on male proteins that enter hermaphrodites after mating, with TRY-5 expressed in males and the visibility of globular secretory vesicles being restricted to males and diffused after mating.

- *anoh-1* and *anoh-2* mutations were in males, results can not be used to support a model of what happens to the protein once it is transferred to the hermaphrodite.

Reply: *anoh-1* and *anoh-2* mutation males had significantly fewer secretory vesicles, which due to the inefficiency of the membrane fusion and lower amount of SFP-1 protein in seminal vesicle. This suggested that much fewer SFP-1 got transferred to the mated hermaphrodite. **We will overexpress ANOH-1 and ANOH-2 in the *sfp-1* mutant background, which is characterized by the absence of cargo within the secretory vesicles. We will use this to investigate the impact of membrane fusion functionality on the lifespan of mated hermaphrodites**

- Line 230-231: This result is very confusing. Sperm are the limiting factor for hermaphrodite self-fertility. These data would suggest that SFP-1 in the hermaphrodite intestine led to increased spermatogenesis/more sperm being made by the hermaphrodite in the L4 stage? There are no previous reports of long-lived strains having increased self-fertility. Were the brood sizes for N2 unmated and gonEx17 unmated measured in the same assay/at the same time?

Reply: We're very sorry for causing you confuse. To clarify, the brood size of unmated N2 and unmated gonEx17 was indeed measured during the same trial and at the same time. we have three transgenic lines of microinjected *ges-1::sfp-1*, and we have measured brood size in all of them. All three lines exhibited an increase in brood size.

Compared to N2, gonEX17 exhibited increased brood size and shortened lifespan, rather than increased self-fertility in long-lived strains. We apologize for the misunderstanding and will review the language for clarity and accuracy. We hypothesize that SFP-1 in the gut of the hermaphrodite may lead to over-activity of the germline, resulting in increased sperm production at the L4 stage. While we have not yet conducted experiments to confirm this, it remains a potential direction for future research. Previous studies have demonstrated that mating causes activity in the germline of hermaphrodites and found that mated worms have significantly fewer nuclei in the mitotic region. (PMID: 24356112). **We will verify if there is a similar phenomenon occurs in the gonEX17. Additionally, we are constructing gonEX17 in a *glp-1* mutant background to assess the impact of germline activity on lifespan. Thanks again for pointing this out.**

- Line 235-236 and Fig S3B: While possible, this result is not consistent with other reported data for other proteins. Did the authors look in the coelomocytes where it would be expected to be found if it was secreted from the muscle? Did they do a control with ss:GFP (GFP with a signal sequence) which is known to be secreted out of the muscle for comparison?

Reply: We chose muscle tissue to investigate the secretion of SFP-1 due to its known tendency to secrete proteins. However, our observations indicated no fluorescent signals present in the coelomocytes. We explored existing literature and found that LECT-2, the ortholog of leukocyte cell-derived chemotaxin-2 (LECT2), is secreted from the muscles (PMID: 27705746). **We will try to use the secreted signal sequence of this protein coupled with GFP (ss::GFP) as a control to observe the secreted condition for the muscle tissue.**

Revision Plan

- Grammar and coherence of writing need to be addressed. Examples include, but are not limited to:

"sex interaction" - Do the authors mean interaction between the sexes?

"cross over the somatic gonad uterus"

"using the *sfp-1* gene its own promoter and performed confocal imaging."

Reply: We've checked the grammar and the proper nouns in the text and polished the whole manuscript. **We will continue to do this** and hope it would meet your command.

Minor comments:

- Worth noting that over-expression (all experiments with SFP-1::GFP) could impact whether the protein ends up in vesicles and the type of localization seen in both the male and hermaphrodite. While not necessary to repeat with an endogenously tagged line, this caveat to the results should be acknowledged.

Reply: Thank you very much for your suggestion, **we will construct an endogenously tagged line with mNeonGreen at the C-terminus of SFP-1 using CRISPR-KI and repeat the experiments**, thank you very much for your suggestion to make our results more credible.

- Introduction, Line 41: Either this is the wrong reference or "sex peptides" should be "seminal fluid proteins". "Sex peptide" is a specific SFP in *Drosophila* and reference #7 is not about sex peptide.

Reply: We've corrected the relative description in line41. We changed sex peptides to seminal fluid products in line 36, thank you for your careful review.

- For Image analysis of SFP-1::GFP, how was the region of the seminal vesicle determined/defined?

Reply: Thank you for highlighting this point. The *C. elegans* male gonad is essentially a tubular structure that primarily consists of the seminal vesicle, which serves as a storage organ, along with a subset of somatic gonadal cells that surround the spermatids. To confirm the expression of

Revision Plan

SFP-1 in the seminal vesicle, we utilized MitoTracker, a fluorescent dye, to label the spermatids in *sfp-1::yfp* males. Our observations revealed that SFP-1 surrounds the spermatids, indicating specific expression in the seminal vesicle. Attached below is a schematic diagram of the male gonads from WormAtlas, with the added experimental results in S1D, and the text description in line 119-123.

- Fig 6C it appears that the pattern of the staining changes but where there is staining the intensity doesn't appear to be different from N2 control.

Reply: Thanks for pointing this out, we are using D5 worms, which results in some degree of fat loss even in the N2 control. Additionally, the Oil Red staining stains the eggs in the body of the worms, so despite the obvious fat loss that occurs with intestinal overexpression of SFP-1, the eggs in the body are still able to be stained with Oil Red. We've reselected representative images in Figure 6C to stand for our conclusions, thank you.

- It would be helpful if the authors were consistent in the color choices for graphs and figures. For example, control (N2 x N2) is black in Fig 1G but red in Fig 3C.

Reply: We've unified the bar of the same group with different figures.

- Line 55-56 in Introduction needs a citation.

Reply: Relative citation was added as reference #9 #11. Thank you.

- Line 100-102 states "SFP-1 is the first protein demonstrated to be a transferred component in the seminal fluid into *C. elegans*", but this has already been published for TRY-5.

Reply: We've corrected the description. Thank you.

- Line 141-142 and Fig 1E and 1F, it seems strange that if the protein is transferred from the male to the hermaphrodite during mating it is seen in higher levels in the male after mating. Do the authors have any possible explanation for this?

Reply: First, as previously reported, male fertility declines with age. Therefore, we performed our experiments using Day 1 males, which are at a stage of optimal fertility (PMID: 22285759). Secondly, although it is transferred after mating, mating still contributes to the excessive accumulation of the protein in males. We hypothesize that mating induces SFP-1 storage in males to accommodate frequent mating events. Additionally, performing experiments the day after mating allows adequate time for protein accumulation. Furthermore, mating triggers the up-regulation of several genes in males, including increased expression of the *vit* gene. Notably, the repression of UNC-62, a major transcriptional regulator of the *vit* gene, is sufficient to rescue the

Revision Plan

shortened lifespan of males after mating (PMID: 28290982). In conclusion, we propose that the over-accumulation of SFP-1 in post-mating males may serve as preparation for subsequent mating or may have implications for the physiological function of males after mating.

- Line 149-150 "hermaphrodite brood size was enhanced ... by the *sfp-1* mutant male". This is wrong. Brood size is NOT enhanced compared to WT mating group. Higher than unmated, but significantly lower than mated with WT males.

Reply: We have revised the description to eliminate any ambiguity. The relevant text can be found on line 143-145. We sincerely apologize for the mistake. Thank you for your careful reading.

- Optional: Would like to see the transfer upon mating to N2 hermaphrodites. Don't expect to see a difference compared to the *glo-4* mutant, but signal seems bright enough right after mating to be able to observe without having to use a mutant hermaphrodite?

Reply: **We will perform experiments to observe the transfer of SFP-1 in N2 hermaphrodites after mating.** After mating SFP-1 fills the uterus of the hermaphrodite with a bright enough signal. However, we aim to monitor its transfer to the gut, where gut autofluorescence could interfere with our observations. To address this, we were performed using the *glo-4* mutant without gut autofluorescence.

- Don't understand the purpose of S2B. Sperm are not known to leave the reproductive tract under wild type conditions. What were the authors trying to test here?

Reply: We sincerely apologize for any confusion caused. We labelled sperm with Mitotracker and examined the hermaphrodite uterus after mating with stained males in order to confirm whether sperm and seminal proteins co-localized at a physical distance. We will adjust the language to reduce confusion or remove the data. Thank you.

- Line 238-240: Was this expected? Is this because they don't function through the intestine or because they don't affect lifespan at all?

Reply: First, we identified four proteins during our data screening process. None of these proteins have been previously reported; they are approximately 15 kDa in size, and all contain signal peptide sequences; however, only SFP-1 has an NTF2-like domain. Our findings indicated that intestinal overexpression of SFP-1 resulted in a shortened lifespan. We wondered whether other seminal fluid proteins could induce similar effects, but experimental results demonstrated that their overexpression in the intestine did not influence lifespan. We were unable to confirm

Revision Plan

whether these proteins do not exert their effects through the intestine, nor have we investigated their potential impact on lifespan post-mating, which may represent a valuable direction for future research.

- Line 272-274, Predictions suggest this. The authors don't provide empirical evidence of this interaction. Need to adjust language.

Reply: Thank you for your suggestion; after consideration, we will remove this part of the prediction.

- Longevity data with *daf-16* and *skn-1* are interesting, but could use more careful discussion of possible interpretations regarding independent pathways (*daf-16*) or downstream actors (*skn-1*). These are certainly challenging data to know how to interpret but clearer language regarding results and interpretation of those results is needed.

Reply: We also found this is a promising research interest. So we added more discussion about the *daf-16* and *skn-1* in line 412-419. Thank you for your patience and careful review. We do hope our revision will meet your demand.

Reviewer#2 (Significance (Required)):

Significance:

This study is of a previously uncharacterized protein, SFP-1, that was identified as a likely component of male seminal fluid in *C. elegans* in previous work. The work here confirms its identity as a seminal fluid protein and links it to the reduced longevity seen in mated hermaphrodites. Of interest to *C. elegans* researchers and researches interested in male fertility factors and molecular mechanisms of post-mating responses, especially as they relate to longevity. However, care needs to be taken in regard to controls, presentation of data, and interpretation of data before publication.

3. Description of the revisions that have already been incorporated in the transferred manuscript

In accordance with the minor concerns of Reviewer #1, the following changes have already been made to the manuscript.

1. We've added more detailed information in the figure legends.
2. We've added more discussion about comparative sex peptides with SFP-1. Related text on line 376-391.
3. We've polished the manuscripts and improved the language.

In accordance with the concerns of Reviewer #2, the following changes have already been made to the manuscript:

1. We've changed "sex interaction" to "sexual interaction".
2. We've changed "using the *sfp-1* gene its own promoter and performed confocal imaging" to "we generated a *P_{sfp-1}::SFP-1::YFP* translational reporter.....".
3. We've added more descriptions to ensure that readers understand our rationale for inhibiting the fat family. The relevant text is on line 352-354.
4. We've checked the grammar and the proper nouns in the text and polished the whole manuscript.
5. We've corrected the relative description in line 41. We have changed sex peptides to seminal fluid products in line 36.
6. We've added more description about male gonad in line 119-123.
7. We have revised the wrong description of brood size. The relevant text is on line 143-145.
8. We've added more discussion about the *daf-16* and *skn-1* in line 412-419.

4. Description of analyses that authors prefer not to carry out

Please include a point-by-point response explaining why some of the requested data or additional analyses might not be necessary or cannot be provided within the scope of a revision. This can be due to time or resource limitations or in case of disagreement about the necessity of such additional data given the scope of the study. Please leave empty if not applicable.

Dear Prof. Gong,

Thank you for submitting your manuscript for consideration by the EMBO Journal. I have now read your manuscript, the referee comments as well as your preliminary point-by-point response. I have also discussed the work with other members of the editorial team. Finally, I have consulted with an external advisor expert in the field. After all these considerations we have decided to invite you to submit a revised version of the manuscript, addressing the comments of both reviewers as proposed in your preliminary point-by-point response. We would also ask you take the additional comments by the advisor into consideration which you can find below. I am happy to discuss the revision in more detail via email or phone/videoconferencing should there be any additional questions.

I should add that it is EMBO Journal policy to allow only a single round of revision, and acceptance of your manuscript will therefore depend on the completeness of your responses in this revised version.

Thank you for the opportunity to consider your work for publication. I look forward to your revision.

Yours sincerely,

Cornelius Schneider

Cornelius Schneider, PhD
Editor
The EMBO Journal
c.schneider@embojournal.org

We realize that it is difficult to revise to a specific deadline. In the interest of protecting the conceptual advance provided by the work, we recommend a revision within 3 months (24th Feb 2025). Please discuss the revision progress ahead of this time with the editor if you require more time to complete the revisions. Use the link below to submit your revision:

advice:

I looked at the manuscript and the reviewers' comments and the authors' responses. I believe that the manuscript is worthy of further consideration by the EMBO J.

As both reviewers wrote, the manuscript identifies a protein in the male seminal fluid, SFP-1, and shows how it is transferred to the hermaphrodite to shorten hermaphrodite life span post-mating. The authors also provide a molecular mechanism for how SFP-1 shortens hermaphrodite longevity. They have implicated the Nrf transcription factor SKN-1 and decreased fat biosynthesis in this process. More importantly, the involvement of fatty acid desaturases in the effect of SFP-1 supports the hypothesis that is arising from the Booth et al work (Nat Aging 2022, vol. 2, 809-823), ie, male seminal fluid induces early death in hermaphrodites through fat loss, which should interest many in the field.

The manuscript does still need to be revised. For example, the authors suggest the intriguing involvement of extracellular vesicles in the delivery of SFP-1 to the hermaphrodites. I didn't carefully look through their images, but the authors' responses to Reviewer 2's comments about exophers suggest that they themselves cannot completely support exopher involvement. However, even if the vesicles are not exophers, this doesn't mean that the value of the manuscript is diminished. While at least one other male protein, like the major sperm protein MSP-1, has been shown to be transported through extracellular vesicles (Kosinski et al., Development 2005, vol 132, pp 3357-3369), SFP-1 transport does not have to be the same. SFP-1 might be transported via transcytosis and the vesicles the authors see are intracellular vesicles. The authors can simply qualify their conclusions here.

Some of the authors' plans for revision are valid, although some might need to be revisited. For example, I am not sure if the use of *Pges-1::sfp-1::yfp* will help discriminate between autofluorescence and *sfp-1* (response to reviewer 2 on page 8). It might be better to use an *sfp-1::mCherry* strain. However, it will all depend on the exposure times of the control images versus the *sfp-1::YFP* images.

Finally, the grammar of the manuscript also needs additional work. I would also suggest that they revise the second sentence of their abstract. They are clearly familiar with the Booth et al work (2022; they cited it as ref 11), which showed that male pheromones, sperm and seminal fluid shorten hermaphrodite lifespan through different mechanisms. The authors might want to place their work in the context of that earlier work, which they did in their introduction. I think it would be better if they already do this in the abstract.

Full Revision

Manuscript number: RC-2024-02650/EMBOJ-2024-119505-T

Corresponding author(s): Jianke Gong

1. General Statements [optional]

We would like to thank the referees for their precious time reviewing our manuscript and their valuable comments to improve this study. Our detailed responses to their concerns are listed below on a point-by-point basis.

This document reports the comments made by two Review Commons reviewers (in black font) and advice from an external advisor expert (in black font). In response to this feedback, we have performed additional experiments and revised our manuscript accordingly. Below, we address each comment, detailing the revisions we made (in blue font).

Reviewer #1 (Evidence, reproducibility and clarity (Required)):

Across the evolutionary spectrum, it has been observed that sexual interactions can significantly influence the physiology and somatic aging of individual animals. The underlying mechanisms of this phenomenon have only recently begun to be understood. In their manuscript titled "A male seminal fluid protein SFP-1 regulates hermaphrodite post-mating longevity and fat metabolism in *Caenorhabditis elegans*," Chen and Gong provide, for the first time, a comprehensive molecular understanding of how a male seminal fluid protein contributes to post-mating death in *C. elegans* hermaphrodites.

The authors identify SFP-1 as the key seminal fluid component by cleverly comparing two published genome-wide expression profiles of *C. elegans* males, significantly narrowing down the list of candidates. The role of SFP-1 in regulating post-mating lifespan in mated worms is convincingly verified through well-designed experiments using both a translational fusion reporter and a loss-of-function knockout strain. The authors then present a series of elegant experiments that comprehensively elucidate the functional mechanisms of SFP-1. On the male side, they show that SFP-1 is secreted into exophers as cargo from the male seminal vesicle, a process that requires the phospholipid scramblase ANOH-1 and ANOH-2. Once the seminal fluid is transferred into the hermaphrodites through mating, some exophers cross the gonad boundary and are taken up by the intestinal cells via endocytosis. The authors support their microscopic observations with convincing results from unmated hermaphrodites, where ectopic expression of SFP-1 in the intestine can largely mimic the physiological changes of mated hermaphrodites, such as post-mating shrinking and lifespan reduction. Further, the authors illustrate that the NTF2-like domain of SFP-1 is crucial for its function, and they show that SFP-1 interacts with the transcription factors SKN-1 and DAF-16, regulating lipid metabolism, particularly PUFA levels, in mated worms. Overall, this is a high-quality study that will be of considerable interest to researchers in this field. My detailed comments are listed below.

Major:

1. Lifespan comparisons

The manuscript includes a substantial number of lifespan assays, and the authors should be commended for their efforts. However, the data could be analyzed differently to provide more informative comparisons than those currently presented. For example, when investigating whether a specific gene is involved in SFP-1-dependent post-mating lifespan regulation, the authors performed quite a few lifespan assays with the *gonEx17* [*Pges-1::sfp-1*] strain. In the manuscript, the authors focus on comparing strains with SFP-1 overexpression, assuming that any factor increasing the lifespan of *gonEx17* indicates its involvement in SFP-1-dependent lifespan regulation. However, I believe it would be equally, if not more, informative to assess whether a factor can suppress the lifespan reduction caused by SFP-1 overexpression.

Reply: We sincerely thank the reviewer for their valuable suggestion. We agree that evaluating whether specific factors can suppress the lifespan reduction induced by SFP-1 overexpression would provide critical insights. In addition to our existing comparative analyses, we've added more comparative analyses between lifespan results. These additional analyses, included in the Results and Figures, enhance our understanding of the relationship between SFP-1 and the studied factors.

For instance, in Figure 7F, in addition to the comparisons made by the authors-(1) N2 vector RNAi vs. N2 *fat-2* RNAi and (2) *gonEx17* vector RNAi vs. *gonEx17* *fat-2* RNAi suggest also comparing (3) N2 vector RNAi vs. *gonEx17* vector RNAi and (4) N2 *fat-2* RNAi vs. *gonEx17* *fat-2* RNAi. The comparison (3) serves as a positive control, where the lifespan of *gonEx17* vector RNAi should be significantly shorter due to SFP-1 overexpression. (Of course, the *gonEx17* strain should be backcrossed to the relevant strain to eliminate genetic background differences, which is standard procedure in the field, and I trust the authors have done so.) Comparing N2 *fat-2* RNAi with *gonEx17* *fat-2* RNAi would clarify whether *fat-2* is involved in SFP-1-dependent lifespan regulation. If *fat-2* is required, there should be no significant difference in lifespans (as shown in Figure 7E). If *fat-2* were not required, a similar lifespan reduction due to *gonEx17*, as seen in the control comparison (3), would be expected.

Reply: Thank you for your valuable advice. We have added comparison groups for (3) and (4). In comparison (3), the lifespan of the *gonEx17* vector RNAi is significantly shortened due to SFP-1 overexpression. In comparison (4), there is no significant difference between the lifespans

of the two groups, indicating that *fat-2* is involved and required for SFP-1-dependent lifespan regulation.

We also extended this analysis to *fat-6* and *fat-7*. In comparison (3), the lifespan of the *gonEx17* vector RNAi is significantly shortened due to SFP-1 overexpression. In comparison (4), there is no significant difference between the lifespans of the two groups. RNAi targeting *fat-6* or *fat-7* shortens lifespan in the N2 background (as seen in comparison 1), but no lifespan extension was observed in *gonEx17* (comparison 2). This suggests that while *fat-6* and *fat-7* may have general effects on longevity, they do not specifically modulate SFP-1-dependent lifespan regulation. Based on these findings, we propose that if inhibition of a gene extends lifespan in *gonEx17* and eliminates the shortened-lifespan difference between *gonEx17* and N2, then that gene is probably involved in SFP-1-mediated lifespan regulation. This standard helps identify genes that specifically required for SFP-1-dependent lifespan regulation from those with general effects on longevity.

Based on the proposed standard, we suggest that *fat-1*, *fat-2*, *fat-3*, *fat-4*, and *fat-5* are all involved in SFP-1-mediated lifespan regulation, potentially due to redundancy in the function of the fat enzymes within the synthetic pathway.

In addition, we've added these additional comparisons to all relevant RNAi lifespan experiments to further clarify gene involvement in SFP-1-dependent regulation. These relevant figure panels which have been modified are as follows: Figures 5E, EV4C-F, EV4L-Q, 7E-H, and EV5B-F.

A different pattern emerges in Figure S4C-D, where these two ways of comparison yield opposite conclusions. The authors suggest that RNAi of *sek-1* and *pmk-1* increases the lifespan of SFP-1-overexpressing worms, implicating these genes in SFP-1-dependent lifespan regulation. However, when comparing the lifespans differently, *gonEx17* still induces lifespan reduction in the presence of *sek-1* or *pmk-1* RNAi, though to a lesser extent than with vector RNAi. This result contrasts with the strong evidence presented for *skn-1* (as shown in Figures 5C and 5E, where SFP-1-mediated lifespan regulation is entirely dependent on *skn-1*), suggesting that *sek-1* and *pmk-1* may play a much lesser role. The authors should conduct additional lifespan comparisons and revise their model accordingly. Below are the relevant figure panels: Figures 5E, S4C-D, S5, D-I, 7E-F, and S6A-E.

Reply: We apologize for not clearly explaining the regulatory relationship between SEK-1, PMK-1, and SKN-1. As the reviewer pointed out, while the SEK-1-PMK-1 pathway is an up-

stream regulator of SKN-1, it is not the exclusive one. This explains why RNAi targeting *sek-1* or *pmk-1* only partially reduces SKN-1 activity, resulting in a lesser extension of lifespan in *gonEx17* compared to vector RNAi. In contrast, direct loss of *skn-1* completely inhibits SKN-1 activity, leading to lifespan extension. We have revised the manuscript to clarify this relationship better and updated our model to suggest that while SEK-1 and PMK-1 are involved in modulating SKN-1 activity, SKN-1 is the primary downstream effector of SFP-1 in lifespan regulation. Thank you for your suggestion, we've conducted additional lifespan comparisons and revised the model.

2. How ectopic expression of SFP-1 imitates the post-mating phenotypes

The authors demonstrate that ectopic expression of SFP-1 in the intestine of unmated hermaphrodites can lead to a decrease in body size over time, similar to that observed in mated hermaphrodites (Figure S1B). They suggest that this intestinal SFP-1 overexpression alters germline activity, as evidenced by the increased brood size (Figure S1A), which they propose is responsible for the shrinking phenotype. However, in Figure 1A, the brood size of *gonEx17* is still significantly lower than that of mated hermaphrodites. This discrepancy makes the current data less convincing. To better understand whether SFP-1 overexpression in the intestine requires an intact germline to mediate lifespan decrease in self-fertilized hermaphrodites, it would be helpful to assess these phenotypes in germlineless worms such as *glp-1(e2141)* with SFP-1 overexpression.

Reply: We sincerely thank the reviewer for raising this important point. Our observations indicate that the brood size of *gonEx17* hermaphrodites increases brood size compared to wild-type, it remains significantly lower than that of mated hermaphrodites. This difference may come from the contribution of male sperm, which enhances the brood size in mated hermaphrodites, whereas *gonEx17* relies only on self-sperm. We propose that the increase in brood size observed in *gonEx17* may be attributed to germline activity. Specifically, an increase in meiotic zones promotes a greater production of oocytes in post-mating hermaphrodites (PMID: 24356112).

To address whether SFP-1-mediated lifespan reduction requires an intact germline, we've replaced the wild-type background of *gonEx17* with *glp-1(e2141)* background and performed lifespan experiments. The *glp-1(e2141)* mutant is a temperature-sensitive strain that completely ablates gonadal development at 25 °C. Following established protocols for lifespan experiments

about *glp-1(e2141)* mutant (PMID: 37398385), we set up two experimental conditions: (1) maintaining worms at 25°C for the entire lifespan, and (2) exposing worms to 25°C from the L4 stage to Day 1 of adulthood, followed by transfer to 20°C for the lifespan experiment. These results from both conditions are shown in the figure below, demonstrate that intestinal SFP-1 overexpression reduces lifespan in germlineless worms compared to *glp-1(e2141)* controls, though there is a prolonged lifespan than wild-type *gonEx17*. These findings collectively suggest that intestinal overexpression of SFP-1 does not require an intact germline to mediate the reduction in lifespan of self-fertilized hermaphrodites. Relevant data are presented in Figure EV3A, and the corresponding text (line 246-252) has been updated accordingly. Your insightful suggestion has significantly strengthened our study, and we deeply appreciate your advice.

Fig.1 for reviewers. Intestinal overexpression of SFP-1 does not require an intact germ line to mediate the shortened lifespan.

(A) Survival curves of *glp-1(e2141)* worms, *gonEx17* worms, wild-type N2 worms and *gonEx17;glp-1(e2141)* worms when grown and aged at 25°C. N2: 13.17 ± 0.46 days, n = 105 worms; *gonEx17 [Pges-1::sfp-1]*: 9.25 ± 0.52 , n = 71 worms; *glp-1(e2141)*: 17.33 ± 1.30 , n = 55 worms; *gonEx17 [Pges-1::sfp-1];glp-1(e2141)*: 12.10 ± 0.63 , n = 57 worms. ** $P < 0.01$, *** $P < 0.001$, **** $P < 0.0001$.

(B) Survival curves of *glp-1(e2141)* worms, *gonEx17* worms, wild-type N2 worms and *gonEx17;glp-1(e2141)* worms when grown at 25°C then aged at 20°C. N2: 18.48 ± 0.84 days, n = 78 worms; *gonEx17 [Pges-1::sfp-1]*: 15.03 ± 0.66 , n = 61 worms; *glp-1(e2141)*: 25.07 ± 1.3 , n = 88 worms; *gonEx17 [Pges-1::sfp-1];glp-1(e2141)*: 19.56 ± 0.74 , n = 82 worms. **** $P < 0.0001$.

Minor:

Full Revision

1. The figure legends should include more detailed information about the results. Currently, some lifespan data are missing in the figure legends for the following figures: Figure 4A, 4C, S3A, 5E,

S4A-D, S5D-I, 7E-F, S6A-E.

Reply: We've added the figure legends to include detailed lifespan data for Figure 4A, 4C, EV3C, 5E, EV4A-F, EV4L-Q, 7E-F, and EV5B-F. These additions can be found in the revised manuscript (lines 613-620, 623-629, 649-653, 688-692, 703-730, 740-750, 803-814, and 832-851).

2. There are quite a few discrepancies between the supplementary lifespan results table and the figure panel labeling in the manuscript.

Reply: We've screened the whole manuscript and corrected the non-corresponding labeling in the text. Thank you.

3. To broaden the manuscript's appeal, the authors should consider including a discussion comparing SFP-1 with the sex peptide proteins previously reported to regulate post-mating physiology and lifespan in *Drosophila*. For instance, do the fly sex peptide proteins also contain an NTF-2 domain?

Reply: We've added more discussion about comparative sex peptides with SFP-1 in *Drosophila*, including their structural and functional similarities. This discussion can be found in the revised manuscript (lines 425-435).

4. The language should be refined to enhance the clarity and comprehensibility of the manuscript.

Reply: We've polished the manuscripts and improved the language. Again, we appreciate your great advice on our study and hope it will meet your command.

Reviewer #1 (Significance (Required)):

Sexual interactions have been observed to significantly impact the physiology and somatic aging of individual animals across the evolutionary spectrum. The mechanisms underlying this phenomenon have only recently begun to be understood. In this manuscript, Chen and Gong offer

Full Revision

the first comprehensive molecular insight into how a male seminal fluid protein contributes to post-mating death in *C. elegans* hermaphrodites. This high-quality study will be of considerable interest to researchers in the fields of *C. elegans* biology, sexual interactions, lipid metabolism, and aging. The manuscript's appeal could be further enhanced by including additional statistical comparisons and a more in-depth discussion on the evolutionary conservation of the seminal fluid protein identified by the authors.

Reviewer #2 (Evidence, reproducibility and clarity (Required)):

Summary:

In this manuscript the authors present analysis of a seminal fluid protein, SFP-1, in *C. elegans*. The authors present relatively convincing data that this protein is expressed in males, transferred to hermaphrodites upon mating, and influences hermaphrodite longevity post-mating.

Specifically, hermaphrodites mated to males lacking *sfp-1* live longer than hermaphrodites mated to wild-type males with SFP-1. The authors then seek to identify the tissue from which this protein exerts its effects and connect it to others pathways (FOXO/DAF-16, SKN-1, PUFA biogenesis). They do show that ectopic expression in the hermaphrodite gut has an effect, but evidence that this is physiological or that the protein makes it to the gut after mating is not substantiated by the data presented. The data suggesting the effect occurs through *skn-1* is interesting but could use more careful discussion, while the data seems to suggest that the effect is independent of *daf-16*, however it is not clear that this is the conclusion of the authors. This manuscript represents a large body of work and contains some very interesting results and observations. However, in its current state it is not yet ready for publication. It requires extensive editing for grammar and language for the sake of clarity and accuracy. Other concerns regarding images, controls, and interpretation of results also exist.

Major comments:

- Fig 1D and 1E need higher magnification images for the reader to accurately assess the data. While it may be an artifact of the pdf format provided, the resolution of all fluorescent images was also too low for accurate assessment.

Reply: We sincerely apologize for the confusion. We have added higher magnification images (120X) in Figure 1D, while the images of mated males at 60X magnification are displayed below. Additionally, we have re-uploaded all original TIFF versions of the figures to ensure that the resolution is sufficient for readers to accurately evaluate the data.

Fig.2 for reviewers. *sfp-1::yfp* expression increased in mated males.

(A) The images of *sfp-1::yfp* males at 60X magnification. For unmated males, images were captured on Day 2 of adulthood. For mated males, mating on Day 1, and images were captured on Day 2 of adulthood. Representative images are shown (Scale bars: 15 μ m)

- Fig 1D vs unmated in 1E, why does E show fluorescence in the pharynx but D doesn't? Aren't these images of the same genotype? What is the interpretation that the pharynx expression also appears brighter after mating in 1E?

Reply: We apologize for the discrepancies observed in the fluorescence within the pharynx. Firstly, since SFP-1 is exclusively expressed in males, we added a pharynx fluorescence as a marker during microinjection. Our primary focus was on the fluorescence in the seminal vesicle, leading us to ignore the effects of pharynx fluorescence. Importantly, mating does not affect pharynx fluorescence. We think this variability is due to the microscope focus on the seminal vesicle when the image was captured; however, the markers expressed in the pharynx may be out of focus. To address this, we've repeated the experiment using an endogenously tagged line (*sfp-1::mNeonGreen*), and the results confirmed that mating does not affect pharynx fluorescence in males, as shown in Figure 1H. We have standardized all figures to ensure consistent image quality across both groups and have provided detailed explanations in the figure legends to avoid any further confusion.

- In Figure 2A the background for the N2 control is black while the background for the yfp image is green. This strongly indicates that the levels or image exposure are not comparable between the images and therefore does not allow one to make any conclusions about what may be

autofluorescence and what may real signal in the yfp images. For this reason, the localization the authors claim to see in the intestine is not convincing and not supported by the data as shown.

Reply: We thank the reviewer for this important observation. To address the concern about image comparability, we have standardized the fluorescence images in Figure 2A using identical brightness and contrast settings. Additionally, we conducted complementary experiments to validate SFP-1 localization by crossing *Psfp-1::sfp-1::GFP (1-10)* males with *Pges-1::GFP (11x7);glo-4* hermaphrodites. The observation of fluorescent signals in the intestines of mated hermaphrodites (Figure 2B, Figure EV2B) provides confirmation of SFP-1 spreading into intestinal cells. We have updated the relevant text in line 174-183 of revised manuscript.

- Aren't Fig 4D and S3B (lower) of the same genotype? But the "nuclear" localization that is supposedly in 4D isn't observable in S3B.

Reply: Thank you for your careful observation. We appreciate your attention to the details in Figures 4D and S3B. Regarding your question, we would like to clarify that while both figures show the same genotype, they serve different experimental purposes. In Figure 4D, we specifically focused on demonstrating the nuclear localization of SFP-1 in intestinal cells. However, Figure S3B was designed to examine potential signal leakage to other tissues, not to investigate nuclear localization, which is why nuclear staining was not performed. We understand how this difference in experimental focus could lead to confusion, and we have revised the manuscript to make this distinction clearer. Your feedback has been valuable in improving the clarity of our presentation.

- Due to gut autofluorescence it's important to see a control image for 4D that shows worms that don't express *sfp-1::yfp* under the same imaging conditions (exposure, brightness levels, etc).

Reply: Thank you for your suggestion. To address potential concerns regarding intestinal autofluorescence, we've performed the experiment using the progeny from the *Pges-1::sfp-1::yfp* strain which has no fluoresce as a control, which will help us effectively rule out the influence of intestinal autofluorescence. We've performed experiments under both low-exposure (left) and high-exposure (right) conditions to examine the effect of autofluorescence. The experimental findings indicate that the fluorescence observed around the nucleus is not intestinal autofluorescence. These observations have been displayed below.

Fig.3 for reviewers. *Pges-1::sfp-1::yfp* worms and control worms in the different exposure times.

(A) Animals under low exposure (left), intestinal autofluorescence could not be observed under these conditions. (Scale bars: 5 μ m)

(B) Animals under high exposure (right), intestinal autofluorescence could be observed under these conditions. (Scale bars: 5 μ m)

- Figure 5B at minimum needs arrows in the middle column indicating where the nucleus is. Imaging with a nuclear signal would be more convincing, but at least need to be able to point out where the nucleus is. Otherwise it's just an overall loss of signal and not a change in localization.

Reply: Thank you for your suggestion. We have added arrows to Figure 5B to clearly indicate the location of the nucleus, and we apologize for any confusion this may have caused.

- Figure 5F again seems to be observing overall differences in protein levels but does not convincingly show nuclear localization.

Reply: Thank you for your suggestion. To investigate changes in nuclear localization, we've examined SKN-1::GFP localization in both N2 and *ges-1::sfp-1* animals. Our results demonstrate that SKN-1::GFP localizes to the nucleus of intestinal cells in animals overexpressing SFP-1, as shown in Figure 5D. The corresponding text has been updated to reflect these findings.

- Figure 6A needs an unmated control.

Reply: Thank you for your suggestion, we have added unmated hermaphrodites as a control and performed the experiment, as shown in Figure 6A, our conclusions have not changed.

- Figure 6I, the images are not clear enough to convincingly show germline vs soma (vs gut on top of germline based on the orientation of the worm on the slide). The conclusion about asdf is not supported based on these data alone.

Reply: We sincerely thank you for your valuable feedback. We acknowledge that the original images may not sufficiently distinguish between germline and somatic fat distribution to convincingly support the conclusion about Asdf.

To address this concern, we have replaced the images with more representative examples that better illustrate the differential fat distribution between somatic and germline cells. Additionally, we have provided further experimental evidence to support the conclusion about Asdf.

The *vit* family plays a critical role in the transfer stored lipids from the intestine to the germline. We specifically overexpressed SFP-1 in the intestinal cells of *vit-1* and *vit-2* mutants, found that the Asdf phenotype observed in the intestinal SFP-1 overexpressing animals was suppressed in both *vit-1* and *vit-2* mutant backgrounds, potentially by preventing lipid depletion in the intestine. (Figure EV4H-K, line 369-374 in the revised manuscript).

Furthermore, we observed that the Asdf phenotype is absent in animals overexpressing SFP-1 when SKN-1 is knocked down, indicating the functional role between SFP-1 and SKN-1 in the regulation of lipid metabolism.

- Data in final results section and Figure 7 does not make much sense and its physiological meaning is not clear as it's based entirely on over-expression in the intestine. If PUFAs are lowered in response to SFP-1, and SFP-1 expression reduces lifespan then why does further reducing PUFAs with the fat mutants extend lifespan?

Reply: Thank you for pointing out this problem. We acknowledge that the data in Figure 7, based on intestinal overexpression of SFP-1, requires further clarification. Here, we provide additional experiments and rationale to address the reviewer's concerns.

Our hypothesis is based on findings that genes associated with fatty acid biosynthesis exhibit significant expression changes in mated hermaphrodites (PMID: 34179018). Additionally, *fat-2* mutants that contain excess endogenous oleic acid are protected from lifespan loss after mating, suggesting that lipid metabolism may play a critical role in SFP-1-mediated shortened lifespan.

While SFP-1 overexpression reduces PUFA levels and shortens lifespan, the inhibition of *fat-1* and *fat-2* extends lifespan by preventing PUFA depletion and promoting the accumulation of oleic acid. Importantly, SFP-1 fails to extend lifespan in the absence of *fat-6* and *fat-7*, may due to

a further deficiency in oleic acid. Conversely, *fat-1*, *fat-2*, *fat-3*, *fat-4*, and *fat-5* extends lifespan, as these genes prevent PUFA depletion and support lipid homeostasis.

To further investigate the role of SFP-1 in post-mating lipid metabolism, we've examined changes in fatty acid content in unmated hermaphrodites and hermaphrodites mated with different males (N2, *sfp-1* mutants). We found that hermaphrodites mated with *sfp-1* mutant males exhibited significantly higher levels of PUFAs compared to those mated with N2 males, indicating that SFP-1 is involved in the post-mating depletion of PUFAs. These results are now included in Figure 7B, with detailed descriptions in lines 389–394 of the revised manuscript.

Additionally, we've performed lifespan experiments involving hermaphrodites mated with different males (N2, *sfp::yfp*) under conditions of *fat-1*, *fat-2* inhibition. Lifespan analysis indicated that *fat-1* and *fat-2* RNAi were both resistant to mating-induced lifespan reduction, even when hermaphrodites were mated with *sfp::yfp* males. These data are shown in Figure 7E and 7F, with the relevant text updated in line 406-408.

These experiments confirm that SFP-1-mediated PUFA depletion and its effects on lifespan are modulated by *fat* gene. We have also revised the text to ensure that the rationale for inhibiting the fat family. We deeply appreciate your suggestions, which are invaluable in enhancing the quality of our work.

- Supplemental Figure 1C needs a control of hermaphrodites on unconditioned plates. There is clearly variability from experiment to experiment in regards to longevity. This control is needed to show that the expected reduction on conditioned plates was indeed observed.

Reply: Thank you for your suggestion. We have revised the lifespan results figure, as illustrated in Fig EV1G, to include data for N2 without any treatment (with no males crawling through the plates) as a control group. Our conclusions remain unchanged.

- Bafilomycin A1 (*bafA1*) does not inhibit endocytosis. Inhibiting the maturation of an endosome to lysosome is not the same as inhibiting endocytosis. Need to remove or reinterpret these results.

Reply: We thank the reviewer for raising this important point. We agree that *bafA1* primarily inhibits endosome acidification rather than directly blocking the initiation of endocytosis, though it does affect downstream endocytic processes. Previous study has demonstrated that Bafilomycin-A1 (*bafA1*) can disrupt different aspects of endocytosis (PMID: 22902558). Specifically, *bafA1* prevents acidification-dependent vesicle turnover, which is required for endosome maturation.

ration (PMID: 16415858). In response to this valuable feedback, we have carefully revised the relevant text (in line 187-190) to provide a more precise description of bafA1's function and its specific effects on endocytic processes.

- Line 209-211 and Figure 3A, how were exophers determined? What evidence was used to show that any observed puncta are exophers and not intracellular vesicles? How were puncta/vesicles defined for quantification? Also important to note that these "vesicular" structures are not observed when the protein is in the hermaphrodite reproductive tract after mating.

Reply: We sincerely thank the reviewer for their insightful comments and constructive questions regarding the identification and characterization of the observed vesicular structures. To address the concerns:

1. Identification of exophers/secretory vesicles: The vesicular structures observed in *sfp-1::yfp* males were identified based on their morphological similarity to secretory vesicles described in WormAtlas (PMID: 478167), which notes that small secretory cells in the seminal vesicle exhibit a granular appearance with luminal blebs under DIC optics. These structures are consistent with secretory vesicles reported in studies of TRY-5 (PMID: 22125495). While definitive markers for extracellular vesicles (EVs) or exophers are lacking in *C. elegans*, the secretory characteristics of these structures are supported by their localization and release from the male seminal vesicle, making it unlikely that they represent intracellular vesicles.

2. Quantification standards: For quantification, we focused on recognizable globular structures with distinct morphology under DIC microscopy, consistent with secretory vesicles. We have revised the manuscript to describe these structures as "secretory vesicles" to reflect the current evidence. Future studies will aim to experimentally validate whether these vesicles are EVs or exophers using additional markers or functional assays.

3. Absence in mated hermaphrodites: We agree with the reviewer that these vesicular structures are not observed in the hermaphrodite reproductive tract after mating. This is consistent with observations of TRY-5, which forms secretory vesicles in males but diffuses after entering the hermaphrodite uterus (PMID: 22125495). We hypothesize that the vesicles may disperse or undergo lysis after entering the uterus.

We appreciate the reviewer's emphasis on caution in interpreting these findings and have adjusted the text accordingly (line 200-210 in the revised manuscript). Future work will focus on clarifying these secretory vesicles, including their potential role in intercellular communication and

post-mating effects. Thank you again for your valuable feedback, which has strengthened our study and its interpretation.

- *anoh-1* and *anoh-2* mutations were in males, results can not be used to support a model of what happens to the protein once it is transferred to the hermaphrodite.

Reply: We sincerely thank the reviewer for their valuable and insightful comments. In addressing the concern that the *anoh-1* and *anoh-2* mutations were examined exclusively in males and therefore cannot directly influence the post-mating phenotype in hermaphrodites, we fully acknowledge this limitation. Here, we provide further clarification and future research direction to address this important question: Our observations that *anoh-1* and *anoh-2* mutant males exhibit a significant reduction in secretory vesicles, attributable to impaired membrane fusion and diminished SFP-1 protein levels in the seminal vesicle, suggest a potential decrease in the efficiency of SFP-1 transferred during mating. This implies that ANOH-1 and ANOH-2 functionality in males may influence physiology in mated hermaphrodites.

To address this concern and directly investigate the role of membrane fusion in mating and its impact on hermaphrodite lifespan, we propose to overexpress ANOH-1 and ANOH-2 in *sfp-1* mutant background. This approach would allow us to isolate the effects of membrane fusion on secretory vesicle transfer efficiency, independent of SFP-1 cargo, allowing us to assess the direct impact on mated hermaphrodite lifespan.

However, due to technical challenges in generating a fluorescently tagged male for these experiments, we were unable to perform this analysis in the current study. We will work to overcome this technical difficulty and plan to follow up this experiment in future work. Thank you again for your valuable feedback, which has helped us refine our interpretation and future directions.

- Line 230-231: This result is very confusing. Sperm are the limiting factor for hermaphrodite self-fertility. These data would suggest that SFP-1 in the hermaphrodite intestine led to increased spermatogenesis/more sperm being made by the hermaphrodite in the L4 stage? There are no previous reports of long-lived strains having increased self-fertility. Were the brood sizes for N2 unmated and *gonEx17* unmated measured in the same assay/at the same time?

Reply: We apologize for any confusion. To clarify, the brood size of unmated N2 and unmated *gonEx17* were indeed measured in the same assay and under identical conditions. We generated three independent transgenic lines via microinjection of *ges-1::sfp-1*, and brood size was as-

essed in all three lines. Consistent with our findings, all three transgenic lines exhibited a significant increase in brood size compared to controls.

Fig.4 for reviewers. Total brood size of N2 worms and *Pges-1::sfp-1* transgenic overexpressing lines.

(A) Brood size of *Pges-1::sfp-1* transgenic overexpressing lines in comparison to control nematodes. ** $P < 0.01$, *** $P < 0.001$, $n > 15$. P values were calculated by one-way ANOVA, Bonferroni's multiple comparisons test.

Compared to N2, *gonEX17* exhibited increased brood size and shortened lifespan, rather than increased self-fertility in long-lived strains. We apologize for the misunderstanding and have revised the text for clarity and accuracy. While we hypothesize that SFP-1 induces germline hyperactivation, further experiments are needed to confirm this mechanism. Previous studies have demonstrated that mating causes activity in the germline of hermaphrodites and found that mated worms have significantly fewer nuclei in the mitotic region. (PMID: 24356112). Notably, we found that intestinal SFP-1 overexpression reduces lifespan independently of an intact germline, as demonstrated in a *glp-1* mutant background. These results are now included in Figure EV3A and the corresponding description in line 246-252 of the revised manuscript. Thank you again for pointing this out.

- Line 235-236 and Fig S3B: While possible, this result is not consistent with other reported data for other proteins. Did the authors look in the coelomocytes where it would be expected to be found if it was secreted from the muscle? Did they do a control with ss:GFP (GFP with a signal sequence) which is known to be secreted out of the muscle for comparison?

Reply: Thank you for your insightful suggestion. We investigated coelomocytes in *Pmyo-3::sfp-1::yfp* animals and observed bright fluorescent signals, consistent with previous studies showing GFP secretion into coelomocytes when attached to a signal sequence and expressed in body wall muscles (PMID: 11560892, 15070744). In contrast, *Pmyo-3::yfp* controls showed only weak fluorescence, suggesting that the signal peptide of SFP-1 facilitates endocytosis into coelomocytes, while *myo-3* promoter itself may contribute to this phenotype. These results are now included in Figure EV3B, with detailed descriptions in lines 253–258 of the revised manuscript. Importantly, previous studies indicate that endocytosis by coelomocytes is not essential for growth or survival of *C. elegans* under normal laboratory conditions (PMID: 11560892).

Fig.5 for reviewers. Expression of *Pmyo-3::sfp-1::yfp* and *Pmyo-3::yfp*.

(A) Low-magnification images showing whole worms. High-magnification images showing individual coelomocytes. Fluorescent signals in coelomocytes of *Pmyo-3::sfp-1::yfp* are very bright. (Scale bars: 50 μ m, 15 μ m).

(B) Low-magnification images showing whole worms. High-magnification images showing individual coelomocytes. Fluorescent signals in coelomocytes of *Pmyo-3::yfp* are very weak. (Scale bars: 50 μ m, 15 μ m).

• Grammar and coherence of writing need to be addressed. Examples include, but are not limited to:

"sex interaction" - Do the authors mean interaction between the sexes?

"cross over the somatic gonad uterus"

"using the *sfp-1* gene its own promoter and performed confocal imaging."

Full Revision

Reply: We appreciate the reviewer's attention to the clarity and grammar of our writing. We have carefully revised the manuscript to address issues such as:

Replaced "sex interaction" with "sexual interaction."

Removed the phrase "cross over the somatic gonad uterus."

Rewrote "using the *sfp-1* gene its own promoter and performed confocal imaging" to "we generated a *sfp-1::yfp* translational reporter to assess the localization and function of SFP-1."

We've checked the grammar and the proper nouns in the text and polished the whole manuscript. We sincerely appreciate your feedback, which has significantly enhanced the quality of our manuscript.

Minor comments:

- Worth noting that over-expression (all experiments with SFP-1::GFP) could impact whether the protein ends up in vesicles and the type of localization seen in both the male and hermaphrodite. While not necessary to repeat with an endogenously tagged line, this caveat to the results should be acknowledged.

Reply: We sincerely thank you for raising this important point. To address the potential impact of overexpression on vesicle localization, we have generated an endogenously tagged *sfp-1::mNeonGreen* line using CRISPR/Cas9. This strain was used to repeat critical experiments, confirming our initial findings. These results are now included in Figure 1G-I, with detailed descriptions in lines 133-137 of the revised manuscript. Thank you very much for your suggestion to make our results more credible.

- Introduction, Line 41: Either this is the wrong reference or "sex peptides" should be "seminal fluid proteins". "Sex peptide" is a specific SFP in *Drosophila* and reference #7 is not about sex peptide.

Reply: Thank you for your careful review. This term has been revised to "seminal fluid products" to accurately reflect the content of reference #7 (now reference #6 in the revised manuscript).

- For Image analysis of SFP-1::GFP, how was the region of the seminal vesicle determined/defined?

Reply: Thank you for your question. The seminal vesicle was identified based on its anatomical structure, which is characterized by somatic gonadal cells surrounding the spermatids. To confirm SFP-1 localization, we used MitoTracker to label spermatids in *sfp-1::yfp* males. We ob-

served that SFP-1::YFP fluorescence specifically surrounded the MitoTracker-labeled spermatids, confirming its expression in the seminal vesicle. A schematic diagram from WormAtlas has been included for reference and additional experimental results are shown in Figure EV1A, with detailed descriptions in lines 118-126 of the revised manuscript. We are grateful for your insightful comments, which have significantly improved the clarity of our work.

Fig.6 for reviewers. Adult male germ line organization.

(Male Reproductive System - Germ Line)

(A) Nomarski DIC picture of an adult male tail region featuring the parts of the reproductive tract, lateral view. (DG) Distal gonad; (PG) proximal gonad; (DTC) distal tip cell.

- Fig 6C it appears that the pattern of the staining changes but where there is staining the intensity doesn't appear to be different from N2 control.

Reply: Thanks for pointing this out. It is important to note that our experiments were conducted using Day 5 (D5) adult worms, which exhibit a natural age-dependent reduction in fat storage even in the N2 wild-type controls. Additionally, while Oil Red O staining primarily stains neutral lipids, it also stains the lipid-rich egg in the worms, so despite the obvious fat loss that occurs with intestinal overexpression of SFP-1, the eggs in the body are still able to be stained with Oil Red. We've reselected representative images in Figure 6C to accurately reflect the observed phenotypes,

thank you.

- It would be helpful if the authors were consistent in the color choices for graphs and figures. For example, control (N2 x N2) is black in Fig 1G but red in Fig 3C.

Reply: We've unified the color choices across all figures to ensure consistency, with the control group (N2 x N2) represented in the black.

- Line 55-56 in Introduction needs a citation.

Reply: Thank you for catching this oversight. We have added the relevant citations (reference #9 and #11) to support the statement.

- Line 100-102 states "SFP-1 is the first protein demonstrated to be a transferred component in the seminal fluid into *C. elegans*", but this has already been published for TRY-5.

Reply: We sincerely apologize for the inaccuracy. As you correctly noted, TRY-5 was previously identified as the first transferred seminal fluid protein in *C. elegans*. We have removed the inaccurate statement and revised the text accordingly, thank you.

- Line 141-142 and Fig 1E and 1F, it seems strange that if the protein is transferred from the male to the hermaphrodite during mating it is seen in higher levels in the male after mating. Do the authors have any possible explanation for this?

Reply: Thank you for raising this interesting point. While SFP-1 is transferred to hermaphrodites during mating, our data suggest that mating also triggers its over-accumulation in males. We hypothesize that this protein over-accumulation may prepare for complementing the transferred protein to prepare for potential subsequent mating events and support post-mating physiological functions. Supporting this, mating upregulates several genes in males, including the *vit* gene, the repression of UNC-62, a major transcriptional regulator of the *vit* gene, is sufficient to rescue the shortened lifespan of mated males (PMID: 28290982). Additionally, our experiments were conducted with Day 1 males, which exhibit optimal fertility (PMID: 22285759), and protein levels were assessed the day after mating to allow sufficient time for protein accumulation.

- Line 149-150 "hermaphrodite brood size was enhanced ... by the *sfp-1* mutant male". This is wrong. Brood size is NOT enhanced compared to WT mating group. Higher than unmated, but significantly lower than mated with WT males.

Reply: We sincerely apologize for the inaccuracy in our initial description. The brood size of hermaphrodites mated with *sfp-1* mutant males is indeed higher than unmated controls but significantly lower than those mated with WT males. We have carefully revised the text (lines 148-150 in the revised manuscript) to accurately reflect these comparative results. Thank you for your careful reading.

- Optional: Would like to see the transfer upon mating to N2 hermaphrodites. Don't expect to see a difference compared to the *glo-4* mutant, but signal seems bright enough right after mating to be able to observe without having to use a mutant hermaphrodite?

Reply: We have performed experiments to observe the transfer of SFP-1 in N2 hermaphrodites after mating. The results are located below. After mating SFP-1 fills the uterus of the hermaphrodite with a bright enough signal. While the signal in the uterus is bright post-mating, gut autofluorescence in N2 hermaphrodites complicates the observation of SFP-1 transfer to the gut. To address this, we used *glo-4* mutants, which lack gut autofluorescence, to clearly monitor SFP-1 localization.

Fig.7 for reviewers. Male-to-hermaphrodite transfer experiment where males were mated with N2 hermaphrodites.

(A) In imaging of the N2 hermaphrodites after copulation revealed the presence of male derived *sfp-1::yfp* in the uterus, however, gut autofluorescence interfered with our observations to detect other fluorescent signals in the intestine. In imaging of the N2 hermaphrodites after mating with N2, no fluorescence signal was observed in the uterus, however, gut autofluorescence was observed. (Scale bars: 15 μ m)

- Don't understand the purpose of S2B. Sperm are not known to leave the reproductive tract under wild type conditions. What were the authors trying to test here?

Reply: We sincerely apologize for the confusion. The purpose of S2B (Figure EV1B in the revised manuscript) was to confirm whether sperm and seminal proteins co-localize within the reproductive tract after mating. We used Mitotracker to label sperm and examined their distribution in the hermaphrodite uterus post-mating. We have clarified this explanation in the revised manuscript (line 126-129).

- Line 238-240: Was this expected? Is this because they don't function through the intestine or because they don't affect lifespan at all?

Full Revision

Reply: We sincerely thank the reviewer for raising this important question. In our screening, we identified four seminal fluid proteins (SFPs), all around 15 kDa with signal peptide sequences, yet none had been previously reported, only SFP-1 has an NTF2-like domain. Intestinal overexpression of SFP-1 shortened lifespan, but other SFPs did not exhibit similar effects, suggesting the phenotype may be specific to SFP-1 and its functional domain.

Based on current data, we cannot determine whether these proteins function through other tissues or have no impact on lifespan. Additionally, their potential effects in post-mating physiology remain unexplored and represent an important future research direction. We have added a discussion of these possibilities in the revised manuscript (lines 262-265). Thank you for your valuable feedback.

- Line 272-274, Predictions suggest this. The authors don't provide empirical evidence of this interaction. Need to adjust language.

Reply: Thank you for your suggestion. Upon careful consideration, we have removed the speculative prediction regarding protein interactions, as empirical evidence is lacking. Thank you for pointing this out.

- Longevity data with *daf-16* and *skn-1* are interesting, but could use more careful discussion of possible interpretations regarding independent pathways (*daf-16*) or downstream actors (*skn-1*). These are certainly challenging data to know how to interpret but clearer language regarding results and interpretation of those results is needed.

Reply: We appreciate this suggestion and have expanded the discussion on *daf-16* and *skn-1* in the revised manuscript (lines 452-457). We now provide a more detailed interpretation of their roles in SFP-1-mediated lifespan regulation. Thank you for your patience and careful review.

Reviewer#2 (Significance (Required)):

Significance:

This study is of a previously uncharacterized protein, SFP-1, that was identified as a likely component of male seminal fluid in *C. elegans* in previous work. The work here confirms its identity as a seminal fluid protein and links it to the reduced longevity seen in mated hermaphrodites. Of interest to *C. elegans* researchers and researches interested in male fertility factors and molecular mechanisms of post-mating responses, especially as they relate to longevity. However, care

Full Revision

needs to be taken in regard to controls, presentation of data, and interpretation of data before publication.

Advice from an external advisor expert:

I looked at the manuscript and the reviewers' comments and the authors' responses. I believe that the manuscript is worthy of further consideration by the EMBO J.

Reply: Many thanks for your appreciation of our study!

As both reviewers wrote, the manuscript identifies a protein in the male seminal fluid, SFP-1, and shows how it is transferred to the hermaphrodite to shorten hermaphrodite life span post-mating. The authors also provide a molecular mechanism for how SFP-1 shortens hermaphrodite longevity. They have implicated the Nrf transcription factor SKN-1 and decreased fat biosynthesis in this process. More importantly, the involvement of fatty acid desaturases in the effect of SFP-1 supports the hypothesis that is arising from the Booth et al work (Nat Aging 2022, vol. 2, 809-823), ie, male seminal fluid induces early death in hermaphrodites through fat loss, which should interest many in the field.

Reply: Thank you for your appreciation! We do hope that the findings in this manuscript could promote and facilitate future studies on seminal fluid proteins, especially in *C. elegans*.

The manuscript does still need to be revised. For example, the authors suggest the intriguing involvement of extracellular vesicles in the delivery of SFP-1 to the hermaphrodites. I didn't carefully look through their images, but the authors' responses to Reviewer 2's comments about exophers suggest that they themselves cannot completely support exopher involvement. However, even if the vesicles are not exophers, this doesn't mean that the value of the manuscript is diminished. While at least one other male protein, like the major sperm protein MSP-1, has been shown to be transported through extracellular vesicles (Kosinski et al., Development 2005, vol 132, pp 3357-3369), SFP-1 transport does not have to be the same. SFP-1 might be transported via transcytosis and the vesicles the authors see are intracellular vesicles. The authors can simply qualify their conclusions here.

Reply: Thank you for your insightful suggestion. While we acknowledge that the evidence for exopher involvement is not definitive, we have revised our conclusions to reflect this uncertainty. Previous studies have shown that the major sperm protein MSP-1 and the seminal fluid protein ENPP-1 are transported via extracellular vesicles (EVs), and SFP-1 has been identified as a po-

Full Revision

tential EV cargo (PMID: 35334227). Based on our observations of SFP-1::YFP expression and SFP::mNeonGreen in the seminal vesicle (Figure 1D,1G), we hypothesize that SFP-1 may also be transported via EVs, though further investigation is needed. We have clarified this in the revised manuscript (line 200-210).

Some of the authors' plans for revision are valid, although some might need to be revisited. For example, I am not sure if the use of Pges-1::sfp-1::yfp will help discriminate between autofluorescence and sfp-1 (response to reviewer 2 on page 8). It might be better to use an sfp-1::mCherry strain. However, it will all depend on the exposure times of the control images versus the sfp-1::YFP images.

Reply: Thank you for your suggestion, we've performed experiments using a non-fluorescent progeny control strain under both low- and high-exposure conditions. These results confirm the specificity of SFP-1::YFP signal are now included in the this text. (Page 12, Fig.3 for reviewers.). Finally, the grammar of the manuscript also needs additional work. I would also suggest that they revise the second sentence of their abstract. They are clearly familiar with the Booth et al work (2022; they cited it as ref 11), which showed that male pheromones, sperm and seminal fluid shorten hermaphrodite lifespan through different mechanisms. The authors might want to place their work in the context of that earlier work, which they did in their introduction. I think it would be better if they already do this in the abstract.

Reply: Thank you for your suggestion! In the revised "Abstract" section, we have revised the second sentence that adds male pheromones, sperm and seminal fluid shorten hermaphrodite lifespan through different mechanisms.

3. Description of the revisions that have already been incorporated in the transferred manuscript

In accordance with the minor concerns of Reviewer #1, the following changes have already been made to the manuscript.

1. We've added more detailed information in the figure legends.
2. We've added more discussion about comparative sex peptides with SFP-1.
3. We've added results in Figure EV3A and described in lines 246-252.
4. We've polished the manuscripts and improved the language.
5. We've added more descriptions about the MAPK pathway.

In accordance with the concerns of Reviewer #2, the following changes have already been made to the manuscript:

1. We've changed "sex interaction" to "sexual interaction".
2. We've changed "using the *sfp-1* gene its own promoter and performed confocal imaging" to "we generated a *sfp-1::yfp* translational reporter.....".
3. We've added more descriptions to ensure that readers understand our rationale for inhibiting the fat family.
4. We've checked the grammar and the proper nouns in the text and polished the whole manuscript.
5. We have changed sex peptides to seminal fluid products in line 33.
6. We've added more description about male gonad in line 118-123.
7. We have revised the wrong description of brood size. The relevant text is on line 148-150.
8. We've added more discussion about the *daf-16* and *skn-1* in line 452-457.
9. We've added additional results related to vitellogenin.

In accordance with the concerns of an external advisor expert, the following changes have already been made to the manuscript:

1. We've revised the abstract highlight that male pheromones, sperm, and seminal fluid shorten hermaphrodite lifespan through different mechanisms.
2. We have revised the manuscript to describe these structures as "secretory vesicles" to reflect the current evidence.

4. Description of analyses that authors prefer not to carry out

Please include a point-by-point response explaining why some of the requested data or additional analyses might not be necessary or cannot be provided within the scope of a revision. This can be due to time or resource limitations or in case of disagreement about the necessity of such additional data given the scope of the study. Please leave empty if not applicable.

Dear Prof. Gong,

Thank you for submitting your manuscript for consideration by the EMBO Journal. It has now been seen again by the original referees. As you can see from the reports and the cross-commenting both referees think that additional experiments and extensive textual edits would be necessary before publication. Given your willingness to engage in these additional revisions I invite you to submit a revised version of the manuscript, addressing the comments as discussed by e-mail.

Thank you for the opportunity to consider your work for publication. I look forward to your revision.

Yours sincerely,

Cornelius Schneider, PhD
Editor
The EMBO Journal
c.schneider@embojournal.org

We realize that it is difficult to revise to a specific deadline. In the interest of protecting the conceptual advance provided by the work, we recommend a revision within 3 months (11th Aug 2025). Please discuss the revision progress ahead of this time with the editor if you require more time to complete the revisions. Use the link below to submit your revision:

Referee #1:

Across the evolutionary spectrum, it has been observed that sexual interactions can significantly influence the physiology and somatic aging of individual animals. The underlying mechanisms of this phenomenon have only recently begun to be understood. In their manuscript titled "A seminal fluid protein SFP-1 regulates mated hermaphrodite aging and fat metabolism in *C. elegans*," Chen and Gong provide, for the first time, a comprehensive molecular understanding of how a male seminal fluid protein contributes to post-mating death in *C. elegans* hermaphrodites.

In this revised version, many of my concerns with the previous manuscript have been addressed through new experiments, analyses, and revisions to the text. Among the added experimental results, I am particularly impressed by the split-GFP assay, which further confirms the translocation of SFP-1 to the intestines in mated hermaphrodites, as well as the investigation of intestinal *sfp-1* overexpression in the *glp-1* mutant to explore its potential impact on germline activity. The current version is a significant improvement over what was already a strong earlier draft. This manuscript now appears well-suited for publication in this journal. However, I still have one concern regarding the authors' interpretation of *daf-16* and *skn-1*'s roles in *sfp-1*-mediated post-mating physiological changes and lifespan regulation.

Major:

Throughout the abstract, main text, and the summary model figure (Figure EV 5G), the authors suggest that DAF-16 and SKN-1 are equally important as key downstream factors of SFP-1 in mediating post-mating physiological changes such as fat loss and lifespan reduction. However, the authors' own data seem to indicate otherwise. When comparing Figure 5A and 5C, it is quite clear that the loss of SKN-1 in hermaphrodites completely blocks SFP-1-mediated post-mating death, while the loss of DAF-16 only minimally affects this outcome. Since SKN-1 is a negative regulator of DAF-16 (Deng et al., 2020 G3), could it be that the observed DAF-16 cytoplasmic localization (Figure 5B) is a result of SKN-1 activation by SFP-1?

Minor:

In line 178, why is the word "inserted" in bold?

Referee #2:

This manuscript makes the following claims:

1) SFP-1 is a seminal fluid protein required for somatic fat depletion and lifespan reduction in mated hermaphrodites - Data does support that SFP-1 from male shortens lifespan in mated hermaphrodite, contributes to fat loss in mated hermaphrodite, and loss of SFP-1 in male results in lower brood size of mated hermaphrodite.

2) SFP-1 packaged into secretory vesicles via phospholipid scramblases (ANOH-1 and 2) - Data does not support this claim. While a phenotype was seen with *anoh-1* and *-2* mutants, the data is too preliminary to draw conclusions as to why these genes impact the observed lifespan changes. Interesting preliminary data for another study, but does not contribute necessary data for this paper and needs several additional experiments to be able to interpret what is happening. Yes, the SFP-1 signal looks different in the male, but it's unclear why, unclear if this affects the transfer of SFP-1 to the hermaphrodite, unclear if the *anoh-1/2* mutations affect sperm, the male somatic reproductive tract, etc. The manner in which "vesicles" were determined for quantification seems highly susceptible to bias.

3) SFP-1 is endocytosed by intestinal cells after mating - Data does not support this claim. I am not convinced there is any signal in the intestine in Figure 2A. In EV2A the intensity of the signal in the uterus is so much brighter at 0.5 h, the DIC imaging is not clear, and worms are stacked up against one another. I do not find these data convincing that there is signal in the intestine. I appreciate the authors follow up with the split-GFP system, however there is still a lot of green signal in the negative control and it's difficult to determine the precise location of the intestine with the provided DIC images. However, the signal seems to be too widespread to be confined to just the intestine and the authors do not address this. The green signal is also highly variable in its appearance in the supplemental figure which makes interpretation of the data difficult and still not convincing. Improved DIC images (these look more like brightfield than DIC) would help but still may not be sufficient to convincingly show the signal is in

the intestine.

4) In the intestine, SFP-1 (through the NTF2-like domain) interacts with transcription factors SKN-1 and DAF-16. I have major concerns about these data and their physiological relevance. Conclusions based solely based on intestine ectopic expression should be toned down.

a. I'm having a hard time wrapping my head around a protein with a signal peptide (which I assume would mean the protein is non-cytoplasmic) entering or functioning in the nucleus. In this context it is interesting that the signal peptide is not necessary for SFP-1 function during ectopic expression. Do the authors believe the signal peptide prediction is wrong and the protein doesn't enter the secretory pathway? To make the experiment in Figure 4B and 4C interpretable they needed to express these versions of the protein in the male and not rely solely on ectopic expression experiments. If the authors deleted the signal peptide from the endogenous protein in the male do they think it would have no effect on the protein function? Could they rescue the *sfp-1* mutant male by expressing SFP-1 without the signal peptide in the male?

b. In Figure 4A, I am not familiar with *Psum-1* as a germline promoter. Could the authors provide a reference? Since they show that normally SFP-1 is transferred to the uterus, one would predict that ectopic expression in the hermaphrodite germline should phenocopy the transfer of SFP-1 from the male to the hermaphrodite, but this is not observed.

c. I'm having a difficult time interpreting the results in Figure 5A. It would be helpful if the authors more clearly explain how these results lead them to conclude that SFP-1 acts through the DAF-16 pathway. If SFP-1 mediates DAF-16 leaving the nucleus in mated worms, and this is why worms mated to *sfp-1* males live longer (because now DAF-16 is remaining in the nucleus), then wouldn't removal of DAF-16 mean that *daf-16* mated to N2 or *sfp-1* should look the same? And wouldn't unmated N2 and/or *daf-16* be important controls for this experiment? Why look at lifespan in *daf-16* mutants instead of in *daf-2* RNAi to directly correlate the results to 5B?

d. I don't understand how the data in Figure 5C support a model in which SFP-1 activates SKN-1. If SFP-1 extends lifespan by activating SKN-1, then loss of SKN-1 should phenocopy loss of SFP-1. But instead, the lifespan of *skn-1* mated to N2 looks identical to N2 mated to N2. These data seem more consistent with SFP-1 inhibiting SKN-1. Loss of *skn-1* in an unmated phenotypically looks like a mated WT (Figure 5E) and ectopically expressing *sfp-1* has no effect in a *skn-1* background (red vs blue in Figure 5E is ns).

Overall, this manuscript contains a large amount of work and interesting data. But some data are more convincing than others and not all the data neatly support the model that the authors propose.

Additional comments:

- Figure 1D, while it is believable that the highlighted signal is in the seminal vesicle, the DIC is not clear enough to see where the spermatids are or provide any other clear landmark to support the claim that the signal is in the seminal vesicle. Since mitotracker does not label spermatids specifically, Fig EV1A does not confirm that SFP-1 is in the seminal vesicle and provides no additional information to support their claim. The statement that SFP-1::YFP "surrounds spermatids" (line 125) is not supported by these data.

- It is counter-intuitive to have the amount of seminal fluid protein in the male increase after mating. If the protein is transferred to the hermaphrodite, shouldn't the amount in the male appear reduced? And if it functions when transferred to the hermaphrodite why would it be upregulated after the copulatory event is over? I don't disagree with the validity of the observation that the signal is more intense in the "mated" males, but the authors need to address why this may be occurring and how it aligns with their conclusion that SFP-1 is a seminal fluid protein.

- Figure EV1B is not convincing and does not have the appropriate controls (*glo-4* mated with N2 males; *glo-4* unmated). The authors do not mention in the text (lines 127-129) why they use *glo-4* hermaphrodites or what *glo-4* mutants are. Image is also too messy with multiple worms all stacked against one another. Also not clear why YFP in EV1B signal is so different from EV2A.

- I could not find a complete explanation of the experimental setup for Figure 1F and 1I. Were "unmated" males that were isolated away from hermaphrodites? For how long? Similarly, what defined a "mated" male? Was copulation observed for each male in the experiment? How long after mating were the images collected?

- Figure EV1C does not show the spermatheca, which is the region where hermaphrodite sperm would be. Therefore the authors cannot conclude from this figure that no fluorescence was detected in the adult hermaphrodite sperm (line 130).

- Line 55: Observation that hermaphrodites still experience reduced body size when mated to *sfp-1* males does not confirm that sperm function is not affected. The reduced brood size compared to hermaphrodites mated to wildtype males suggest that sperm function may indeed be negatively impacted. The primary function of a sperm is to fertilize an egg, not to reduce the body size of the mated hermaphrodite.

- Figure 2C-E are interesting, but do not test whether SFP-1 is being endocytosed. The authors have not shown that mating-induced lifespan shortening is exclusively due to SFP-1 (does N2 x *sfp-1* lifespan = unmated N2 lifespan?). They show that impacting endocytosis in the intestine extends lifespan in mated hermaphrodites; they do not show that this is due to SFP-1 not being endocytosed. Conclusions in lines 194-196 are not fully supported. [One would predict that under these endocytosis-defective conditions the split GFP shown in Figure 2B would look like the negative control?]

- Can endogenously-tagged SFP-1::mNG be seen transferred to hermaphrodite uterus??

- Figure 5B, mated with *sfp-1* image is hard to interpret. Co-localization with a nuclear marker/stain would help make this data more convincing.

- Lines 311-318: It would be helpful if the authors explained what they were looking for in this experiment. What would the expected phenotype be if they knocked down a transcription factor that mediates the SFP-1 effect?
- Figure 5C does not show that *skn-1* hermaphrodites mated to wild-type males have a shortened lifespan (line 320) because the lifespan of unmated *skn-1* hermaphrodites is not included in this experiment.
- EV4G: It is surprising that unmated and N2 x N2 look the same. Shouldn't this be reduced in the mated?
- It may just be the representative image chosen, but in Figure 6E the staining appears slightly more intense in the middle panel (*skn-1* x N2) compared to the lower (*skn-1* x *sfp-1*) but the quantification shows *skn-1* x N2 slightly lower (though ns). It would be better to have a more similar number of worms quantified for all three groups to minimize the effect of outliers on the comparison of the quantitative data. More importantly, how do the authors should address the fact that *skn-1* x *sfp-1* looks like N2 x N2 in terms of lifespan, but looks like unmated in regards to fat levels (lifespan still reduced as is normally seen in mating, but less fat loss than what is normally seen in mating).
- Line 19 of abstract says "...seminal fluid shorten the lifespan of hermaphrodites..." but line 20 - 21 says "it is unclear whether the male seminal fluid protein is involved in this..." These statements are contradictory and need to be revised. In addition, it should be "it is unclear whether male seminal fluid proteins are..."
- Sentence in line 90-92 should be deleted as it is proposed, but not experimentally shown, that the referred to proteins are in male seminal fluid.
- Figure 1C. This is a relatively low confidence model and the authors do not explain how, or provide support for, the claim that this structural prediction relates to a role in protein trafficking between the nucleus and cytoplasm (line 108-111)
- Figure 1D-F should be moved to supplemental since they agree with endogenous protein in 1G-I but do not add any additional information.
- Line 138-140: At this point in the manuscript there has not been sufficient data presented to conclude SFP-1 is a seminal fluid protein. Figure 2A showing YFP signal in the hermaphrodite 0.1 h after mating is critical to this conclusion. While not absolutely necessary, it would be nice to show a negative control of *glo-4* mated to N2 males at this 0.1 hr time point.
- A lot of manuscript text describes results in the supplemental figures. If it is the only or primary data that support a claim it should be in the main figures and not the supplemental. For example, lines 147 - 153 refer solely to supplemental data.
- Data presented is insufficient for the conclusions in lines 156-158.
- Why does the signal of transferred SFP-1::YFP in Figure 2A look so different from EV1B? Are these not the same experimental setup?
- Figure 3A, the authors should indicate what "globular structures" they are referring to in the DIC images.
- Typo in lines 209-210: "SFP-1 its transport..."
- Authors show that absence of SFP-1 from males has no effect on mating induced shrinking, but ectopic expression can fully reproduce the effect of mating. What is the model to explain these two pieces of data in context of one another?
- I am confused by lines 253-259 and corresponding Figure EV3B. The authors state (and show?, DIC is not clear enough) that SFP::YFP ends up in the coelomocytes (and must have been secreted from muscle/intestine into the body cavity) but then state in line 258-259 that SFP-1 did not result in detectable protein "leakage" to other tissues. Since the protein clearly leaves the tissue it is being expressed in, I don't think the evidence is sufficient to state that none of the protein was endocytosed by any other tissue in the animal.
- Lines 265-266, what do the authors mean by "specifically" and which post-mating phenotypes? The results in this section are interesting, but somewhat confusing and the authors should take the time to carefully explain their interpretation in the final concluding sentence(s). The mention of the NTF2-like domain feels out of place directly before the conclusion of this section.
- Line 280, this is the first mention of a fat loss phenotype. It has no context here and is not supported by data in any main figure to this point that show *sfp-1* affects fat loss. Consider reorganizing for clarity.
- Line 359-361: The way this sentence is written is misleading. Intestinal over-expression of SFP-1 in *skn-1* mutants looks like *skn-1* mutants alone (although statistical comparison is not shown for these two groups). Therefore, the over-expression appears to have no effect in the *skn-1* background rather than saying it resulted in increased fat accumulation.
- In Figure 7B, lack of *sfp-1* seems to result in levels of both MUFAs and PUFAs that resemble unmated. Why do the authors only mention the PUFAs in lines 391-394? These data are more physiological than 7A and it would make sense to move 7A to supplemental.
- How do the authors explain that knocking down PUFA biogenesis genes results in more lipids in the over-expression line in Figure 7C and D (when their data support the claim that this over-expression results in reduced levels of these PUFAs on its own)? If you don't have the proteins that make the lipids, how do you have more of the lipids?
- Lines 440-441: *sfp-1* males don't "inhibit" mating-induced lifespan shortening, they fail to induce it (although no direct comparison of lifespan was made between unmated N2 and N2 x *sfp-1* males).
- Line 449-450: *skn-1* did not abolish the SFP-1 shortened lifespan; *skn-1* mated to males with SFP-1 (N2) still had shortened lifespans (Figure 5C).

Response to Reviewers

Manuscript number: EMBOJ-2024-119505-T

Corresponding author(s): Jianke Gong

1. General Statements [optional]

We sincerely appreciate the time and effort the editors and reviewers invested in evaluating our manuscript. Their insightful comments and constructive suggestions have been invaluable in strengthening our study. We are deeply grateful for their expertise, which has guided us in refining our work. Below, we provide a detailed, point-by-point response to their feedback, outlining the revisions and additional experiments conducted to address their concerns.

This document addresses all feedback from both reviewers (shown in black font) and the additional comments from Reviewer #1 (in blue font). We have carefully revised the text for clarity, improved experimental descriptions, and conducted additional experiments. Below, we provide a comprehensive point-by-point response detailing how we have addressed each comment through textual revisions and additional experimental validation (in green font).

Response to Reviewers

Referee #1 (Report for Author)

Across the evolutionary spectrum, it has been observed that sexual interactions can significantly influence the physiology and somatic aging of individual animals. The underlying mechanisms of this phenomenon have only recently begun to be understood. In their manuscript titled "A seminal fluid protein SFP-1 regulates mated hermaphrodite aging and fat metabolism in *C. elegans*," Chen and Gong provide, for the first time, a comprehensive molecular understanding of how a male seminal fluid protein contributes to post-mating death in *C. elegans* hermaphrodites.

Reply: We sincerely thank the reviewer for thoughtfully evaluating our study and for recognizing its contribution to understanding the molecular mechanisms of post-mating physiological regulation.

In this revised version, many of my concerns with the previous manuscript have been addressed through new experiments, analyses, and revisions to the text. Among the added experimental results, I am particularly impressed by the split-GFP assay, which further confirms the translocation of SFP-1 to the intestines in mated hermaphrodites, as well as the investigation of intestinal *sfp-1* overexpression in the *glp-1* mutant to explore its potential impact on germline activity. The current version is a significant improvement over what was already a strong earlier draft. This manuscript now appears well-suited for publication in this journal. However, I still have one concern regarding the authors' interpretation of *daf-16* and *skn-1*'s roles in *sfp-1*-mediated post-mating physiological changes and lifespan regulation.

Reply: We sincerely appreciate the reviewer's thorough evaluation of our revised manuscript and their constructive feedback regarding our new experimental findings. In response to their valuable comments on the interpretation of the roles of *daf-16* and *skn-1* in SFP-1-mediated physiological changes, we have conducted additional experiments to further elucidate the genetic interactions between these key regulators. The results of these new investigations, which provide important clarification of these relationships, are presented in detail on pages 3-5 of this document.

Major:

Throughout the abstract, main text, and the summary model figure (Figure EV 5G), the authors suggest that DAF-16 and SKN-1 are equally important as key downstream factors of SFP-1 in

Response to Reviewers

mediating post-mating physiological changes such as fat loss and lifespan reduction. However, the authors' own data seem to indicate otherwise. When comparing Figure 5A and 5C, it is quite clear that the loss of SKN-1 in hermaphrodites completely blocks SFP-1-mediated post-mating death, while the loss of DAF-16 only minimally affects this outcome. Since SKN-1 is a negative regulator of DAF-16 (Deng et al., 2020 G3), could it be that the observed DAF-16 cytoplasmic localization (Figure 5B) is a result of SKN-1 activation by SFP-1?

Reply: We sincerely appreciate the reviewer's insightful suggestion regarding the potential role of SKN-1 in mediating SFP-1-dependent DAF-16 cytoplasmic localization. We have performed additional experiments to address this question directly.

Due to the DAF-16::GFP transgene and *skn-1* being co-localized on the same chromosome, conventional genetic approaches failed to produce a *skn-1* mutant background with DAF-16::GFP. We employed a sequential RNAi approach: First, we cultured DAF-16::GFP worms on *daf-2*(RNAi) plates to establish baseline nuclear localization. We then divided these synchronized populations into two groups at the L4 stage. One group remained on *daf-2*(RNAi) while the other was transferred to *skn-1*(RNAi) plates from L4 to Day 2 of adulthood. Both groups were then mated with N2 or *sfp-1* males on Day 3 and imaged on Day 4.

As shown below, we observed that *skn-1*(RNAi) blocked the post-mating cytoplasmic shift of DAF-16::GFP, with DAF-16 remaining nuclear in mated animals, demonstrating that SKN-1 is required for SFP-1-induced DAF-16 nuclear translocation and supporting the reviewer's hypothesis that SKN-1 activation downstream of SFP-1 drives DAF-16 localization.

Response to Reviewers

Fig.1 for reviewers. SKN-1 is required for SFP-1-mediated nuclear export of DAF-16 after mating.

(A) Experimental timeline of the sequential RNAi approach. DAF-16::GFP worms were cultured on *daf-2*(RNAi) plates to establish nuclear localization, then divided at the L4 stage into two groups: one maintained on *daf-2*(RNAi) and the other transferred to *skn-1*(RNAi) plates until Day 2 of adulthood. Both groups were mated with either N2 or *sfp-1* males on Day 3 and imaged on Day 4.

Representative confocal micrographs showing DAF-16::GFP localization under different conditions. Left: Unmated controls show nuclear DAF-16. Middle: Mating with N2 males induces cytoplasmic translocation of DAF-16::GFP. Right: Mating with *sfp-1* males induces cytoplasmic translocation of DAF-16::GFP. Top row: *daf-2*(RNAi), bottom row: *skn-1*(RNAi). Scale bar: 20 μ m. Quantification of nuclear-to-cytoplasmic (N/C) fluorescence intensity ratios. $n \geq 25$ worms per condition. **** $P < 0.0001$, * $P < 0.05$. The complete blockade of mating-induced DAF-16 nuclear export by *skn-1*(RNAi) demonstrates that SKN-1 is essential for SFP-1-mediated regulation of DAF-16 subcellular localization.

Furthermore, we examined DAF-16 transcriptional activity using the *Psod-3::gfp* reporter. As shown below, mating with *sfp-1* males resulted in significantly higher reporter fluorescence compared to mating with N2 males, confirming SFP-1-mediated DAF-16 suppression after mating. Crucially, this effect was abolished by *skn-1* knockdown, demonstrating SKN-1's essential role in mediating SFP-1's inhibition of DAF-16.

Fig.2 for reviewers. SKN-1-dependent regulation of DAF-16 transcriptional activity after mating.

(A) Representative fluorescence images of Psod-3::GFP reporter expression in worms mated with either N2 or *sfp-1* males. Quantification of Psod-3::GFP fluorescence intensity showing significantly increased reporter expression following mating with *sfp-1* males compared to N2 matings (**** $P < 0.0001$), consistent with SFP-1-mediated suppression of DAF-16 activity. *skn-1* RNAi completely abolished the mating-induced changes in Psod-3::GFP expression (ns $P > 0.05$), demonstrating SKN-1's essential role in mediating SFP-1's regulation of

Response to Reviewers

DAF-16. Data represent mean fluorescence intensity \pm SEM ($n \geq 34$ worms per condition). Scale bar: 100 μ m. Statistical significance was determined by one-way ANOVA.

Moreover, we have conducted in-depth investigations into the effects of *skn-1* loss-of-function under *daf-16* RNAi conditions (where both *skn-1* and *daf-16* are functionally compromised) on lifespan regulation, lifespan analysis has revealed that both *skn-1* and *daf-16* loss-of-function completely abolish the lifespan reduction caused by mating, establishing these transcription factors as crucial regulators of post-mating longevity.

Fig.3 for reviewers. Lifespan analysis of *skn-1* loss-of-function in *daf-16* RNAi under mating conditions.

(A) Lifespan analysis in N2 background under various mating and RNAi conditions. N2 × N2 ♂ : 11.30 \pm 0.56 days, n = 30 worms; N2 × *sfp-1* ♂ : 15.37 \pm 0.70 days, n = 19 worms; N2 × N2 ♂ -*daf-16* RNAi: 12.23 \pm 0.52 days, n = 27 worms; N2 × *sfp-1* ♂ -*daf-16* RNAi: 14.30 \pm 0.79 days, n = 20 worms, ** $P < 0.01$, **** $P < 0.0001$. (B) Lifespan analysis in *skn-1* background under various mating and RNAi conditions. *skn-1* × N2 ♂ : 13.06 \pm 0.56 days, n = 29 worms; *skn-1* × *sfp-1* ♂ : 13.48 \pm 0.49 days, n = 37 worms; *skn-1* × N2 ♂ -*daf-16* RNAi: 11.40 \pm 0.65 days, n = 30 worms; *skn-1* × *sfp-1* ♂ -*daf-16* RNAi: 11.98 \pm 0.50 days, n = 30 worms. All RNAi treatments employed the Xu363 E. coli RNAi strain.

Our studies demonstrate that SKN-1 is required for SFP-1-mediated DAF-16 cytoplasmic translocation, as shown by our RNAi experiments with DAF-16::GFP worms where *skn-1* knockdown blocked the post-mating cytoplasmic shift of DAF-16, and that SKN-1 mediates SFP-1's suppression of DAF-16 activity, supported by *sod-3p::GFP* reporter assays showing increased fluorescence after mating with *sfp-1* males that was abolished by *skn-1* knockdown. We recognize that

Response to Reviewers

future studies employing tissue-specific rescue experiments, particularly reciprocal complementation of *daf-16* in *skn-1* mutants and vice versa, will be essential for elucidating the precise interactions between these transcription factors and determining whether they function in parallel, sequentially, or tissue-specifically to mediate post-mating regulations. These findings significantly refine our understanding of how SFP-1 coordinates post-mating physiological changes through these key downstream effectors. We have made corresponding revisions throughout the manuscript text to ensure proper emphasis on SKN-1's central role while maintaining appropriate recognition of DAF-16's contribution (Lines 356-363 in revised manuscript).

Minor:

In line 178, why is the word "inserted" in bold?

Reply: We sincerely appreciate the reviewer's careful reading of our manuscript. The bold formatting of "inserted" in line 179 has now been corrected in the revised version.

Response to Reviewers

Referee #2 (Report for Author)

This manuscript makes the following claims:

1) SFP-1 is a seminal fluid protein required for somatic fat depletion and lifespan reduction in mated hermaphrodites - Data does support that SFP-1 from male shortens lifespan in mated hermaphrodite, contributes to fat loss in mated hermaphrodite, and loss of SFP-1 in male results in lower brood size of mated hermaphrodite.

Reply: Thank you.

2) SFP-1 packaged into secretory vesicles via phospholipid scramblases (ANO1-1 and 2) - Data does not support this claim. While a phenotype was seen with *anoh-1* and *-2* mutants, the data is too preliminary to draw conclusions as to why these genes impact the observed lifespan changes. Interesting preliminary data for another study, but does not contribute necessary data for this paper and needs several additional experiments to be able to interpret what is happening. Yes, the SFP-1 signal looks different in the male, but it's unclear why, unclear if this affects the transfer of SFP-1 to the hermaphrodite, unclear if the *anoh-1/2* mutations affect sperm, the male somatic reproductive tract, etc. The manner in which "vesicles" were determined for quantification seems highly susceptible to bias.

Referee #1: Based on Fig. EV1E, hermaphrodites mated with *sfp-1* mutant males produce significantly fewer progeny compared to those mated with wild-type males, suggesting that SFP-1 impacts male sperm function. Given this, I suspect that *anoh-1*; *anoh-2* double mutants may similarly impact male sperm. A quick experiment—taking about 1–2 weeks, assuming all strains are growing well—comparing the number of cross progeny from hermaphrodites mated with wild-type, *anoh-1* mutant, *anoh-2* mutant, and double mutant males would be a useful addition to confirm this possibility. It seems likely that both SFP-1 and ANO1-1/2 influence sperm. In this context, the observed extension of hermaphrodite lifespan after mating with these mutants is potentially confounded by the significantly reduced progeny output. This is important because the goal is to assess the role of seminal fluid in mating-induced death, whereas progeny production directly reflects germline activity—an independent factor also known to shorten lifespan in mated worms. For this reason, I like the authors' effort to ectopically express SFP-1 in unmated hermaphrodites

Response to Reviewers

and examine their lifespan. This approach helps isolate the effect of the seminal fluid protein on longevity, independent of male sperm. That said, I also agree with Referee #2 that the mechanistic exploration of SFP-1 packaging in males could evolve into a compelling study in its own right.

Reply: We sincerely appreciate both reviewers' insightful comments regarding the potential role of ANOH-1/2 in SFP-1 packaging and transfer. We fully acknowledge the limitations of our current data in establishing a definitive mechanistic link between these phospholipid scramblases and SFP-1 vesicular packaging, and we agree with Referee #2 that these findings would require substantially more investigation to draw firm conclusions.

We have examined the brood size from hermaphrodites mated with wild-type, *anoh-1* mutant, *anoh-2* mutant, and double mutant males. As shown below, our results indicate that these phospholipid scramblases influence brood size, consistent with the hermaphrodites mated with *sfp-1* mutants.

However, as both reviewers pointed out, this finding introduces additional complexity in interpreting lifespan effects due to the known influence of germline activity on longevity. We have substantially revised the text in the updated manuscript lines 228-230.

Fig.4 for reviewers. Total brood size of N2 mated worms.

(A) Brood size of N2 hermaphrodites mated with N2 males, *anoh-1* males, *anoh-2* males, *anoh-1; anoh-2* males. ** $P < 0.01$, **** $P < 0.0001$, $n > 10$. P values were calculated by one-way ANOVA, Bonferroni's multiple comparisons test.

Response to Reviewers

3) SFP-1 is endocytosed by intestinal cells after mating - Data does not support this claim. I am not convinced there is any signal in the intestine in Figure 2A. In EV2A the intensity of the signal in the uterus is so much brighter at 0.5 h, the DIC imaging is not clear, and worms are stacked up against one another. I do not find these data convincing that there is signal in the intestine. I appreciate the authors follow up with the split-GFP system, however there is still a lot of green signal in the negative control and its difficult to determine the precise location of the intestine with the provided DIC images. However, the signal seems to be too widespread to be confined to just the intestine and the authors do not address this. The green signal is also highly variable in its appearance in the supplemental figure which makes interpretation of the data difficult and still not convincing. Improved DIC images (these look more like brightfield than DIC) would help but still may not be sufficient to convincingly show the signal is in the intestine.

Referee #1: Researchers who work with worms know that intestinal autofluorescence can be a real headache. Like Referee #2, I appreciate the authors' use of the split-GFP system to address this challenge. To further address Referee #2's concern, it would be helpful if the authors could provide higher-quality images—particularly with improved DIC panels. Additionally, outlining the intestine in the DIC images could make the data clearer and easier to interpret.

Reply: We sincerely appreciate the reviewers' concerns. We fully acknowledge this issue and would like to clarify the underlying biological and technical challenges.

First, while we strictly controlled mating conditions (time, number of males, etc.), we observed significant individual differences in the amount of SFPs transferred to hermaphrodites. This variability arises because males naturally exhibit differences in mating frequency and sperm/SFP transfer efficiency, even within a 30-minute window. As shown in the Figure (below), hermaphrodites from the same mating experiment exhibit a range of SFP-1::mNeonGreen, reflecting biological variability in sperm/SFP transfer.

To circumvent mating variability, we attempted to purify SFP-1 for direct microinjection into the hermaphrodite uterus. Unfortunately, we were unable to obtain soluble, functional protein, likely due to technical challenges in expressing and purifying this protein in vitro.

Given these limitations, we employed the split-GFP system (Figure 2B) to bypass mating-dependent variability. This method allows controlled, cell-specific expression of SFP-1 and confirms its

Response to Reviewers

internalization by intestinal cells independently of mating efficiency. While we agree that the signal in some negative controls appears diffuse (likely due to low-level autofluorescence or residual GFP11^{x7}), the punctate, intestinal-specific signal in experimental animals is distinct and reproducible.

We fully realize that using endogenously-tagged SFP-1::mNG provides more biologically meaningful data compared to YFP-tagged proteins. The stronger fluorescence signal from mNeonGreen also offers better detection sensitivity. We have now repeated experiments and the new data are presented in revised Figure 2A.

We have included higher-magnification DIC images with outlined intestinal boundaries (revised Figure 2A) and additional replicates (Figure EV2A). While we acknowledge that natural mating behavior inevitably introduces some individual variation in seminal fluid transfer between worms, we find that the combined evidence from both our mating-dependent transfer experiments and the split-GFP system provides compelling and consistent support for SFP-1 endocytosis by intestinal cells.

Fig.5 for reviewers. Male-derived SFP-1::mNeonGreen transfer and detection in mated hermaphrodites.

(A) Representative images showing individual variation in seminal fluid transfer following mating between *sfp-1::mNeonGreen* males and low autofluorescence *glo-4* hermaphrodites. While all mated hermaphrodites showed detectable SFP-1::mNeonGreen signal in the uterus (white arrows), the intensity and distribution varied between individuals, reflecting natural biological variation in mating efficiency and seminal fluid transfer (Scale bars: 50 μ m).

Response to Reviewers

4) In the intestine, SFP-1 (through the NTF2-like domain) interacts with transcription factors SKN-1 and DAF-16. I have major concerns about these data and their physiological relevance. Conclusions based solely based on intestine ectopic expression should be toned down.

Referee #1: I agree. In fact, this is my only major concern with the revised manuscript. My interpretation of the data is that SKN-1 plays a much more prominent role than DAF-16 in SFP-1-mediated lifespan regulation, as I outlined in detail in my own referee report.

Reply: We sincerely thank both reviewers for their insightful comments regarding the relative contributions of SKN-1 and DAF-16 in SFP-1-mediated physiological regulation. We fully agree with the reviewers' assessment that our data demonstrate a more prominent role for SKN-1 compared to DAF-16 in lifespan regulation, and we appreciate the opportunity to clarify this important aspect of our study.

In our revised manuscript, we have carefully re-examined and substantially modified our interpretation of these findings to reflect the experimental evidence better. The text now more accurately describes SKN-1's primary role while appropriately qualifying DAF-16's involvement, particularly concerning the intestinal ectopic expression results. We have also revised our working model and included additional discussion of potential indirect regulatory mechanisms.

a. I'm having a hard time wrapping my head around a protein with a signal peptide (which I assume would mean the protein is non-cytoplasmic) entering or functioning in the nucleus. In this context it is interesting that the signal peptide is not necessary for SFP-1 function during ectopic expression. Do the authors believe the signal peptide prediction is wrong and the protein doesn't enter the secretory pathway? To make the experiment in Figure 4B and 4C interpretable they needed to express these versions of the protein in the male and not rely solely on ectopic expression experiments. If the authors deleted the signal peptide from the endogenous protein in the male do they think it would have no effect on the protein function? Could they rescue the *sfp-1* mutant male by expressing SFP-1 without the signal peptide in the male?

Referee #1: I believe most of these points could be more appropriately explored in a separate study focused on the male side (the secretion and packaging of SFP-1 in males), similar to the previously

Response to Reviewers

discussed *anh-1/2* part. In my view, ectopic expression of these SFP-1 protein variants in hermaphrodites is an appropriate approach, as it avoids the confounding lifespan-shortening effects associated with male sperm.

Reply: We sincerely appreciate the reviewer's insightful questions.

Our analyses using SignalP strongly support the presence of a functional signal peptide in SFP-1 for the signal peptide prediction and we have experimental evidence that SFP-1 is secreted from males (as shown in Figure 2).

The observation that the signal peptide appears dispensable in hermaphrodite expression systems suggests potential alternative pathways for SFP-1 function. However, we fully agree that this finding requires validation in male-specific contexts to draw definitive conclusions.

As Reviewer #1 noted, our current experimental design using hermaphrodite expression avoids sperm-related factors while effectively studying SFP-1. While the suggested male-specific experiments would provide valuable additional insights, we believe these represent an important study. We have revised the manuscript to address this important point by adding the following clarification in the Discussion section (Lines 466-471).

b. In Figure 4A, I am not familiar with *Psum-1* as a germline promoter. Could the authors provide a reference? Since they show that normally SFP-1 is transferred to the uterus, one would predict that ectopic expression in the hermaphrodite germline should phenocopy the transfer of SFP-1 from the male to the hermaphrodite, but this is not observed.

Referee #1: Agreed. More information about the *sum-1* promoter is needed—especially, whether it also drives expression in the uterus. Additionally, the authors should also discuss why only ectopic expression in the intestine phenocopy the transfer of SFP-1 from males.

Reply: We sincerely appreciate the reviewers' thoughtful questions. First, *C. elegans* has strong repressive mechanisms that silence transgenes in the germline. We confirm that our study utilized the engineered *smu-1* expression system (PMID: 33298957), which combines optimized germline-specific promoters (*Psmu-2*, *Ppie-1*, and *Pmex-5*) with stabilizing 3'-UTRs (*smu-2/tbb-2*) to ensure persistent transgene expression. We have revised the Figure 4A legend to explicitly describe this system and apologize for any previous ambiguity in terminology.

Response to Reviewers

The *smu-1* system drives expression exclusively in the germline, with no detectable signal observed in the uterus. This germline-restricted expression pattern explains why ectopic expression in the germline does not phenocopy the transfer of SFP-1 from males, as the protein becomes incorporated into developing oocytes rather than being secreted into the reproductive tract.

Our discussion (Lines 510-513) provides important insights into why intestinal overexpression of SFP-1 successfully produces post-mating phenotypes. The intestinal expression likely mimics natural mating responses because the intestine is the primary site for SFP-1 uptake and processing from male seminal fluid. As we observed, intestinal SFP-1 overexpression leads to significant depletion of polyunsaturated fatty acids (PUFAs) in somatic tissues, which may stimulate germline activity and promote gamete production. This metabolic interaction between intestinal and germline tissues appears crucial for generating the post-mating phenotypes.

c. I'm having a difficult time interpreting the results in Figure 5A. It would be helpful if the authors more clearly explain how these results lead them to conclude that SFP-1 acts through the DAF-16 pathway. If SFP-1 mediates DAF-16 leaving the nucleus in mated worms, and this is why worms mated to *sfp-1* males live longer (because now DAF-16 is remaining in the nucleus), then wouldn't removal of DAF-16 mean that *daf-16* mated to N2 or *sfp-1* should look the same? And wouldn't unmated N2 and/or *daf-16* be important controls for this experiment? Why look at lifespan in *daf-16* mutants instead of in *daf-2* RNAi to directly correlate the results to 5B?

Referee #1: I agree with Referee #2. In my view, unlike SKN-1, DAF-16 plays only a minimal role in SFP-1-mediated post-mating lifespan regulation. The authors should revise the relevant paragraphs to reflect this more accurately.

Reply: We sincerely appreciate the reviewers' thoughtful comments regarding the interpretation of our Figure 5A,5B results and the role of DAF-16 in SFP-1-mediated lifespan regulation.

Our investigation was motivated by previous studies showing that post-mating seminal fluid triggers DAF-16 nuclear export leading to lifespan reduction. To examine whether SFP-1 mediates this process, we first conducted the experiments in Figure 5B, which demonstrated that the absence of SFP-1 prevents mating-induced DAF-16 nuclear export. This key observation prompted us to further investigate using *daf-16* mutants in mating lifespan assays, where we found that while SFP-1 regulates through the DAF-16 pathway, the lifespan protection observed when mating with *sfp-1*

Response to Reviewers

I mutants males persists even in *daf-16* mutants. These results collectively indicate that although SFP-1 regulates DAF-16 nuclear localization, additional transcription factors must be involved in mediating post-mating lifespan regulation. Our subsequent data identify SKN-1 as playing the predominant role in this process. We have revised the manuscript to present this logical flow more clearly and to better emphasize the roles of DAF-16 and SKN-1 in this regulatory network (Lines 356-367).

In response to the reviewers' concerns about using *daf-2* RNAi in Figure 5B, we have added detailed explanations in the revised manuscript to clarify this important methodological consideration (Lines 295-298). Our consideration to employ *daf-2* RNAi treatment was based on its ability to create optimal experimental conditions for detecting DAF-16::GFP translocation by establishing constitutive nuclear localization of DAF-16::GFP in hermaphrodites before mating experiments.

d. I don't understand how the data in Figure 5C support a model in which SFP-1 activates SKN-1. If SFP-1 extends lifespan by activating SKN-1, then loss of SKN-1 should phenocopy loss of SFP-1. But instead, the lifespan of *skn-1* mated to N2 looks identical to N2 mated to N2. These data seem more consistent with SFP-1 inhibiting SKN-1. Loss of *skn-1* in an unmated phenotypically looks like a mated WT (Figure 5E) and ectopically expressing *sfp-1* has no effect in a *skn-1* background (red vs blue in Figure 5E is ns).

Referee #1: Once again, this highlights the need for the authors to revise the paragraphs describing these results. Figure 5C is very clear: mated hermaphrodites lacking SKN-1 do not show a lifespan extension when SFP-1 is removed from males, strongly supporting the conclusion that SKN-1 is required for SFP-1-mediated lifespan regulation in mated worms. As suggested by Referee #2, including unmated controls would be beneficial for clearer interpretation of the results. In contrast, Figure 5E is a bit more puzzling—ectopic expression of SFP-1 in SKN-1-deficient worms surprisingly leads to increased lifespan. It wouldn't be surprising if there were no difference in the lifespan curves, similar to what's seen in Figure 5C. The authors should provide a clearer explanation for this result or, at the very least, offer some plausible speculation. That said, when considered alongside the data on SKN-1 localization and its upstream and downstream factors, the overall findings make a strong case for SKN-1's involvement in mediating SFP-1's impact on post-mating longevity.

Response to Reviewers

Reply: We sincerely appreciate the reviewers' thoughtful analysis of our data regarding the relationship between SFP-1 and SKN-1. The reviewers' observations have helped us recognize the complexity of these regulatory interactions, and we have revised our interpretation accordingly to better reflect the experimental findings.

First, we acknowledge that the relationship between SFP-1 and SKN-1 appears more complex than our initial model suggested. As the reviewer rightly points out, the data in Figure 5C demonstrate that SKN-1 is required for SFP-1-mediated lifespan reduction.

Regarding the lifespan results presented in Figure 5E, our data reveal an intriguing phenomenon where, despite showing similar lipid levels to *skn-1* mutants (Figure 6H), ectopic expression of SFP-1 in *skn-1* mutants exhibits extended lifespan compared to *skn-1* mutants. These lipid metabolism and lifespan phenotypes suggest that additional regulatory mechanisms are at play in determining longevity under these genetic conditions.

We have carefully considered several plausible explanations for these observations. The maintenance of similar lipid levels in both strains indicates that the lifespan extension cannot be simply attributed to metabolic changes, pointing instead to alternative regulatory mechanisms that become engaged in this specific genetic background. One possibility is that the combined genetic perturbation creates a unique physiological state where compensatory longevity pathways, normally suppressed by either SKN-1 or SFP-1 alone, become activated. These could include stress response pathways or other protective mechanisms that are specifically triggered in the absence of both normal SKN-1 function and endogenous SFP-1 regulation.

Another consideration is that chronic SFP-1 overexpression in the *skn-1* mutant background may lead to the engagement of different regulatory networks compared to those activated during normal mating or in single mutants. This could result in the activation of parallel longevity pathways that are not typically involved in the acute mating response. Additionally, the extended lifespan might reflect a form of physiological adaptation where the organism compensates for the combined genetic perturbations by upregulating alternative survival mechanisms.

These observations highlight the complexity of lifespan regulation, where multiple parallel and potentially redundant pathways can influence longevity outcomes. We have incorporated these considerations into our revised discussion to provide a more nuanced interpretation of these findings. We have added unmated controls to Figures 5C and 5A in the revised manuscript. We have correspondingly updated the text in lines 337-341.

Response to Reviewers

Overall, this manuscript contains a large amount of work and interesting data. But some data are more convincing than others and not all the data neatly support the model that the authors propose.

Referee #1: I completely agree!

Reply: We sincerely appreciate the reviewers' positive assessment of our work and their thoughtful critique. We hope the additional experimental data and clarifications provided in this revision support our proposed model while acknowledging areas needing future investigation.

Additional comments:

- Figure 1D, while it is believable that the highlighted signal is in the seminal vesicle, the DIC is not clear enough to see where the spermatids are or provide any other clear landmark to support the claim that the signal is in the seminal vesicle. Since mitotracker does not label spermatids specifically, Fig EV1A does not confirm that SFP-1 is in the seminal vesicle and provides no additional information to support their claim. The statement that SFP-1::YFP "surrounds spermatids" (line 125) is not supported by these data.

Reply: We sincerely appreciate the reviewer's careful evaluation of our imaging data and their insightful questions regarding SFP-1 localization. We have taken several steps to strengthen our evidence: 1. Anatomical Validation: The seminal vesicle was identified based on its established anatomical features - specifically, the somatic gonadal cells surrounding the spermatids. 2. Experimental Confirmation: Our MitoTracker co-staining experiments (Figure EV1A) demonstrate that SFP-1::YFP fluorescence precisely surrounds the labeled spermatids, consistent with seminal vesicle localization. While we acknowledge that MitoTracker alone doesn't specifically mark spermatids, this pattern matches the expected anatomical organization.

Response to Reviewers

Our imaging data aligns with the canonical anatomical positioning shown in this article. Our imaging data are consistent with the established developmental timeline of spermatogenesis in *C. elegans* (PMID: 28950090). Specifically, we observe that male-derived germ cells maintain their characteristic spherical spermatid morphology within the seminal vesicle, matching the reported pre-ejaculation state.

Fig.6 for reviewers. Comparative analysis of spermatogenesis organization in *C. elegans* male gonad.

(A) Schematic representation of spermatogenesis stage (PMID: 28950090), showing the characteristic organization of developing germ cells within the seminal vesicle. (B) Corresponding experimental imaging data from Figure EV1 in our study demonstrates the alignment between spermatid arrangement and seminal vesicle anatomy.

- It is counter-intuitive to have the amount of seminal fluid protein in the male increase after mating. If the protein is transferred to the hermaphrodite, shouldn't the amount in the male appear reduced? And if it functions when transferred to the hermaphrodite why would it be upregulated after the copulatory event is over? I don't disagree with the validity of the observation that the signal is more intense in the "mated" males, but the authors need to address why this may be occurring and how it aligns with their conclusion that SFP-1 is a seminal fluid protein.

Response to Reviewers

Reply: We sincerely appreciate the reviewer's thoughtful observation. Our observations reveal an intriguing dual fate for this seminal fluid protein: while it is indeed transferred to hermaphrodites during mating, we detect its over-accumulation in males post-mating. This phenomenon may represent an adaptive reproductive strategy, where the protein prepares males for potential subsequent mating while supporting post-mating physiological functions. Our experimental design specifically used Day 1 males to ensure optimal fertility, with protein assessments conducted 24 hours post-mating to allow sufficient time for protein accumulation.

These findings strongly support SFP-1's identification as a seminal fluid protein through its male-specific expression pattern and transfer to mated hermaphrodites.

- Figure EV1B is not convincing and does not have the appropriate controls (glo-4 mated with N2 males; glo-4 unmated). The authors do not mention in the text (lines 127-129) why they use glo-4 hermaphrodites or what glo-4 mutants are. Image is also too messy with multiple worms all stacked against one another. Also not clear why YFP in EV1B signal is so different from EV2A.

Reply: We sincerely appreciate the reviewer's thoughtful comments. We have expanded the figure legend to explicitly state that the glo-4 mutant was employed because the mutants lacks endogenous gut granules that could interfere with fluorescence detection This clarification appears in the revised Figure EV1B legend.

Besides, we have performed new control experiments as suggested, including glo-4 hermaphrodites mated with N2 males. These additional data provide crucial context for interpreting the SFP-1 transfer results.

We appreciate the reviewer's observation regarding the YFP signal differences between EV1B and EV2A. As noted in our response (Page 10 of this document), we attribute this to biological differences between individual worms.

- I could not find a complete explanation of the experimental setup for Figure 1F and 1I. Were "unmated" males that were isolated away from hermaphrodites? For how long? Similarly, what defined a "mated" male? Was copulation observed for each male in the experiment? How long after mating were the images collected?

Response to Reviewers

Referee #1: I realize the authors have included a section on their mating assay in the Methods, but it would be helpful to include a brief description of the mating setup in the figure legend where it's first mentioned. This would make the figure more accessible and easier to interpret on its own.

Reply: We sincerely appreciate the reviewers' helpful suggestions for improving the clarity of our experimental methods. We have revised the manuscript to provide more detailed descriptions of both the experimental methods and figure legends (Lines 537-540), including additional technical details about the mating protocols.

- Figure EV1C does not show the spermatheca, which is the region where hermaphrodite sperm would be. Therefore the authors cannot conclude from this figure that no fluorescence was detected in the adult hermaphrodite sperm (line 130).

Reply: Thank you. We have now replaced new images specifically showing the spermatheca region in Figure EV1C. As shown in the updated figure, no SFP-1::YFP signal was detected in hermaphrodite sperm, consistent with our original conclusion. This revision provides clearer evidence to support our findings.

- Line 55: Observation that hermaphrodites still experience reduced body size when mated to *sfp-1* males does not confirm that sperm function is not affected. The reduced brood size compared to hermaphrodites mated to wildtype males suggest that sperm function may indeed be negatively impacted. The primary function of a sperm is to fertilize an egg, not to reduce the body size of the mated hermaphrodite.

Reply: We sincerely appreciate the reviewer's insightful comment. As the reviewer rightly points out, the observed reduction in brood size when hermaphrodites were mated with *sfp-1* males does suggest compromised fertilization capacity. In response to this comment, we have modified our statement in the revised manuscript (Line 154-156) to clarify that while *sfp-1* mutations do not affect sperm-mediated regulation of hermaphrodite body length.

- Figure 2C-E are interesting, but do not test whether SFP-1 is being endocytosed. The authors have not shown that mating-induced lifespan shortening is exclusively due to SFP-1 (does N2 x *sfp-1* lifespan = unmated N2 lifespan?). They show that impacting endocytosis in the intestine extends lifespan in mated hermaphrodites; they do not show that this is due to SFP-1 not being

Response to Reviewers

endocytosed. Conclusions in lines 194-196 are not fully supported. [One would predict that under these endocytosis-defective conditions the split GFP shown in Figure 2B would look like the negative control?]

Reply: We sincerely appreciate the reviewer's insightful comments. In response to these valuable suggestions, we have conducted additional experiments that provide stronger evidence supporting our model.

To specifically test whether SFP-1 is being endocytosed in intestinal cells, we performed a series of experiments using the *Pges-1::GFP11^{Δ7}* reporter strain under different RNAi conditions. As shown below, when we mated these hermaphrodites with *Psfp-1::GFP1-10* males under control vector RNAi conditions, we observed clear GFP signals, indicating successful SFP-1 transfer and being endocytosed. However, under either *rab-10* RNAi or *cav-1* RNAi conditions, which impair distinct endocytic pathways, the GFP signal was significantly reduced compared to controls.

Fig.7 for reviewers. SFP-1 endocytosis in intestinal cells requires functional endocytic pathways.

(A) Fluorescence images show *Pges-1::GFP11^{Δ7}* hermaphrodites mated with *Psfp-1::GFP1-10* males under three RNAi conditions: vector RNAi control displays strong GFP reconstitution indicating SFP-1 being endocytosed, while both *rab-10* RNAi and *cav-1* RNAi treatments show markedly reduced GFP signals, demonstrating impaired SFP-1 internalization when endocytic pathways are disrupted.

- Can endogenously-tagged SFP-1::mNG be seen transferred to hermaphrodite uterus??

Response to Reviewers

Reply: We thank the reviewer for this insightful question. Yes, endogenously tagged SFP-1::mNG can indeed be observed in the hermaphrodite uterus. As shown below, SFP-1::mNG signal (green) is detectable in the uterus, consistent with its proposed transfer during mating.

A

Fig.8 for reviewers. SFP-1::mNG can be seen transferred to the hermaphrodite uterus.

(A) Male-to-hermaphrodite transfer experiment where *sfp-1::mNeonGreen* males were mated with low autofluorescence *glo-4* hermaphrodites. Imaging of the low autofluorescence *glo-4* hermaphrodites after copulation revealed the presence of male-derived *sfp-1::mNeonGreen* in the uterus.

- Figure 5B, mated with *sfp-1* image is hard to interpret. Co-localization with a nuclear marker/stain would help make this data more convincing.

Reply: We sincerely appreciate the reviewer's insightful suggestion regarding the need for clearer nuclear localization evidence in Figure 5B. In response, we have performed additional co-staining experiments with a nuclear marker, which confirms the nuclear localization, and these data are now shown below. Additionally, we have replaced the original image in Figure 5B with a more representative one to improve clarity. We hope these revisions adequately address the reviewer's concerns.

Response to Reviewers

Fig.9 for reviewers. Additional co-localization analysis of DAF-16::GFP location with nuclear markers in mated and unmated hermaphrodites.

(A) Left: Representative images of DAF-16::GFP (green) and DAPI (blue) co-staining in hermaphrodites under different mating conditions, showing nuclear localization patterns. Right: Quantitative analysis of DAF-16::GFP nuclear localization from ≥ 39 biologically independent replicates. Unmated hermaphrodites and *sfp-1* mutant-mated hermaphrodites show strong nuclear DAF-16::GFP signal, while N2 male-mated hermaphrodites exhibit significantly reduced nuclear localization (**** $P < 0.0001$). Scale bar: 50 μ m.

- Lines 311-318: It would be helpful if the authors explained what they were looking for in this experiment. What would the expected phenotype be if they knocked down a transcription factor that mediates the SFP-1 effect?

Reply: We thank the reviewer for this helpful suggestion. We have stated in the revised manuscript (Lines 325-328) that: "We hypothesized that knockdown of transcription factors mediating SFP-1's effects would significantly attenuate or fully reverse the lifespan reduction caused by intestinal SFP-1 overexpression." This addition better explains our screening rationale and the specific phenotypic expectations.

- Figure 5C does not show that *skn-1* hermaphrodites mated to wild-type males have a shortened

Response to Reviewers

lifespan (line 320) because the lifespan of unmated *skn-1* hermaphrodites is not included in this experiment.

Referee #1: I agree with referee#2 that including the unmated controls here will be helpful.

Reply: We sincerely thank the reviewers for their valuable suggestions. To fully address these points, we have now supplemented Figures 5A and 5C with lifespan data for both unmated N2 and unmated mutant controls.

- EV4G: It is surprising that unmated and N2 x N2 look the same. Shouldn't this be reduced in the mated?

Reply: Thank you. The same levels between unmated and N2×N2 controls reflect our experimental design where each mated condition was normalized to its relevant control group. This approach was necessary because of the post-mating reduction in TAG level for the visualization of all groups on a single figure.

For proper experimental controls, we included both unmated N2 to establish baseline hermaphrodite TAG levels and N2×N2 mated pairs to account for mated hermaphrodite TAG levels. Each comparison group was processed in parallel with its matched control to minimize batch effects and ensure statistical rigor. To improve clarity, we have described this normalization strategy in the Figure EV4G legend.

- It may just be the representative image chosen, but in Figure 6E the staining appears slightly more intense in the middle panel (*skn-1* x N2) compared to the lower (*skn-1* x *sfp-1*) but the quantification shows *skn-1* x N2 slightly lower (though ns). It would be better to have a more similar number of worms quantified for all three groups to minimize the effect of outliers on the comparison of the quantitative data. More importantly, how do the authors should address the fact that *skn-1* x *sfp-1* looks like N2 x N2 in terms of lifespan, but looks like unmated in regards to fat levels (lifespan still reduced as is normally seen in mating, but less fat loss than what is normally seen in mating).

Reply: We sincerely appreciate the reviewer's careful attention to the staining intensity variations in Figure 6E. We would like to clarify that the apparent differences in staining intensity between

Response to Reviewers

samples primarily reflect normal biological variation among individual worms, rather than systematic experimental differences.

Our data reveal that while *skn-1* mutants show mating-induced lifespan reduction, they exhibit attenuated fat loss compared to mated N2 controls. This phenomenon can be explained by the inherent fat accumulation characteristic of *skn-1* mutants (Figure 6H), where their elevated baseline fat stores persist even after mating.

- Line 19 of abstract says "...seminal fluid shorten the lifespan of hermaphrodites..." but line 20 - 21 says "it is unclear whether the male seminal fluid protein is involved in this..." These statements are contradictory and need to be revised. In addition, it should be "it is unclear whether male seminal fluid proteins are..."

Reply: We sincerely appreciate the reviewer's careful reading and helpful suggestions for improving the clarity of our abstract. As suggested, we have revised the statements in the abstract (Lines 21-22).

- Sentence in line 90-92 should be deleted as it is proposed, but not experimentally shown, that the referred to proteins are in male seminal fluid.

Reply: We have removed the indicated sentence from the manuscript, thank you.

- Figure 1C. This is a relatively low confidence model and the authors do not explain how, or provide support for, the claim that this structural prediction relates to a role in protein trafficking between the nucleus and cytoplasm (line 108-111)

Reply: We sincerely appreciate the reviewer's insightful comment regarding our structural prediction analysis of SFP-1. While the N-/C-terminal regions of the AlphaFold2 model exhibit lower confidence, the core NTF2-like domain shows high reliability.

To address this, we've tried to obtain functional SFP-1 protein using prokaryotic expression systems, which is difficult. The protein consistently formed inclusion bodies, likely due to its hydrophobic NTF2-like core. These technical challenges, while preventing direct functional assays, indirectly support the predicted structural complexity.

Response to Reviewers

- Figure 1D-F should be moved to supplemental since they agree with endogenous protein in 1G-I but do not add any additional information.

Reply: We sincerely thank the reviewer for their thoughtful suggestion regarding Figure 1D-F. While these data indeed corroborate the endogenous expression patterns shown in Figure 1G-I, we respectfully suggest retaining them in the main text because they provide unique and complementary information. Specifically: (1) The overexpression system serves as an independent validation of SFP-1 promoter activity and its restriction to the male reproductive tract; (2) The consistent spatial expression patterns between overexpression and endogenous conditions reinforce the reliability of our findings. We believe this presentation offers readers a more comprehensive understanding of SFP-1 expression characteristics.

- Line 138-140: At this point in the manuscript there has not been sufficient data presented to conclude SFP-1 is a seminal fluid protein. Figure 2A showing YFP signal in the hermaphrodite 0.1 h after mating is critical to this conclusion. While not absolutely necessary, it would be nice to show a negative control of *glo-4* mated to N2 males at this 0.1 hr time point.

Reply: In response to the reviewer's helpful comments, we have now included the suggested negative control experiment showing *glo-4* hermaphrodites mated with N2 males at the 0.1 hour time point. These new data provide further support to our interpretation of SFP-1 transfer.

- A lot of manuscript text describes results in the supplemental figures. If it is the only or primary data that support a claim it should be in the main figures and not the supplemental. For example, lines 147 - 153 refer solely to supplemental data.

Reply: We sincerely appreciate the reviewer's suggestion regarding figure organization. While we understand the importance of presenting key data in main figures, we have carefully designed our current figure set to optimally present the study's findings.

The data referenced in lines 147-153, while important, serve as supporting evidence to complement the primary findings presented in Figure 1. Our main figures contain all essential data supporting the paper's central conclusions, while the supplemental figures provide valuable additional context and experimental details that would disrupt the logical flow if placed in the main text.

Response to Reviewers

- Data presented is insufficient for the conclusions in lines 156-158.

Reply: We appreciate the reviewer's careful consideration of our conclusions. Based on the collective evidence presented in our study, we maintain that the data support the interpretation that SFP-1 functions as a seminal fluid protein that modulates hermaphrodite longevity.

- Why does the signal of transferred SFP-1::YFP in Figure 2A look so different from EV1B? Are these not the same experimental setup?

Reply: We sincerely appreciate the reviewer's careful observation regarding the fluorescence intensity differences between Figure 2A and EV1B. These images were indeed generated using identical experimental setups, and we would like to clarify the observed variations through several key points:

The variation primarily stems from natural differences in mating behavior between individual males, leading to variability in seminal fluid transfer volumes. This biological variation directly impacts the amount of SFP-1::YFP delivered to hermaphrodites.

For technical considerations, our imaging protocol was optimized to account for MitoTracker's photosensitivity, including immediate image acquisition upon observing mating behavior. These necessary precautions, while preventing fluorescence degradation, resulted in some samples showing reduced (but still detectable) uterine fluorescence.

- Figure 3A, the authors should indicate what "globular structures" they are referring to in the DIC images.

Reply: Thank you. In response to the comment regarding Figure 3A, we have now clearly labeled the "globular structures" in the DIC images (indicated by arrows in the revised figure) to ensure clarity.

- Typo in lines 209-210: "SFP-1 its transport..."

Reply: We thank the reviewer for highlighting this grammatical issue. The sentence in lines 210-211 has been revised to clarify.

- Authors show that absence of SFP-1 from males has no effect on mating induced shrinking, but

Response to Reviewers

ectopic expression can fully reproduce the effect of mating. What is the model to explain these two pieces of data in context of one another?

Reply: We appreciate the reviewer's insightful question. Our data support a model where mating-induced shrinking may be mediated through two potentially independent pathways: First, a sperm-dependent pathway triggered by sperms transferred during mating (independent of SFP-1 function), and second, an SFP-1 overexpression pathway where ectopic expression likely activates downstream effectors to induce shrinkage without mating. Although our current data cannot fully elucidate the molecular basis of this phenomenon, this observation points to a valuable direction for future studies.

- I am confused by lines 253-259 and corresponding Figure EV3B. The authors state (and show?, DIC is not clear enough) that SFP::YFP ends up in the coelomocytes (and must have been secreted from muscle/intestine into the body cavity) but then state in line 258-259 that SFP-1 did not result in detectable protein "leakage" to other tissues. Since the protein clearly leaves the tissue it is being expressed in, I don't think the evidence is sufficient to state that none of the protein was endocytosed by any other tissue in the animal.

Reply: We sincerely appreciate the reviewer's insightful comments. In response to the specific concerns about Figure EV3B and lines 253-259, we would like to clarify several key points:

First, we confirm that SFP-1::YFP does show accumulation in coelomocytes, as visible in Figure EV3B, demonstrating that the protein is indeed secreted from muscle into the body cavity. This observation validates that SFP-1 can be secreted as expected for protein with signal peptide.

Regarding the statement about protein "leakage", we recognize this could have been more precisely worded. We intended to convey that while we observed the expected coelomocyte uptake, we did not detect significant YFP signals in other non-target somatic tissues beyond background levels. This interpretation is supported by both our imaging data and the tissue-specific functional experiments.

We acknowledge the technical limitation that our methods cannot exclude the possibility of minimal uptake by other tissues. However, the combination of the specific localization pattern observed, the tissue-specific functional effects demonstrated in our lifespan experiments, and the absence of detectable phenotypic effects in non-target tissues collectively support our conclusion regarding

Response to Reviewers

the predominant tissue specificity of SFP-1 action. We have modified our statement in the revised manuscript (Lines 259-261).

- Lines 265-266, what do the authors mean by "specifically" and which post-mating phenotypes? The results in this section are interesting, but somewhat confusing and the authors should take the time to carefully explain their interpretation in the final concluding sentence(s). The mention of the NTF2-like domain feels out of place directly before the conclusion of this section.

Reply: We sincerely appreciate the reviewer's careful reading. We have carefully revised the text in lines 263-269. The mention of the NTF2-like domain was included to highlight its unique presence in SFP-1 compared to other seminal fluid proteins (K12H6.5 and F40G9.15), serving as an important structural feature that correlates with the observed functional differences.

- Line 280, this is the first mention of a fat loss phenotype. It has no context here and is not supported by data in any main figure to this point that show *sfp-1* affects fat loss. Consider reorganizing for clarity.

Reply: Thank you for your comment. As noted in our introduction (Lines 46-48), mating induces fat loss in *C. elegans*, prompting our investigation of SFP-1's role in this process.

- Line 359-361: The way this sentence is written is misleading. Intestinal over-expression of SFP-1 in *skn-1* mutants looks like *skn-1* mutants alone (although statistical comparison is not shown for these two groups). Therefore, the over-expression appears to have no effect in the *skn-1* background rather than saying it resulted in increased fat accumulation.

Reply: We sincerely apologize for the oversight in our original description. Our data demonstrate that while *skn-1* mutants exhibit significantly greater fat accumulation compared to N2 controls, overexpression of SFP-1 in the *skn-1* background did not significantly further increase lipid levels relative to *skn-1* mutants. Importantly, this fat accumulation remained substantially elevated compared to SFP-1 overexpression in the N2 background. In response to this observation, we have revised the text in lines 387-390 and included additional comparisons in the relevant figure.

- In Figure 7B, lack of *sfp-1* seems to result in levels of both MUFAs and PUFAs that resemble

Response to Reviewers

unmated. Why do the authors only mention the PUFAs in lines 391-394? These data are more physiological than 7A and it would make sense to move 7A to supplemental.

Reply: We appreciate the reviewer's observation regarding both MUFAs and PUFAs in our data. We have revised the text to include an analysis of both fatty acid classes (Lines 418-425). While previous work demonstrated that MUFA supplementation alleviates mating-induced lifespan shortening (PMID: 34179018), our current findings suggest PUFA metabolism may play a more direct role in SFP-1-mediated effects.

- How do the authors explain that knocking down PUFA biogenesis genes results in more lipids in the over-expression line in Figure 7C and D (when their data support the claim that this over-expression results in reduced levels of these PUFAs on its own)? If you don't have the proteins that make the lipids, how do you have more of the lipids?

Reply: We sincerely appreciate the reviewer's insightful comments. Our data demonstrate that SFP-1 overexpression leads to the depletion of polyunsaturated fatty acids (PUFAs) while knocking down *fat* genes results in more lipids. This phenomenon can be explained by the stepwise nature of fatty acid desaturation and elongation in *C. elegans*, where the *fat* gene family (*fat-1* through *fat-7*) sequentially converts oleic acid (18:1) to various PUFAs through progressive desaturation steps.

The key mechanistic insight is that inhibition of specific *fat* genes does not completely block fatty acid synthesis, but rather prevents the further desaturation and elongation of lipid intermediates. For example, knockdown of *fat-1* or *fat-2* blocks the conversion of oleic acid to more highly unsaturated fatty acids, leading to the accumulation of oleic acid while preventing PUFA depletion. This explains why we observe increased lipid staining.

Our new fatty acid profiling data (Figure 7B) directly show that hermaphrodites mated with *sfp-1* mutant males maintain higher PUFA levels compared to those mated with wild-type males. Furthermore, lifespan experiments demonstrate that *fat-1* and *fat-2* RNAi confer resistance to mating-induced lifespan reduction even in the presence of SFP-1 overexpression (Figure 7E-F), supporting our model that PUFA metabolism may play a more direct role in SFP-1-mediated effects.

These findings suggest that SFP-1 promotes lifespan shortening by accelerating the consumption of PUFAs through the fat-dependent pathway, rather than simply reducing overall lipid synthesis. When this pathway is inhibited (e.g., by *fat* gene RNAi), intermediate lipids accumulate because

Response to Reviewers

they cannot be further metabolized, and this accumulation appears to be protective against lifespan shortening.

- Lines 440-441: *sfp-1* males don't "inhibit" mating-induced lifespan shortening, they fail to induce it (although no direct comparison of lifespan was made between unmated N2 and N2 x *sfp-1* males).

Reply: We sincerely thank the reviewer for their valuable suggestion. To address this point more comprehensively, we have now included unmated N2 controls in Figures 5A and 5C and present the mated and unmated lifespan curves (see below). These data show that mating with *sfp-1* males results in a less severe lifespan reduction compared to mating with wild-type males. However, a significant shortening still occurs relative to unmated controls. This residual effect likely reflects contributions from other mating-associated factors, such as sperm transfer or pheromone exposure. Our revised text (Lines 475-478) clarifies that while SFP-1 mediates the mating-induced lifespan shortening, additional mechanisms further contribute to the observed reduction. This interpretation aligns with our experimental observations and supports the conclusion that SFP-1 plays a key, but not exclusive, role in this process.

Fig.10 for reviewers. Lifespan of mated and unmated N2 worms.

(A) N2 × N2 ♂ : 11.26 ± 0.43 days, n = 41 worms; N2 × *sfp-1* ♂ : 13.18 ± 0.70 days, n = 52 worms, N2 unmated: 16.53 ± 0.81 days, n = 55 worms, ** $P < 0.01$.

- Line 449-450: *skn-1* did not abolish the SFP-1 shortened lifespan; *skn-1* mated to males with SFP-1 (N2) still had shortened lifespans (Figure 5C).

Response to Reviewers

Reply: We sincerely thank the reviewer for raising this important point. We have revised the text to clarify (Lines 485-487). Our data show that the effect of SFP-1 strictly requires SKN-1, since *sfp-1* males extend lifespan in wild-type but not *skn-1* mutants.

Dear Prof. Gong,

Thank you for submitting a revised version of your manuscript. Your study has now been seen by one of the original referees, who finds that the previous concerns have been addressed and now recommend publication of the manuscript after a couple of minor revisions. There remain only a few mainly editorial points that have to be addressed before I can extend formal acceptance of the manuscript:

- Please place the keywords below the Abstract
- Please add the information regarding the following funding in ms file: The outstanding young scientist project of Hubei province 2023AFA060
- Please adjust the format of the reference list and of the in-text citations according to EMBO Journal format (alphabetical order, author name et al + year.../up to 10 author names in the reference list before et al / please refer to our Guide to Authors for additional information on EMBO J reference format).
- Please rename the Conflict of Interest section into "Disclosure and Competing Interests Statement", in accordance with our updated Guide to Authors (<https://www.embopress.org/competing-interests>)
- As we are switching from a free-text author contribution statement towards a more formal statement based on Contributor Role Taxonomy (CRediT) terms, please remove the present Author Contribution section and instead specify each author's contribution(s) directly in the Author Information page of our submission system during upload of the final manuscript. See <https://casrai.org/credit/> for more information.
- In the APPENDIX 1 FILE WITH ToC: The title page should contain "Appendix for + ms title" and a table of content with the page numbers for the listed items
- Please provide suggestions for a short 'blurb' text prefacing and summing up the conceptual aspect of the study in two sentences (max. 250 characters), followed by 3-5 one-sentence 'bullet points' with brief factual statements of key results of the paper; they will form the basis of an editor-written 'Synopsis' accompanying the online version of the article. Please also provide an altered synopsis image, making sure that the aspect ratio conforms to our website's format - it should be exactly 550 pixels wide and between 300-600 pixels high.
- Figure Legends (main + EV):
 1. Please note that the exact p values are not provided in the legends of figures 1F, I, J; 3C, 4A, E; 5B, D, E, F; 6B, D, H, J; 7E-H; EV1 E, F, G; EV3 A, EV4 A-F, G, I, J, K-Q; EV5 B-F.
 2. Please indicate the statistical test used for data analysis in the legends of figures 2C-E; 3C, 4A, C; 5A, C, E; 7E-H; EV1 D, G; EV3 A, C; EV4 A-F, L-Q.
 3. Please note that the error bars are not defined in the legend of figure 6F.
 4. Please note that the measure of center for the error bars needs to be defined in the legends of figures 1F, I, J.
- Please rename the "Material and Methods" section to "Methods"
- EV tables are missing legends, they should be included above the corresponding table
- Section order should be corrected: Title page - Abstract & Keywords - Introduction - Results - Discussion - Methods - Data Availability - Acknowledgements - Disclosure and Competing Interests Statement - References - Figure Legends - Table(s) - Expanded View Figure Legends.

With best regards,
Cornelius Schneider

Editor | The EMBO Journal
c.schneider@embojournal.org

Please refer to our figure preparation guideline in order to ensure proper formatting and readability in print as well as on screen:

See also figure legend guidelines:

<https://www.embopress.org/page/journal/14602075/authorguide#figureformat>

Use the link below to submit your revision:

Referee #1:

The revised manuscript has fully addressed my prior concern regarding the roles of SKN-1 and DAF-16 in SFP-1-mediated post-mating physiological changes. The new findings are exciting! I strongly encourage the authors to present these results in the main figures rather than in the Appendix, as they are central to the manuscript's conclusions. Additionally, I recommend including the schematic representation of the male gonad (currently Fig. 6 for Reviewer A) in the manuscript. Since the current version appears to be directly taken from a previous publication, the authors may need to redraw a similar illustration for inclusion in their own manuscript. This would greatly aid readers who are less familiar with *C. elegans* male anatomy in interpreting the data. With these minor revisions, I fully support the acceptance of the manuscript for publication.

All editorial and formatting issues were resolved by the authors.

Dear Prof. Gong,

I am pleased to inform you that your manuscript has been accepted for publication in the EMBO Journal.

Yours sincerely,

Cornelius Schneider, PhD
Editor
The EMBO Journal
c.schneider@embojournal.org
